# PIM1 promotes hepatic conversion by suppressing reprogramming-induced ferroptosis and cell cycle arrest

Yangyang Yuan[1,2,7,8], Chenwei Wang [3,8], Xuran Zhuang[2,8], Shaofeng Lin [3,8], Miaomiao Luo[1,8], Wankun Deng [3], Jiaqi Zhou [3], Lihui Liu[1], Lina Mao[1], Wenbo Peng[2], Jian Chen [4,5], Qiangsong Wang[1], Yilai Shu [4,5] ✉, Yu Xue [3,6] ✉ & Pengyu Huang [1,2] ✉

Protein kinase-mediated phosphorylation plays a critical role in many biological processes. However, the identification of key regulatory kinases is still a great challenge. Here, we develop a trans-omics-based method, central kinase inference, to predict potentially key kinases by integrating quantitative transcriptomic and phosphoproteomic data. Using known kinases associated with anti-cancer drug resistance, the accuracy of our method denoted by the area under the curve is 5.2% to 29.5% higher than Kinase-Substrate Enrichment Analysis. We further use this method to analyze trans-omic data in hepatocyte maturation and hepatic reprogramming of human dermal fibroblasts, uncovering 5 kinases as regulators in the two processes. Further experiments reveal that a serine/threonine kinase, PIM1, promotes hepatic conversion and protects human dermal fibroblasts from reprogramming-induced ferroptosis and cell cycle arrest. This study not only reveals new regulatory kinases, but also provides a helpful method that might be extended to predict central kinases involved in other biological processes.

Phosphorylation is one of the most important post-translational modifications (PTMs) in proteins. The process is catalyzed by protein kinases (PKs), and has been well documented as a fundamental regulatory mechanism of cellular activities such as signal transduction[1], cell cycle progression[2], autophagy[3], and cell fate determination[4,5]. As rapidly responsive signal transduction processes, changes in protein phosphorylation are among the early events in response to cell lineage reprogramming signals[4–7]. To fully understand the signaling network of cell lineage determination and reprogramming, it is therefore necessary to identify key regulatory PKs responsible for modification of phosphorylation sites (p-sites) in downstream regulators such as transcription factors (TFs) and other proteins.

PKs are regulated by a variety of cellular mechanisms, such as transcription, translation, PTMs, and ligands. Thus, kinome kinetics cannot be fully delineated from a single type of omics data alone, making it challenging to systematically identify PKs that participate in regulation of a defined cellular process. In the past years, researchers have typically conducted functional screens either using small hairpin

[1]Institute of Biomedical Engineering, Chinese Academy of Medical Sciences and Peking Union Medical College, Tianjin 300192, China. [2]School of Life Science and Technology, ShanghaiTech University, Shanghai 201210, China. [3]MOE Key Laboratory of Molecular Biophysics, Hubei Bioinformatics and Molecular Imaging Key Laboratory, Center for Artificial Intelligence Biology, Institute of Artificial Intelligence, College of Life Science and Technology, Huazhong University of Science and Technology, Wuhan 430074 Hubei, China. [4]ENT institute and Department of Otorhinolaryngology, Eye & ENT Hospital, State Key Laboratory of Medical Neurobiology and Institutes of Biomedical Sciences, Fudan University, Shanghai 200031, China. [5]NHC Key Laboratory of Hearing Medicine, Fudan University, Shanghai 200031, China. [6]Nanjing University Institute of Artificial Intelligence Biomedicine, Nanjing, Jiangsu 210031, China. [7]Present address: Centre for Translational Stem Cell Biology Limited, Hong Kong 999077, China. [8]These authors contributed equally: Yangyang Yuan, Chenwei Wang, Xuran Zhuang, Shaofeng Lin, Miaomiao Luo. ✉e-mail: yilai_shu@fudan.edu.cn; xueyu@hust.edu.cn; huangpengyu@yeah.net

RNAs (shRNAs) or kinase inhibitors to identify potentially important PKs. For example, Sakurai et al. individually examined the effects of 3686 shRNA lentiviruses that targeted 734 PK genes to screen PKs that regulates the generation of induced pluripotent stem cell (iPSC). They successfully determined that TESK1, a serine/threonine PK, promotes reprogramming of human fibroblasts to iPSCs[8]. However, such methods are usually expensive, time-consuming, and labor-intensive[6–8], and only a limited number of PKs crucial for cell lineage reprogramming have been identified thus far. A more efficient approach for systematic analysis is of great importance to facilitate understanding of PK regulatory mechanisms.

Here, we integrated quantitative transcriptomic and phosphoproteomic data, and develop a trans-omics-based algorithm, central kinase inference (CKI), to identify potentially central PKs in regulating a defined biological process. Using CKI, we predicted 28 potentially central PKs and successfully validated three catalytic subunits of cAMP-dependent PKs, *Prkaca*, *Prkacb*, and *Prkx*, to promote mouse hepatocyte maturation. Furthermore, we explored the kinome kinetics of hepatic lineage conversion by quantifying transcriptomes and phosphoproteomes of human dermal fibroblasts (HDFs) infected with lentivirus encoding the hepatic reprogramming TFs, *FOXA3*, *HNF1A*, and *HNF4A* (FHH). A serine/threonine PK, PIM1, was identified as a key regulator in promoting hepatic-lineage transition by overcoming cell fate conversion-induced ferroptosis and cell cycle arrest, and acting as a protective molecule by antagonizing the reprogramming barrier. Taken together, this study not only identifies the central PKs, but also develops a computational method that might be helpful for discovery of regulatory PKs associated with other key biological processes.

## Results
### A trans-omics-based method for prediction of central PKs
To infer potentially central PKs in defined biological processes, we developed CKI, a trans-omics-based computational method to analyze and integrate transcriptomic data derived from RNA sequencing (RNA-seq) and phosphoproteomic data quantified by tandem mass tag (TMT) labeling coupled with liquid chromatography-tandem mass spectrometry (LC-MS/MS) (Fig. 1). The basic hypothesis behind CKI was that molecular changes at both the transcriptomic and phosphoproteomic levels might be informative in predicting the functional importance of PKs. Comparisons were made of the transcriptomes and phosphoproteomes in paired samples (control vs. treated), taking three types of changes into consideration to synergistically predict central PKs in response to treatment: mRNA expression, substrate p-site intensity, and kinase-substrate network (Fig. 1), as described below.

PK expression at the transcriptional level is essential for regulating its constitutive activity[9]. Thus, PKs with differentially expressed mRNAs (DEMs) in response to treatment may be involved in orchestrating downstream signaling. In this study, differentially expressed PKs (DEPKs) were directly identified from transcriptomic data (Fig. 1). From the phosphoproteomic data of each sample, directed relations of PKs with p-sites, referred to as site-specific kinase-substrate relations (ssKSRs), were predicted using a previously developed software package called in vivo Group-based Prediction System (iGPS). This program integrates sequence and protein-protein interaction (PPI) information for predicting p-sites specifically modified by 408 human and 416 mouse PKs[10]. Then, we hypothesized that a PK with higher activity might phosphorylate more p-sites with higher modification levels, and vice versa. We therefore developed an intensity-based approach to identify potentially central PKs based on differential intensity of substrate p-sites between paired samples, e.g., treatment vs. control (Fig. 1).

An alternative hypothesis was that a PK with higher activity may produce a more positive impact on its regulatory phosphorylation network, and vice versa. We therefore developed a network-based method to measure the network change for each PK. From predicted

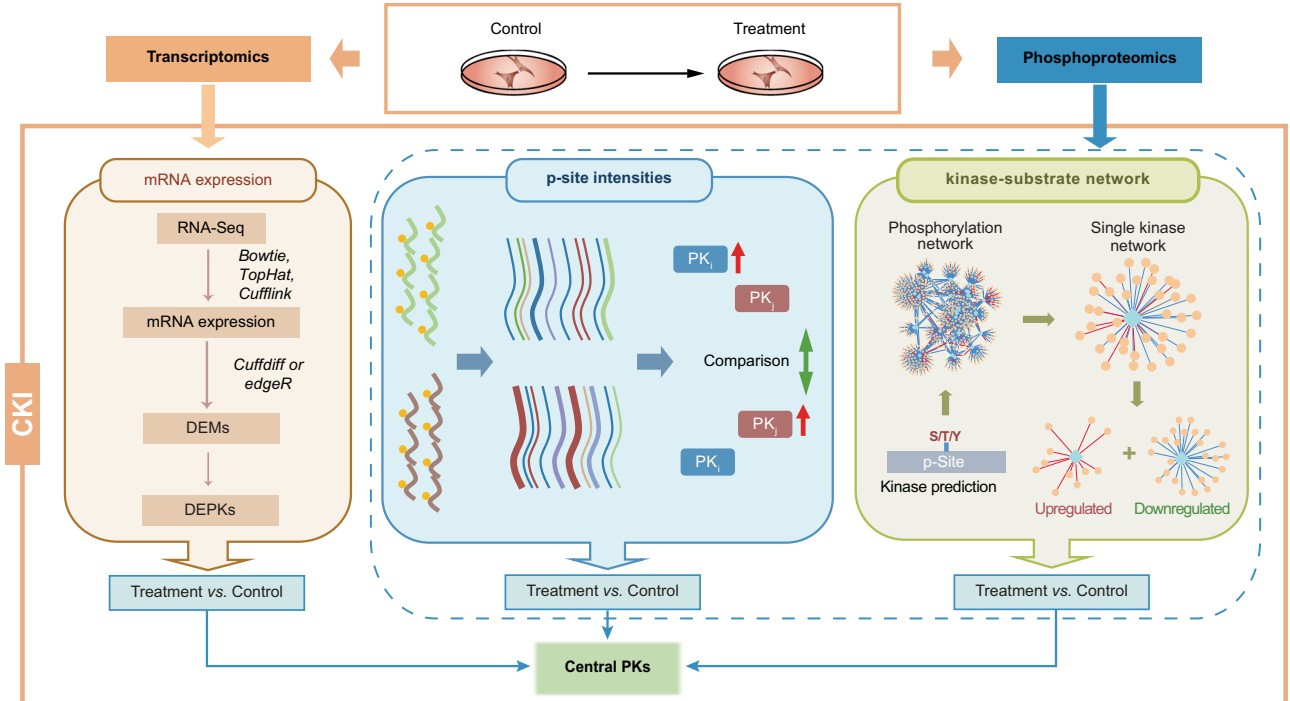

**Fig. 1 | Schematic diagram of the CKI algorithm.** First, the transcriptomes and phosphoproteomes of control and treated samples in a defined biological process were quantified by RNA-seq and TMT-based LC-MS/MS technology. Bowtie[53], TopHat[54], and Cufflinks[55] were used to process the transcriptomic data, then Cuffdiff in Cufflinks[55] or edgeR[49] was used to identify differentially expressed mRNAs (DEMs) and map differentially expressed protein kinases (DEPKs). We developed an intensity-based method and a network-based method to identify differentially altered PKs using the phosphoproteomic data. These three types of data were then combined to synergistically predict potentially central PKs in regulating a defined biological process.

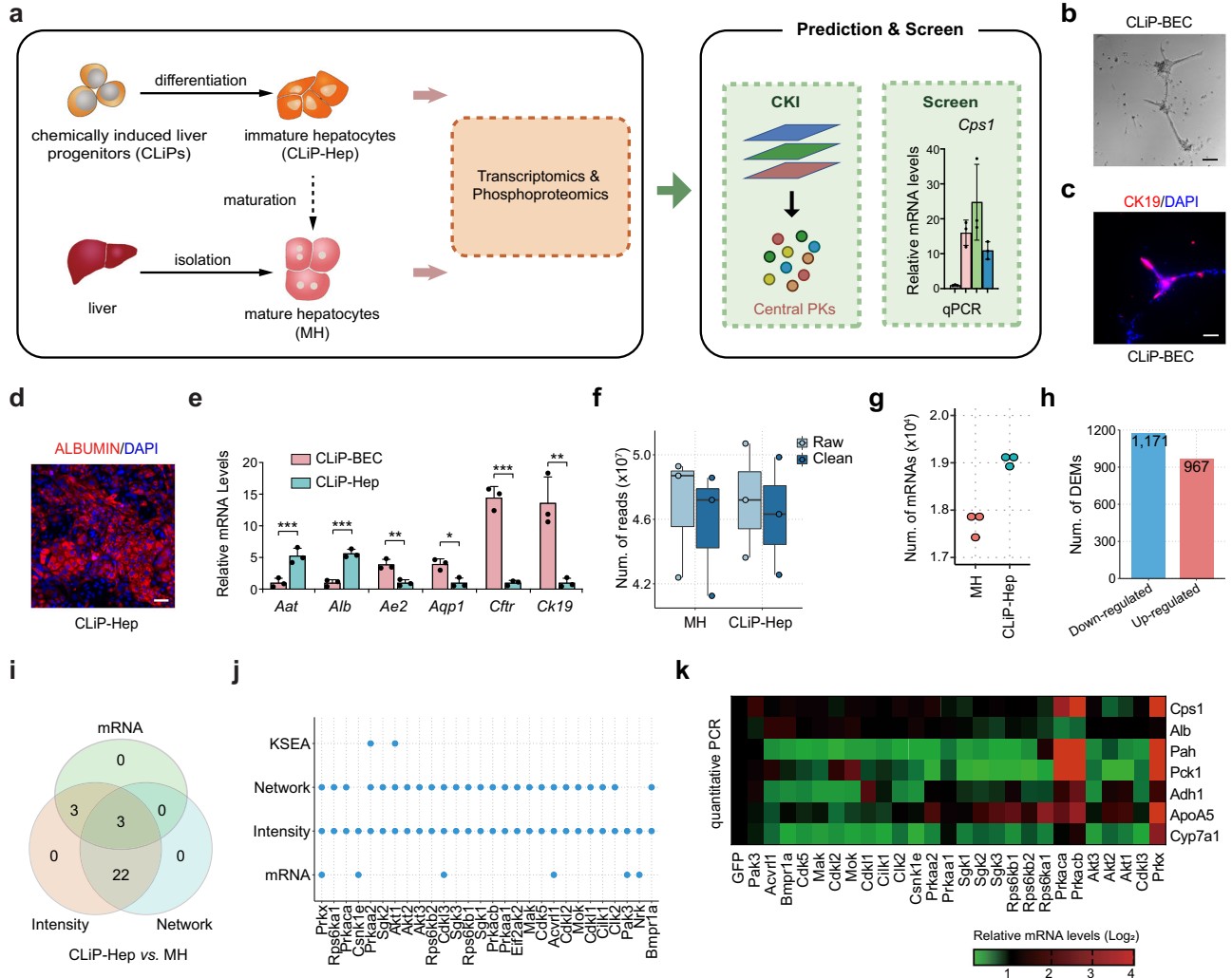

**Fig. 2 | CKI-based analysis of mouse hepatocyte maturation. a** Experimental design of the trans-omics-based analysis of immature hepatocytes generated from liver progenitor cells (CLiP-Hep) and mature hepatocytes (MH) isolated from mouse liver. Representative image of bile duct structure (**b**) and CK19 immunofluorescence staining (**c**) of biliary epithelial cells induced from liver progenitor cells (CLiP-BEC). Scale bars = 100 μm. n = 4 biological replicates. **d** Immunofluorescence staining of ALBUMIN in hepatocytes generated from CLiPs (CLiP-Hep). Scale bars = 100 μm. n = 3 biological replicates. **e** Gene expression analysis by RT-qPCR demonstrated significant differences of hepatic and biliary marker genes between CLiP-BEC and CLiP-Hep cells for *Aat* (p = 0.0067), *Alb* (p = 0.0007), *Ae2* (p = 0.0065), *Aqp1* (p = 0.0109), *Cftr* (p = 0.0002), *Ck19* (p = 0.0065), n = 3 biological samples. Data are shown as the mean + SD. *p < 0.05, **p < 0.01, ***p < 0.001 (unpaired two-sided Student's t-test). **f** Number of raw and clean reads sequenced from MH and CLiP-Hep samples (n = 3 biological samples). Box and whisker plots present the means (lines inside the boxes), the 1st and 3rd quartiles (bottom and top bounds of the boxes), and the extents of the data (whiskers). **g** Number of mapped mRNAs in mouse cell samples (n = 3). **h** Number of up- and down-regulated DEMs in CLiP-Hep compared to MH. **i** Number of potentially central PKs predicted from different data types for CLiP-Hep vs. MH. **j** Comparison of central PKs predicted with CKI, KSEA[15, 16], and the individual datasets comprising CKI. **k** Expression levels of liver metabolic genes in CLiP-Hep cells overexpressing individual candidate central PK as quantified by qRT-PCR (n = 3). Source data are provided as a Source Data file.

ssKSRs mutually quantified in a pair of samples, a kinase-substrate phosphorylation network was re-constructed with directed relations of PKs and p-sites. For each PK, its downstream regulatory network was further split into an up-regulated sub-network (up-regulated substrate p-sites) and a down-regulated sub-network (down-regulated substrate p-sites). The Yate's chi-squared test was performed to identify potentially central PKs statistically associated with up- or down-regulated sub-network modules (Fig. 1). From all pairwise comparisons, the number of positive hits were counted for each PK as the only measure to prioritize the final candidate PKs.

CKI is a model-based method, and no prior data were used for training. To evaluate the performance of CKI, we predicted potentially central PKs using data from two types of previously published drug-resistance studies, doxorubicin (DOX) resistance and genistein resistance (Supplementary Note 1, Supplementary Figs. 1, 2 and Supplementary Data 1)[11–14]. Compared to Kinase-Substrate Enrichment

Analysis (KSEA)[15,16], the receiver operating characteristic (ROC) curves were illustrated, and the area under the curve (AUC) values of CKI were 5.2% (0.8278 vs. 0.7871) and 29.5% (0.7912 vs. 0.6112) higher for DOX and genistein resistance, respectively (Supplementary Fig. 1d, k).

## CKI reveals cAMP-dependent PKs as key regulators for hepatocyte maturation

To further validate the accuracy of CKI, we analyzed the hepatocyte maturation process in mouse liver progenitor cells (Fig. 2a). We chemically induced liver progenitor cells (CLiPs) from mouse hepatocytes using a previously established protocol[17]. The CLiPs could then differentiate into either CK19 + biliary epithelial cells (CLiP-BECs) that can form ductal structures (Fig. 2b, c and e) or ALBUMIN + hepatocytes (CLiP-Heps, Fig. 2d, e). However, CLiP-Heps generated with this differentiation protocol are relatively immature[17], meaning that further CLiP-Hep maturation is required for the application of

liver-progenitor-cell-derived hepatocytes. To identify central PKs that potentially promote the maturation of CLiP-Hep cells, we profiled the transcriptomes and phosphoproteomes of freshly isolated primary mouse hepatocytes (MHs) and CLiP-Hep cells (Fig. 2a).

From the transcriptomic profiling, we detected 2136 DEMs out of 23,558 quantified genes (Fig. 2f–h and Supplementary Data 2). Using pathway and biological process annotations from the Kyoto Encyclopedia of Genes and Genomes (KEGG)[18] and Gene Ontology (GO)[19], respectively, we performed functional enrichment analyses of the DEMs. CLiP-Hep cells were deficient in metabolic pathways compared to MHs (Supplementary Fig. 3a, b). From the phosphoproteomic data, we obtained 10,818 quantified p-sites, including 9478 pS (87.61%), 1251 pT (11.57%), and 89 pY (0.82%) residues in 3575 proteins (Supplementary Fig. 3c, d). To test the quality of the raw MS/MS data, we found that 4128 (44.41%) phosphopeptides could be traced by ≥ 2 spectral counts, with an average spectral count of 2.19 counts per phosphopeptide (Supplementary Fig. 3e). Based on the localization probability (LP) score derived from MaxQuant[20], we identified 9110 class I (LP > 0.75, 84.21%), 1452 class II (0.5 < LP ≤ 0.75, 13.42%), and 256 class III (0.25 ≤ LP ≤ 0.5, 2.37%) p-sites (Supplementary Fig. 3f). From eight public p-site databases, we found that 9839 (90.95%) of the p-sites identified in our data were annotated in at least one database (Supplementary Fig. 3g). The distribution of fragments per kilobase of exon per million fragments mapped (FPKM) values of mRNAs and TMT intensities of p-sites were similar across different samples (Supplementary Fig. 3h, i), suggesting that neither transcriptomes nor phosphoproteomes markedly changed during hepatocyte maturation. Two-way hierarchical clustering was performed by calculating Spearman's correlation coefficient for the transcriptomic or phosphoproteomic data between pairs of samples. The results indicated that CLiP-Hep and MH cells could be unambiguously distinguished (Supplementary Fig. 3j, k).

Using CKI, we predicted 28 potentially central PKs that may promote the maturation of CLiP-Hep cells, and each of the three types of data contributed to the final predictions (Fig. 2i, j and Supplementary Data 2). Again, we found that only one PK activity-associated p-site, Cilk1 T157, was up-regulated in CLiP-Hep cells (Supplementary Fig. 3l, m). Prior to further validations, *Nrk3* was removed because it is too large to be packaged into lentivirus, and *Eif2ak2* was removed because it induced robust cell death. We then screened the 26 remaining candidates by overexpression of individual PKs in liver progenitor cells before induction of hepatic differentiation. Quantitative real-time PCR (qRT-PCR) showed that *Prkaca*, *Prkacb*, and *Prkx*, all of which encode catalytic subunits of cAMP-dependent PKs, promoted the expression of several liver-enriched metabolic genes (Fig. 2k). Previous reports demonstrated that cAMP is critical for the maturation of hepatocytes[21], and our results further identified *Prkaca*, *Prkacb*, and *Prkx* as the key genes in cAMP signaling during hepatocyte maturation. We also analyzed the phosphoproteomic data using KSEA[15,16], which predicted 19 potentially functional PKs, among which the three newly identified PKs were not included (Fig. 2j and Supplementary Fig. 3n). The successful identification of cAMP-dependent PKs as key regulators for hepatocyte maturation further supported the reliability of CKI.

## Trans-omic analyses of early regulatory events during hepatic reprogramming

Next, we set out to investigate central PKs regulating a less studied biological process, hepatic reprogramming. We converted human dermal fibroblasts (HDFs) to hepatocyte-like cells (hiHep cells) by overexpression of *FOXA3*, *HNF1A*, and *HNF4A* (FHH) as previously described[22] (Supplementary Note 2 and Supplementary Fig. 4). We then performed the trans-omic profiling and CKI-based prediction, and validated the candidates to identify potentially central PKs in regulating hepatic reprogramming (Fig. 3a). To systematically interrogate the early regulatory events of hepatic reprogramming, we quantified transcriptomes and phosphoproteomes of HDFs 2.25 days (FHH-2.25d)

and 5 days (FHH-5d) after FHH transduction. These time points were chosen because liver-specific genes were starting to be induced in FHH-2.25d cells and were induced at much higher levels in FHH-5d cells. However, the expression levels of ALBUMIN and E-Cadherin in HDF + FHH (5d) are far below those in HDF + FHH (14d), and are not detectable by immunofluorescence staining (Supplementary Fig. 4e). It is impossible to isolate hiHep cells by FACS sorting during the early stage of hepatic conversion. Thus, the pooled cells were used for the study. HDFs transduced with GFP for 2.25 days (GFP) were used as the control for the trans-omic profiling (Fig. 3a).

From the transcriptomic data, we identified 136, 1044, and 767 DEMs in comparing FHH-2.25d vs. GFP, FHH-5d vs. GFP, and FHH-5d *vs.* FHH-2.25d, respectively (Supplementary Fig. 5a–d, and Supplementary Data 3, $p < 0.01$, 2-sided negative binomial test). Interestingly, we observed that the numbers and expression levels of DEMs increased over time after FHH infection (Supplementary Fig. 5c, d). From the phosphoproteomic profiling, we quantified 5031 unique phosphopeptides from HDFs transduced with FHH or GFP (Supplementary Fig. 5e and Supplementary Data 3). We found that 2477 (49.23%) of the phosphopeptides were supported by ≥2 MS/MS spectra with an average spectral count of 2.89 (Supplementary Fig. 5f). We mapped these phosphopeptides to human protein sequences, and detected 5660 non-redundant p-sites in 2260 phosphoproteins, including 4985 pS (88.08%), 646 pT (11.41%), and 29 pY (0.51%) residues (Supplementary Fig. 5g and Supplementary Data 3). Based on the LP score[20], we obtained 4845 class I (85.60%), 674 class II (11.91%), and 141 class III (2.49%) p-sites (Supplementary Fig. 5h). Using data from nine public p-site databases, we found that 5392 (95.26%) of the p-sites identified in our data were annotated in at least one database (Supplementary Fig. 5i). More detailed analyses of the trans-omic data were also present (Supplementary Note 2 and Supplementary Fig. 5j–p).

Through pairwise comparisons of transcriptomes or phosphoproteomes of GFP, FHH-2.25d, and FHH-5d samples, CKI integrated all predictions from the nine pairwise comparisons and prioritized 15 PKs that were detected in ≥5 pairwise comparisons as the final candidates (Supplementary Fig. 5q, and Supplementary Data 4). For the 15 candidate PKs, the corresponding *p* values were calculated for changes in mRNA expression, substrate p-site intensity, and kinase-substrate network (Fig. 3b). The mRNA levels of the 15 PKs were not markedly altered in the early stage of hepatic reprogramming (FHH-2.25d vs. GFP), and did not contribute to the final predictions (Fig. 3b). Additional analyses were performed to test the reliability of CKI predictions (Supplementary Note 2 and Supplementary Fig. 5r, s).

Prior to further experimental validation, we excluded *KALRN* and *MAPK4* from the study, because *KALRN* is too large to be packaged into lentivirus and *MAPK4* transcripts were not detected during hepatic reprogramming. We overexpressed each of the remaining 13 candidate PKs in HDFs together with FHH to analyze their effects on hepatic conversion. We found that PIM family genes, including *PIM1*, *PIM2* and *PIM3*, showed potent effects on the induction of hepatic-lineage genes (Fig. 3c). We also performed ALBUMIN immunofluorescence staining and validated four PKs that showed statistically significant regulation of *ALB* expression upon overexpression (Fig. 3d).

Moreover, shRNA-mediated knockdown of *PIM1* or *PIM2* markedly decreased the expression of hepatic-lineage genes, further confirming that two members of the PIM family of PKs were important regulators for hepatic reprogramming (Fig. 3e and Supplementary Fig. 6a). Overexpression of *PLK1*, *PLK2* or *NEK2* did not have an obvious effect on the expression of hepatic-lineage genes (Fig. 3c). However, knockdown of *PLK1*, *PLK2* or *NEK2* led to significant inhibition of hepatic conversion (Fig. 3e and Supplementary Fig. 6a–d). Although most of the candidate central PKs showed more or less effects on the regulation of hepatic-lineage genes, two of the 15 predicted central PKs, *PIM1* and *PIM2*, were validated to regulate all examined three hepatic-lineage genes in both overexpression and knockdown experiments.

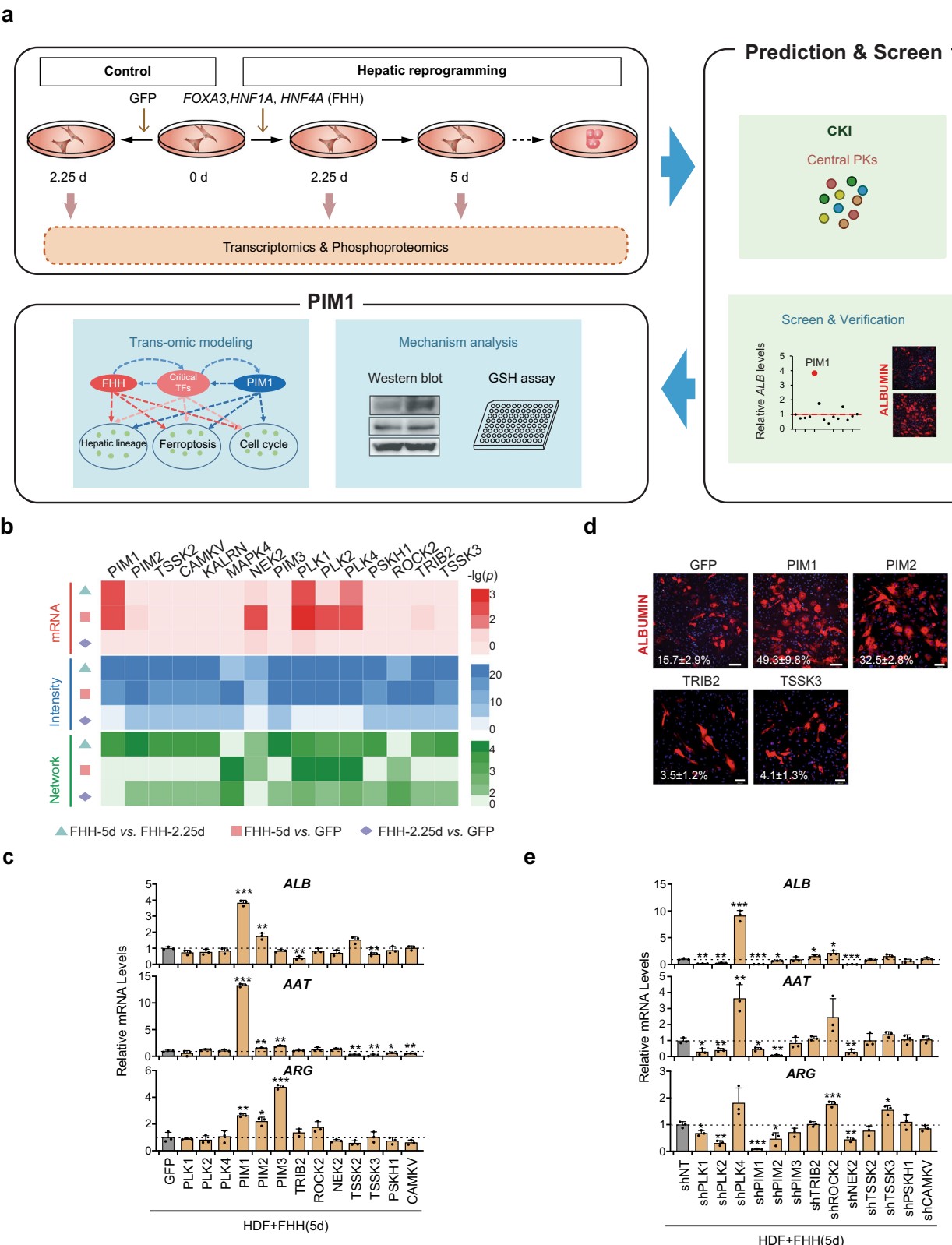

**Fig. caption area (a–e panels)**

a: Control / Hepatic reprogramming — GFP; FOXA3, HNF1A, HNF4A (FHH); time points 2.25 d, 0 d, 2.25 d, 5 d; Transcriptomics & Phosphoproteomics; Prediction & Screen — CKI Central PKs; Screen & Verification — Relative ALB levels, PIM1, ALBUMIN; PIM1 — Trans-omic modeling (FHH, Critical TFs, PIM1; Hepatic lineage, Ferroptosis, Cell cycle); Mechanism analysis — Western blot, GSH assay.

b: Heatmap mRNA / Intensity / Network across PIM1, PIM2, TSSK2, CAMKV, KALRN, MAPK4, NEK2, PIM3, PLK1, PLK2, PLK4, PSKH1, ROCK2, TRIB2, TSSK3. Legends: ▲ FHH-5d vs. FHH-2.25d, ■ FHH-5d vs. GFP, ♦ FHH-2.25d vs. GFP.

c: Relative mRNA Levels — ALB, AAT, ARG; HDF+FHH(5d).

d: ALBUMIN — GFP 15.7±2.9%, PIM1 49.3±9.8%, PIM2 32.5±2.8%, TRIB2 3.5±1.2%, TSSK3 4.1±1.3%.

e: Relative mRNA Levels — ALB, AAT, ARG; HDF+FHH(5d).

*PLK1*, *PLK2* and *NEK2* are also important for hepatic reprogramming, though overexpression of these genes cannot further improve the efficiency.

## PIM1 promotes hepatic reprogramming

There are three human PIM family genes that encode serine/threonine PKs with multiple protein substrates[9]. Previously, PIM1 was reported to act as a downstream effector of interleukin (IL)−6 to enhance reprogramming to iPSCs[23]. In this study, we observed that during the early stage of hepatic reprogramming, PIM1 was markedly induced five days after introduction of FHH to HDFs (Fig. 4a, b). Importantly, inhibition of *PIM1* by shRNA significantly suppressed induction of hepatic genes, storage of glycogen, and uptake of acetylated low density lipoprotein (ac-LDL) (Fig. 4c, d).

**Fig. 3 | CKI-based analysis of hepatic reprogramming. a** Experimental design of the trans-omics-based analysis of HDFs undergoing hepatic reprogramming by overexpression of FHH. **b** Heatmap of 15 potentially central PKs predicted by CKI using a threshold of ≥5 of 9 pairwise comparisons; the minus-log transformed $p$ values were calculated for the indicated comparisons. **c** Expression levels of hepatic functional genes in HDFs overexpressing FHH and individual candidate central PKs for 5 days as quantified by qRT-PCR ($n = 3$). Overexpression of *PIM1* (*ALB* $< 0.0001$, *AAT* $p < 0.0001$, *ARG* $p = 0.0020$), *PIM2* (*ALB* $p = 0.0043$, *AAT* $p = 0.0071$, *ARG* $p = 0.0133$), *PIM3* (*AAT* $p = 0.0022$, *ARG* $p < 0.0001$), *TRIB2* (*ALB* $p = 0.0017$), *TSSK2* (*ALB* $p = 0.0163$, *AAT* $p = 0.0027$), *TSSK3* (*ALB* $p = 0.0077$, *AAT* $p = 0.0048$), *PSKH1* (*AAT* $p = 0.0199$), or *CAMKV* (*AAT* $p = 0.0081$) showed significantly changed transcript levels of hepatic functional genes. **d** Representative image of ALBUMIN

immunofluorescence staining in HDFs overexpressing the indicated genes after infection with FHH for 12 days. Scale bars = 100 μm. $n = 2$ biological replicates. **e** Expression levels of hepatic functional genes in HDFs overexpressing FHH and shRNAs of candidate central PKs for 5 days as quantified by qRT-PCR ($n = 3$). Knockdown of *PLK1* (*ALB* $p = 0.0011$, *AAT* $p = 0.0104$, *ARG* $p = 0.0227$), *PLK2* (*ALB* $p = 0.0038$, *AAT* $p = 0.0084$, *ARG* $p = 0.0010$), *PLK4* (*ALB* $p = 0.0001$, *AAT* $p = 0.0070$), *PIM1* (*ALB* $p = 0.0006$, *AAT* $p = 0.0113$, *ARG* $p = 0.0002$), *PIM2* (*ALB* $p = 0.0398$, *AAT* $p = 0.0010$, *ARG* $p = 0.0222$), *TRIB2* (*ALB* $p = 0.0392$), *ROCK2* (*ALB* $p = 0.0125$, *ARG* $p = 0.0010$), or *TSSK3* (*AAT* $p = 0.0459$, *ARG* $p = 0.0101$) showed significantly changed transcript levels of hepatic functional genes. Data are shown as the mean + SD. *$p < 0.05$, **$p < 0.01$, ***$p < 0.001$ (unpaired two-sided Student's $t$-test). Source data are provided as a Source Data file.

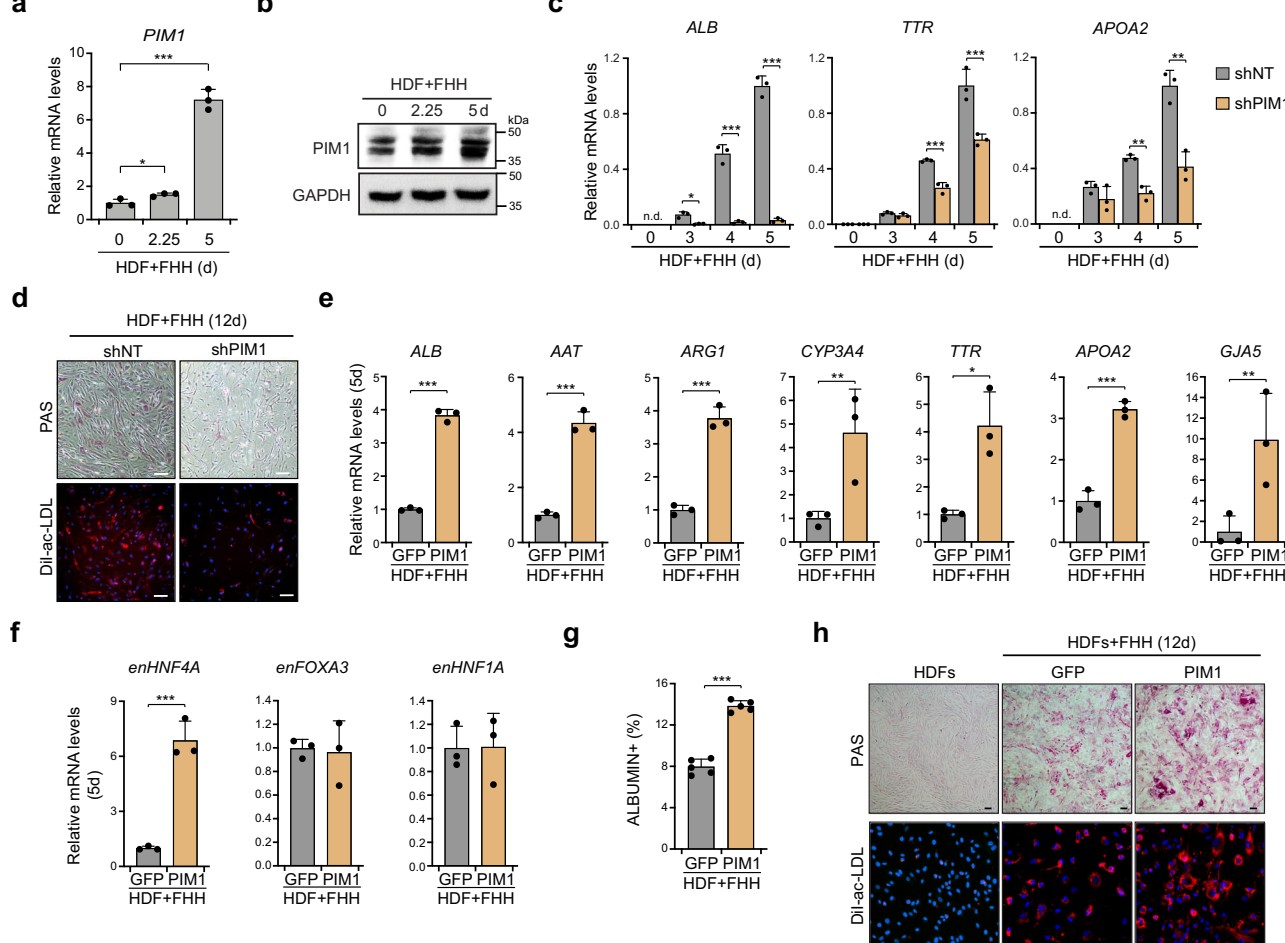

**Fig. 4 | Experimental analysis of PIM1 roles in hepatic reprogramming. a** Transcript levels of *PIM1* significantly increased during hepatic reprogramming as quantified by RT-qPCR (2.25d $p = 0.0251$, 5d $p < 0.0001$, $n = 3$). **b** Immunoblotting of PIM1 in HDFs infected with FHH for the indicated number of days. GAPDH was used as the reference protein. $n = 3$ biological replicates. **c** Expression of hepatic genes in HDFs infected with FHH and PIM1 shRNA as quantified by qRT-PCR ($n = 3$). A non-targeted shRNA (shNT) was used as the control. Knockdown of PIM1 showed significantly decreased transcript levels of *ALB* (3d $p = 0.0026$, 4d $p = 0.0002$, 5d $p < 0.0001$), *TTR* (4d $p = 0.0009$, 5d $p = 0.0055$), *APOA2* (4d $p = 0.0012$, 5d $p = 0.0028$). **d** PAS staining and DiI-ac-LDL uptake assay of HDFs infected with FHH and either PIM1 shRNA or non-targeted shRNA for 12 days. Scale bars = 100 μm. $n = 3$ biological replicates. Expression of hepatic functional genes (**e**) and endogenous

FHH TFs (**f**) in HDFs infected with FHH and either PIM1 or GFP for 5 days as quantified by qRT-PCR ($n = 3$). Overexpression of PIM1 increased the transcript levels of *ALB* ($p < 0.0001$), *AAT* ($p = 0.0002$), *ARG2* ($p = 0.0002$), *CYP3A4* ($p = 0.0292$), *TTR* ($p = 0.0109$), *APOA2* ($p = 0.0003$), *GJA5* ($p = 0.0321$), endogenous *HNF4A* ($p = 0.0007$). **g** Quantification of ALBUMIN⁺ cells by immunofluorescence staining in *GFP*- or *PIM1*-overexpressing HDFs after infection with FHH for 12 days ($p < 0.0001$, $n = 5$). **h** PAS staining and DiI-ac-LDL uptake assay of *GFP*- or *PIM1*-overexpressing HDFs after infection with FHH for 12 days. Scale bars = 100 μm. $n = 3$ biological replicates. Data are shown as the mean + SD. *$p < 0.05$, **$p < 0.01$, ***$p < 0.001$ (unpaired two-sided Student's $t$-test). Source data are provided as a Source Data file.

To interrogate the roles of PIM1 in hepatic reprogramming, we overexpressed *PIM1* in HDFs before induction of hepatic conversion. After FHH infection, HDFs expressing *PIM1* showed significantly increased expression of hepatic functional genes, including the

secretory protein genes *ALB* and *TTR*, lipid metabolism gene *APOA2*, drug metabolism gene *CYP3A4*, urea cycle gene *ARG1*, and gap junction gene *GJA5* (Fig. 4e). Importantly, increased activation of endogenous *HNF4A*, a member of FHH and a master regulator of hepatic functional

genes, was detected in HDFs five days after infection, suggesting an accelerated activation of the endogenous TF (Fig. 4f). Immuno-fluorescence staining also showed that the proportion of ALBUMIN positive cells was increased in response to *PIM1* overexpression (Fig. 4g and Supplementary Fig. 7). Overexpression of PIM1 additionally promoted glycogen storage and DiI labelled acetylated low-density lipoprotein (DiI-ac-LDL) uptake of hiHep cells (Fig. 4h). These results demonstrated that *PIM1* efficiently promotes hepatic reprogramming.

## PIM1 promotes cell proliferation and suppresses ferroptosis

To interrogate how PIM1 regulates hepatic reprogramming, we profiled the transcriptomes of HDFs undergoing hepatic conversion with *PIM1* or *GFP* overexpression (Fig. 5a and Supplementary Data 5). In total, we identified 2123 DEMs (Supplementary Fig. 8a and Supplementary Data 5). The similar distribution of FPKM values in the two transcriptomes suggested that *PIM1* overexpression did not induce a global change in mRNA expression (Supplementary Fig. 8b). Two-way hierarchical clustering of the transcriptomic data demonstrated that FHH-transduced HDFs with *PIM1* or *GFP* overexpression could be clearly distinguished from one another (Supplementary Fig. 8c). Among the 45 genes most significantly up-regulated in response to *PIM1* overexpression, 21 were liver- or fetal liver-enriched genes (according to the Molecular Signature Database [MSigDB][24]; Fig. 5b and Supplementary Data 5). No liver- or fetal liver-enriched genes were found among the down-regulated genes. This result further supported a critical role of PIM1 in hepatic reprogramming.

The early stage of cell fate conversion is crucial for successful generation of target cell types. It has been reported that several molecular barriers are activated during the early stage of cell fate conversion to block robust epigenetic remodeling in response to enforced expression of lineage-specific TFs, induce cell cycle arrest and cell death, and inhibit cell fate conversion[25,26]. Here, we performed KEGG- and GO-based enrichment analyses, and found that 17 pathways were significantly changed upon PIM1 overexpression (Fig. 5c and Supplementary Fig. 8d). Fifteen were liver-enriched metabolic or functional pathways (e.g., the PPAR signaling pathway, drug metabolism, pentose and glucuronate interconversions, and other similar pathways). As we found that PIM1 promotes hepatic reprogramming, it is not surprising that pathways related to liver functions were enriched. Interestingly, we observed that two non-metabolic pathways, cell cycle (KEGG ID: hsa04110) and ferroptosis (KEGG ID: hsa04216), were enriched in response to PIM1 overexpression (Fig. 5c), which is consistent with a previous report[26]. We also observed a relatively reduced cell number of HDFs expressing FHH, which could be rescued by *PIM1* overexpression (Fig. 5d).

Previously, PIM1 was shown to promote proliferation of several cell lineages both in vitro and in vivo[9]. Thus, we investigated whether PIM1 could rescue reprogramming-induced cell cycle arrest. We performed EdU staining, which demonstrated significantly suppressed proliferation of HDFs after infection with FHH (Fig. 5e, f and Supplementary Fig. 8e, f). However, PIM1 overexpression rescued FHH-induced suppression of cell proliferation, and particularly promoted the proliferation of ALBUMIN + cells (Fig. 5g, h and Supplementary Fig. 8e–g)[26]. PIM1 has been found to promote cell proliferation through various downstream signaling pathways, including phosphorylation of 4E-BP1 to promote cap-dependent translation, phosphorylation on Ser62 to stabilize MYC protein, and activation of mTOR signaling[9,27]. Here, we also found that increased protein and phosphorylation levels of 4E-BP1 and MYC occurred with increased expression of PIM1 during hepatic reprogramming (Fig. 5i). Overexpression of PIM1 further increased the phosphorylation levels of 4E-BP1, MYC, and an mTOR substrate P70S6K (Fig. 5j). The increased MYC phosphorylation on Ser62 possibly stabilized MYC protein as previously reported[28]. 4E-BP1 has also been reported to be stabilized by multiple kinases[29]. Whether 4E-BP1 could be stabilized by PIM1 should be further investigated in the

future. Furthermore, we also observed a decrease in MYC expression and phosphorylation after *PIM1* knockdown during hepatic reprogramming (Supplementary Fig. 8h).

Another important function of PIM1 is to promote cell survival[30]. Thus, we investigated whether *PIM1* could rescue hepatic reprogramming-induced cell death. Consistent with a previous report[26], we observed increased cell death during hepatic reprogramming (Fig. 6a, b and Supplementary Fig. 9a). Ferroptosis-related genes were significantly induced during hepatic reprogramming (Fig. 6c), and knockdown of *PIM1* further increased hepatic reprogramming-induced cell death and expression of ferroptotic genes (Fig. 6d, and Supplementary Fig. 9b, c). In contrast, *PIM1* overexpression significantly suppressed cell death and decreased expression of ferroptotic genes (Fig. 6e, and Supplementary Fig. 9d, e). The cell death induced by FHH could be inhibited by ferroptosis inhibitors ferrostatin-1 and liproxstatin-1, but not necroptosis inhibitor Nec-1s or apoptosis inhibitor z-VAD. This supported the hypothesis that ferroptosis contributed to hepatic reprogramming-induced cell death (Fig. 6f, Supplementary Fig. 9f).

To investigate whether PIM1 is involved in suppressing ferroptosis, we investigated the effect of PIM1 in a widely used ferroptosis experimental system. U-2 OS cells overexpressing *GFP* or *PIM1* were treated with RSL3, an inhibitor of glutathione peroxidase 4 (GPX4), to induce ferroptosis. We observed that PIM1 significantly suppressed ferroptosis in U-2 OS cells (Supplementary Fig. 9g). Thus, the suppressing role of PIM1 in ferroptosis is not restricted to hepatic conversion.

To investigate whether suppression of ferroptosis promoted hepatic reprogramming, we treated HDFs with ferroptosis inhibitors, namely N-acetyl cysteine (NAC), ferrostatin-1 (Fer-1), and liproxstatin-1 (Lip-1), after infection with FHH. We found that all three tested ferroptosis inhibitors significantly promoted the expression of hepatic functional genes (Fig. 6g). These results supported the conclusions that cell reprogramming-induced ferroptosis suppressed hepatic cell fate conversion and that PIM1 functioned as a ferroptosis suppressor during hepatic reprogramming.

Ferroptosis is characterized by the depletion of glutathione in cells[31]. Thus, the expression of *Slc7a11* is usually increased to generate more glutathione in response to ferroptotic stimuli[32]. Treatment of cells with the glutathione precursor NAC can inhibit ferroptosis[33]. PIM1 has been shown to increase glucose metabolism[34], which may promote generation of NADPH and glutathione. Indeed, we observed increased generation of NADPH and glutathione after overexpression of *PIM1* in HDFs (Supplementary Fig. 9h, i). Next, we examined HDFs expressing *GFP* or *PIM1* after FHH infection, and found that NADPH and glutathione were significantly increased in response to *PIM1* overexpression (Fig. 6h, i). Knockdown of *PIM1* also decreased the NADPH/NADP + ratio and the generation of glutathione (Supplementary Fig. 9j, k). These results suggest that PIM1 promotes generation of NADPH independent of the hepatic reprogramming process.

Together, these results suggest that PIM1 suppresses ferroptosis by increasing generation of glutathione. Overall, PIM1 rescued hepatic reprogramming-induced cell cycle arrest and ferroptosis by activating MYC and increasing glutathione generation.

## Network analysis of the signal web during hepatic reprogramming

Theoretically, enforced expression of FHH would transcriptionally up-regulate numerous target genes, including TFs and central PKs, in either a direct or indirect manner. Transcriptionally-induced central PKs could be activated to phosphorylate protein substrates, including TFs. Thus, key genes may be doubly regulated by TFs and central PKs at both the transcription and phosphorylation levels to drive hepatic reprogramming. Here, we modeled a transcription-phosphorylation collaborative web (TPCW) that contained the regulatory relationships

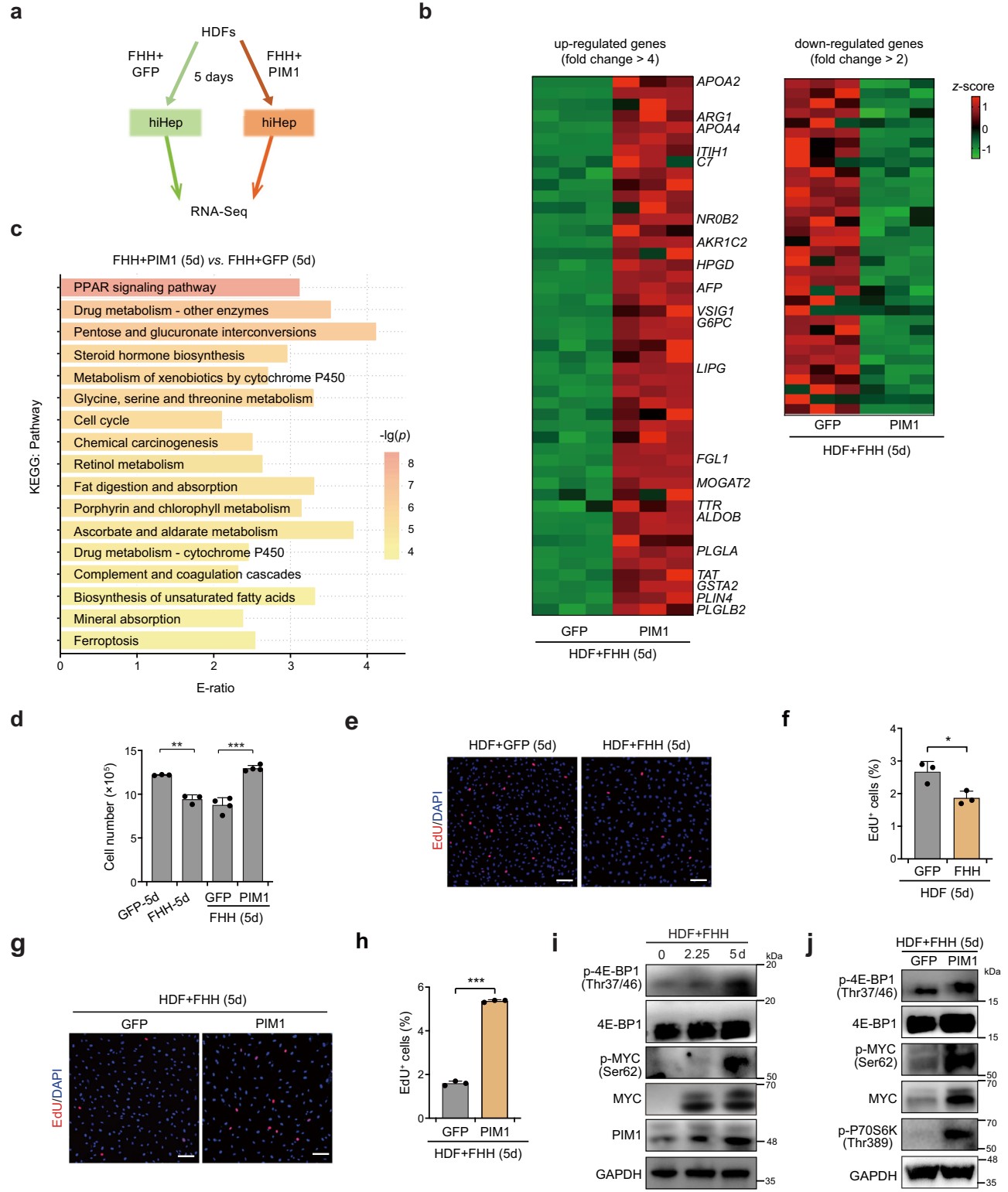

**Fig. 5 | Analysis of PIM in reprogramming-induced cell cycle arrest. a** Diagram of the experimental design. **b** Up-regulated genes (fold change > 4) and down-regulated genes (fold change > 2) in response to PIM1 overexpression in HDFs transfected with FHH for 5 days ($n = 3$). The names of liver- or fetal liver-enriched genes (based on BioGPS data) are shown at right. **c** KEGG-based enrichment analysis of DEMs derived from *GFP*- or *PIM1*-overexpressing HDFs infected with FHH for 5 days. **d** Cell numbers after introduction of the indicated genes into HDFs for 5 days (GFP-5d vs. FHH-5d $p = 0.0007$ $n = 3$, FHH+GFP(5d) vs. FHH+PIM1(5d) $p < 0.0001$ $n = 4$). Representative image (**e**) of EdU positive cells on day 3 of hepatic reprogramming using the EdU staining assay ($n = 3$ biological replicates). The EdU+ cells were quantified by a flow cytometer ($p = 0.0224$, $n = 3$ biological replicates) (**f**).

Representative image (**g**) of EdU positive cells on day 3 of hepatic transdifferentiation with *GFP* or *PIM1* overexpression using the EdU staining assay ($n = 3$ biological replicates). The EdU+ cells were quantified by a flow cytometer ($p < 0.0001$, $n = 3$ biological replicates) (**h**). **i** Immunoblotting of PIM1 downstream substrate proteins in HDFs undergoing hepatic reprogramming. $n = 2$ biological replicates. **j** Immunoblotting of PIM1 downstream substrate proteins in HDFs undergoing hepatic reprogramming with *GFP* or *PIM1* overexpression. GAPDH was used as the loading control. $n = 2$ biological replicates. Data are shown as the mean + SD. *$p < 0.05$, **$p < 0.01$, ***$p < 0.001$ (unpaired two-sided Student's $t$-test). Source data are provided as a Source Data file.

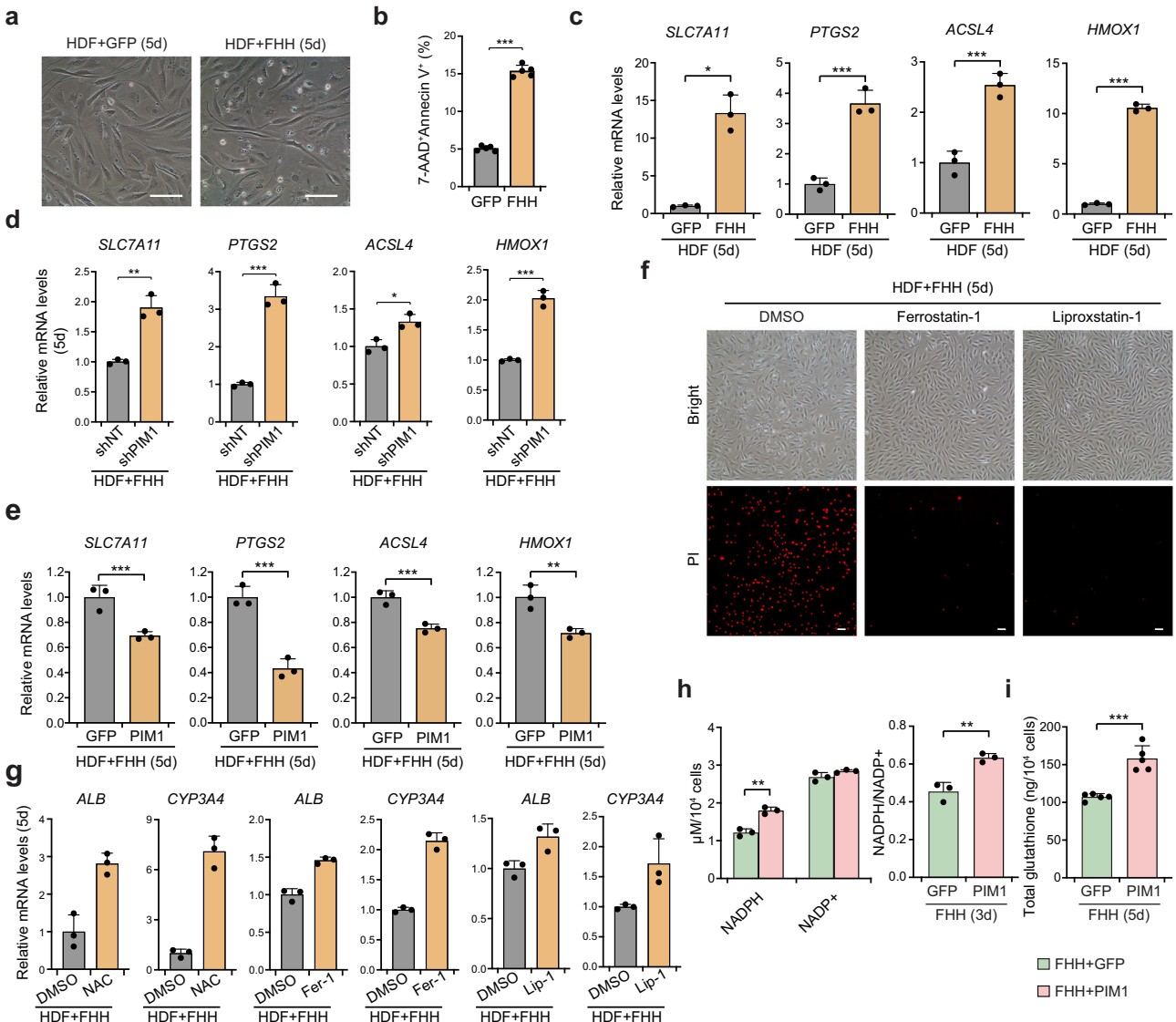

**Fig. 6 | PIM1 suppresses hepatic reprogramming-induced ferroptosis.**
**a** Representative image of FHH-induced cell death of HDFs on day 5. Scale bars = 100 μm. *n* = 4 biological replicates. **b** FHH-induced cell death was analyzed by Annexin V/7-AAD staining on day 5. Proportions of 7-AAD/Annexin V positive cells were quantified (*p* < 0.0001, *n* = 5). **c** Expression levels of ferroptosis-related genes were increased on day 5 of hepatic transdifferentiation (*SLC7A11* *p* = 0.0009, *PTGS2* *p* = 0.0007, *ACSL4* *p* = 0.0013, *HMOX1* *p* < 0.0001, *n* = 3). **d** Knockdown of *PIM1* increased the transcript levels of ferroptosis genes on day 5 of hepatic transdifferentiation (*SLC7A11* *p* = 0.0015, *PTGS2* *p* = 0.0002, *ACSL4* *p* = 0.0121, *HMOX1* *p* = 0.0002, *n* = 3). **e** Overexpression of *PIM1* decreased the transcript levels of ferroptosis genes on day 5 of hepatic transdifferentiation (*SLC7A11* *p* = 0.0062, *PTGS2* *p* = 0.0011, *ACSL4* *p* = 0.0025, *HMOX1* *p* = 0.0080, *n* = 3). The samples were the same

with those used in Fig. 4e, f. **f** Propidium iodide (PI) staining assay showing that treatment with the ferroptosis inhibitors ferrostatin-1 and liproxstatin-1 inhibited FHH-induced cell death in HDFs. Scale bars = 100 μm. *n* = 2 biological replicates. **g** Treatment of NAC, Fer-1, or Lip-1 increased the transcript levels of liver-specific genes on day 5 of hepatic transdifferentiation (NAC treatment: *ALB* *p* = 0.0041, *CYP3A4* *p* = 0.0004; Fer-1 treatment: *ALB* *p* = 0.0008, *CYP3A4* *p* = 0.0001; Lip-1 treatment: *ALB* *p* = 0.0198, *CYP3A4* *p* = 0.0404; *n* = 3). **h** Cellular NADP/NADPH levels on day 3 of hepatic transdifferentiation with *GFP* or *PIM1* overexpression (NADPH *p* = 0.0015, NADPH/NADP+ *p* = 0.0041, *n* = 3). **i** Cellular GSH levels on day 5 of hepatic transdifferentiation with *GFP* or *PIM1* overexpression (*p* = 0.0002, *n* = 3). Data are shown as the mean + SD. *\*p* < 0.05, *\*\*p* < 0.01, *\*\*\*p* < 0.001 (unpaired two-sided Student's *t*-test). Source data are provided as a Source Data file.

among FHH, 24 FHH-regulated TFs, two screened central PKs, and 60 curated genes related to hepatic lineage, ferroptosis, and cell cycle that are potentially modulated by these regulators (Fig. 7a and Supplementary Data 6).

In the TPCW, 24 downstream TFs that were predicted to be directly regulated by FHH were found to be significantly up-regulated during the early stage of hepatic conversion (Fig. 7a). We counted predicted transcription factor binding sites (TFBSs) in the upstream and downstream regions of the 24 TFs; one TF, *MAFF*, had only one predicted TFBS (Supplementary Fig. 10a). Of the 24 TFs, *MAFF* and *HLF* have previously been reported to contribute to hepatic lineage determination in a massively parallel protein activity assay[35]. *SOX17* is

also a critical TF for specification of the endodermal lineage[36]. In particular, many hepatic lineage genes, ferroptosis genes, and cell cycle genes were also transcriptionally regulated by FHH and other TFs. Thus, FHH seems to trigger hepatic cell fate in concert with the TFs they activate downstream.

The TPCW also included a central kinome of two PKs that were experimentally confirmed to significantly regulate the hepatic conversion process in this study. Both of the PKs were predicted to be transcriptionally regulated by FHH and/or the other 24 TFs, either by direct or indirect mechanisms (Fig. 7a). In return, the two PKs could regulate FHH and the other TFs, and at least 10 TFs were predicted to be phosphorylated by one or both central PKs (Fig. 7a). In addition, the

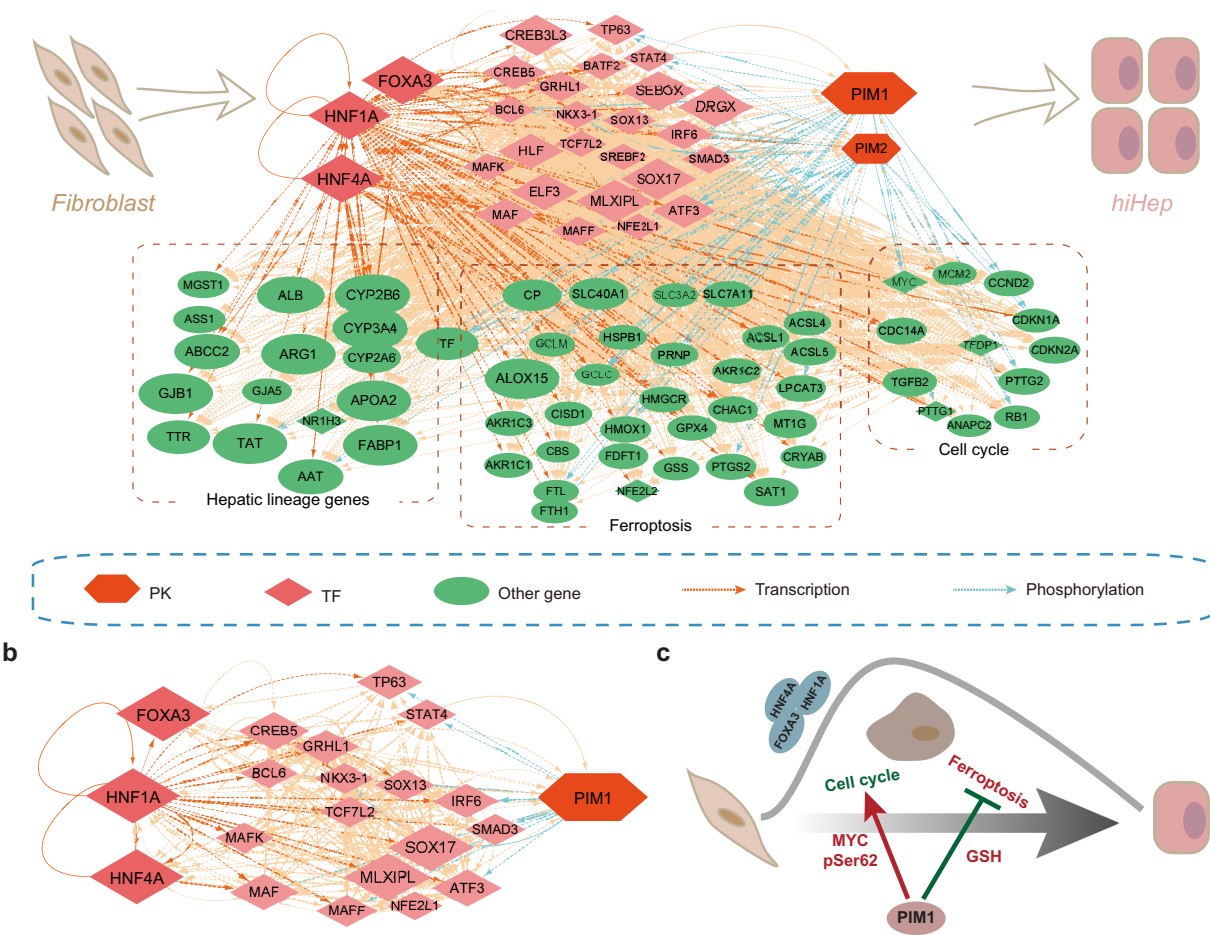

**Fig. 7 | Network analysis of the major molecular landscape in hepatic reprogramming. a** An integrative TPCW based on the findings of this study and existing knowledge illustrates the regulatory relationships among FHH, 24 FHH-regulated TFs, two screened central PKs, and 60 curated genes related to hepatic lineage, ferroptosis, and cell cycle progression. **b** PIM1-centered subnetwork during hepatic lineage reprogramming. **c** Working model showing potential roles of PIM1 in hepatic lineage reprogramming.

two central PKs were predicted to phosphorylate three hepatic lineage proteins, 11 ferroptosis proteins, and seven cell cycle proteins. Thus, the network of transcription and phosphorylation regulation represented a molecular landscape for the early stage of hepatic conversion. The effects of the two PKs in hepatic conversion were experimentally assessed by overexpression or shRNA-mediated gene knockdown. Also, PIM1 and PIM2 were found to promote hepatic conversion (Fig. 3c, e), possibly by phosphorylating downstream TFs and other signaling proteins involved in hepatic conversion.

Based on experimental results, the most potent central PK was PIM1 (Fig. 3c, e). *PIM1* was not predicted to be directly regulated by FHH at the transcriptional level, because there was no FHH binding site found in the promoter region of *PIM1*. However, up to 11 TFs downstream of FHH were predicted to transcriptionally regulate *PIM1* (Fig. 7b and Supplementary Fig. 10b), which may explain the relatively late activation of *PIM1* on day 5 of hepatic conversion (Fig. 4a, b). Direct regulation of PIM1 by SOX17 or BCL6 has previously been reported in different experimental systems[37,38]. Importantly, the TPCW indicated that PIM1 may reciprocally phosphorylate 10 TFs, including NKX-3.1, SMAD3, and MAFF, possibly reinforcing the functions of these TFs (Fig. 7a, b). Indeed, previous studies have reported that PIM1 can phosphorylate and stabilize NKX-3.1[39]. PIM1 may also directly phosphorylate downstream proteins, such as MYC, to promote the hepatic

conversion process. Overall, TF-activated PIM1 functioned together with FHH and 24 additional TFs to trigger hepatic conversion by inducing hepatic-lineage genes, promoting cell cycle progression, and suppressing ferroptosis (Fig. 7c).

## Discussion

PKs catalyze protein phosphorylation to regulate numerous biological processes. Due to the complex regulation mechanisms driven by PKs, it remains a great challenge to identify functionally important PKs that regulate a defined biological process in a high-throughput way or at the whole genome level.

To resolve this problem, we developed CKI to integrate quantitative transcriptomic and phosphoproteomic data for prediction of potentially central PKs. The algorithm was tested by predicting PKs regulating DOX resistance, genistein resistance, mouse hepatocyte maturation, and FHH-mediated direct hepatic reprogramming. Using PKs known to be associated with DOX or genistein resistance as independent testing sets, our results demonstrated that CKI had a higher accuracy than other methods that did not integrate different types of data (Supplementary Fig. 1d, g, k, n). We also validated a total of five predicted central PKs involved in regulating hepatocyte maturation and hepatic reprogramming (Figs. 2k and 3c, e), indicating the reliability of CKI.

In processing the phosphoproteomic data, the directionality of phosphorylation events (activation vs. inhibition) was not considered, because this information was not available for most of the p-sites. Previously, it was reported that the activity of a PK could be monitored by measuring its activity-associated p-sites[40]. For example, the phosphorylation level of PLK1 at T210 is positively correlated with its PK activity[40]. However, due to the relatively low coverage of phosphoproteomic profiling, we only detected two PK activity-associated p-sites that were up-regulated after DOX treatment (ATM S1981[41]) or in CLiP-Hep cells (Cilk1 T157[42]) (Supplementary Fig. 2b, d, Supplementary Fig. 3l, and Supplementary Fig. 5r). We did not find any PK activity-associated p-sites to be differentially regulated in genistein resistance or hepatic reprogramming. Using all pairwise comparisons of the transcriptomic and phosphoproteomic data for each biological process, we analyzed correlations between PK gene expression and the intensity of the corresponding substrate p-sites based on the predicted ssKSRs. The average Spearman's correlation coefficients were calculated as 0, −0.02, 0.14, and 0.10 for DOX resistance, genistein resistance, hepatocyte maturation, and hepatic reprogramming, respectively (Supplementary Fig. 11a–d), implying a weak correlation between PK gene expression and phosphorylation of the target p-sites.

In CKI, the p-site intensity was directly used, without considering the potential impact of protein expression. To evaluate the influence of protein expression on the CKI accuracy, we additionally conducted proteomic quantifications for mouse hepatocyte maturation and human hepatic reprogramming. Then, the CKI predictions were compared, using p-sites with or without normalization by their corresponding protein expression level (Supplementary Note 3, Supplementary Figs. 12, 13). From the results, we found that only one validated PK, PIM1, could still be predicted if normalized by proteomic data (Supplementary Figs. 12f, 13f), indicating proteomic normalization didn't increase the CKI accuracy for our study.

Molecular barriers that antagonize cell lineage reprogramming have been reported in hepatic reprogramming and in many other reprogramming scenarios[25,26,43]. In this study, we also showed robust cell death and cell cycle arrest of HDFs after enforced expression of FHH (Figs. 5e, 6a, b). Specifically, our experiments showed that PIM1 facilitated hepatic conversion by promoting cell cycle progression and suppression of ferroptosis. Knockdown of *PIM1* by shRNAs further demonstrated that *PIM1* was required for efficient hepatic conversion. PIM1 is a serine/threonine PK that is involved in cell cycle progression, cell survival, glucose metabolism, and several other signal transduction pathways[34,44]. We here found that PIM1 suppressed ferroptosis of HDFs undergoing hepatic reprogramming. Ferroptosis is a form of regulated necrosis[31]. A previous study showed that ferroptosis was induced during direct neuronal conversion[43]. Thus, induction of ferroptosis is not restricted to hepatic reprogramming, but also occurs in other cell lineage reprogramming processes.

In this study, 17 pathways were enriched upon PIM1 over-expressing during hepatic reprogramming (Fig. 5c). We only validated two of them, including "cell cycle" and "ferroptosis". Thus, there could be other downstream signaling pathways for PIM1 during hepatic reprogramming. Overall, PIM1 functions as an important cell lineage reprogramming signaling molecule to antagonize cell fate change-induced molecular barriers. The competition between reprogramming signals and molecular barriers eventually determines the final decision of a cell's fate.

There are some limitations of this study that should be noted. First, we only considered transcriptomic and phosphoproteomic data for prediction of potentially central PKs. However, protein expression levels are also important for regulating PK activity[9]. Indeed, we found that one validated PK, *Prkx*, was significantly up-regulated in CLiP-Hep against MH cells, at the proteomic expression level (Supplementary Fig. 12e, $p < 0.01$), supporting the usefulness of proteomic data. Furthermore, epigenetic mechanisms[45] play a critical role in regulating

gene expression in many biological processes, especially cell lineage reprogramming. Thus, it could reasonably be expected that incorporation of quantitative proteomic data, DNA methylomic data, and genome-wide histone modification data would improve model accuracy. Second, the directionality of phosphorylation was not considered due to limitations in the available data. Including perturbation or prior functional data to distinguish between activating and inhibiting phosphorylation events is also expected to improve prediction performance. Finally, we tested the accuracy of CKI in only four biological processes. More processes will be considered in the future to further evaluate the reliability and accuracy of CKI.

## Methods

### Cell lines and cell culture condition

293FT cells (Invitrogen, R70007) were maintained in DMEM high-glucose media (Gibco) supplemented with 10% FBS (Ausbian). U-2 OS cells (ATCC, HTB-96) were maintained in McCoy's 5 A (Modified) Medium (Gibco) supplemented with 10% FBS (Ausbian). Human adult fibroblasts were derived from skin biospies of two 35-year-old healthy male individuals with approval for collection and use of human samples by institutional ethical committees of Eye & ENT Hospital, Fudan University (2020035, 2021007-1). Human skin biopsy tissues were cut into about 1 mm × 1 mm × 1 mm pieces, and placed on 60 mm dishes (Corning) precoated with 15 µg/cm² collagen type I from rat tail (Sigma-Aldrich) in 6 ml DMEM high-glucose media (Gibco) supplemented with 10% FBS (Ausbian) and 5 ng/ml bFGF (PeproTech), and put in the 37 °C incubator. After 3 weeks' incubation, fibroblasts that migrated out of the tissues were passaged to new collagen-I-coated dishes.

### Isolation of mouse hepatocytes

All animal experiment procedures were conducted in compliance with the approval of the Animal Ethics Committee at ShanghaiTech University (20200713002). Mouse hepatocytes were isolated from 8-week-old male C57BL/6 J mouse (Charles River Laboratories) using the procedure of standard two-steps collagenase perfusion method[46]. In brief, the liver was pre-perfused through the portal vein with perfusion solution (1 × EBSS without Ca²⁺ and Mg²⁺(Sangon Biotech) supplemented with 0.5 mM EGTA (Sigma-Aldrich)) for 3–5 min and then perfused with collagenase solution (0.2 mg/ml collagenase type IV (Sigma-Aldrich), 10 mM HEPES (Solarbio), 1× EBSS with Ca²⁺ and Mg²⁺ (Sangon Biotech)) at 2–3 ml/min. The extracted liver was mechanically digested with sterile scissors and then filtered through a 70 µm filter membrane. The cells were collected via centrifugation at $60\,g$ for 2 min. Then the cells were resuspended using 40% Percoll (Cytiva) diluted with DMEM supplemented with 10% FBS and centrifuged at 1000 rpm for 5 min to remove dead and non-hepatic cells. Purified hepatocytes were then counted and used for further experiments.

### Generation of liver progenitor cells

The generation of liver progenitor cells were basically according to previously published protocol[17]. Concisely, isolated primary mouse hepatocytes were seeded on plates coated with collagen type I (Sigma) at $2 \times 10^4$ cells/cm². The cells were cultured in DMEM supplemented with 10% FBS for 6 h and then the medium changed to SHM medium (DMEM/F12 containing 2.4 g/L NaHCO₃ and L-glutamine (Gibco) supplemented with 5 mM HEPES (Solarbio), 30 mg/L L-proline (Alfa Aesar), 0.05% BSA (Solarbio), 10 ng/ml epidermal growth factor (PeproTech), insulin-transferrin-serine (ITS) (Sigma-Aldrich), $10^{-7}$ M dexamethasone (Dex) (Sigma-Aldrich), 10 mM nicotinamide (Solarbio), 1 mM ascorbic acid-2 phosphate (Wako) and 1× antibiotic/antimycotic solution (Solarbio)) supplemented with YAC (10 mM Y-27632 (Medchemexpress), 0.5 mM A-83-01 (Medchemexpress), 3 mM CHIR99021 (Medchemexpress)). Cells were cultured for 14 days to generate liver progenitor cells and the medium was changed every other day thereafter.

## Differentiation of liver progenitor cells

For induction of hepatocytes, liver progenitor cells were collected and seeded onto collagen type I-coated plates at $2 \times 10^4$ cells/cm². The cells were cultured for 2 days in SHM medium and the medium was supplemented with 1 μM dexamethasone (Dex) (Sigma) and 20 ng/ml human Oncostatin M (PeproTech) for 6 days. For induction of biliary epithelial cells, the liver progenitor cells were isolated and co-cultured with pre-inoculated low density mouse embryonic fibroblasts (passage 2). The mouse embryonic fibroblasts were isolated from E13.5 C57BL/6 J mouse embryos and cultured in DMEM high-glucose media (Gibco) supplemented with 10% FBS (Ausbian). The medium was replaced with mTeSRTM1 (STEMCELL Technologies) containing YAC (mTeSR1+YAC), and on day 6, the medium was replaced with mTeSR1+YAC supplemented with 2% Matrigel (Corning) for another 6 days. All media were changed every other day.

## hiHep cell induction

hiHep cells were generated as previously described[22]. Briefly, 2×10⁵ human dermal fibroblasts between passage 5 and 10 were seeded on a collagen-I-coated 60 mm dish and infected with pWPI lentiviruses expressing *FOXA3*, *HNF1A* and *HNF4A* (multiplicity of infection: 1.5 for each virus). For lentivirus production, pWPI lentiviral plasmids encoding *FOXA3*, *HNF1A* or *HNF4A* were introduced into 293FT cells (Thermo) together with psPAXs (Addgene) and pMD2.G (Addgene). The medium containing lentiviruses was collected and passed through 0.45 μm filter (Sangon Biotech) after 48 hours incubation. After 48 hours' infection of human dermal fibroblasts, the medium was changed to hepatocyte maintenance medium (DMEM/F12 (Gibco) supplemented with 0.544 mg/L ZnCl₂ (Sinopharm), 0.75 mg/L ZnSO₄·7H₂O (Sinopharm), 0.2 mg/L CuSO₄·5H₂O (Sinopharm), 0.025 mg/L MnSO₄ (Sinopharm), 2 g/L Bovine serum albumin (Sigma-Aldrich), 2 g/L Galactose (Sigma-Aldrich), 0.1 g/L Ornithine (Alfa Aesar), 0.03 g/L Proline (Alfa Aesar), 0.61 g/L Nicotinamide (Solarbio), 1X Insulin-transferrin-sodium selenite media supplement (Sigma-Aldrich), 40 ng/ml TGFα (PeproTech), 40 ng/ml EGF (PeproTech), and 10 μM dexamethasone (Sigma-Aldrich)). Cells were collected for further analyses at different time points according to specific experiment.

## Molecular cloning and lentivirus package

For gene overexpression, *PIM1* and other candidate targets were PCR amplified from cDNA of human fibroblasts and cloned into the XbaI (NEB) and BamHI (NEB) restriction enzyme sites of the pCDH-CMV-MCS-EF1-copGFP vector (System Biosciences) using Clone Express® II One Step Cloning Kit (Vazyme). PCR was performed with KOD-Plus-Neo (TOYOBO). Cloning primers are listed in Supplementary Data 7. TFs were previously constructed with modified pWPI plasmid (Addgene). For shRNA experiments, the non-targeting shNT and all other shRNA sequences were inserted into the AgeI and EcoRI restriction enzyme sites of the pLKO.1 plasmid (Addgene), respectively. DNA sequences were mainly obtained from Sigma MISSION shRNA library. The oligonucleotide sequences are provided in Supplementary Data 7. Constructed pCDH, pWPI or pLKO.1 plasmid was then introduced to 293FT cells together with packaging plasmid psPAX2 (Addgene) and envelope plasmid pMD2.G (Addgene). After 48 h incubation, the medium containing lentiviruses was collected by 0.45 mm filter and stored at −80 °C. The constructs used here are available upon request.

## Quantitative RT-PCR

Total RNA was isolated from cells by Trizol (Invitrogen) and a total of 1 μg RNA was reversely transcribed into cDNA with PrimeScript™ RT reagent Kit with gDNA Eraser (Perfect Real Time) (Takara) according to the manufacturer's instructions. Quantitative real-time PCR was performed with TB Green™ Premix Ex Taq™ II (Tli RNaseH Plus) (Takara)

on an ABI 7500 fast real-time PCR system. Primer sequences used for qRT-PCR are provided in Supplementary Data 7.

## Immunoblotting

First, cells were washed twice with PBS (Sangon Biotech), lysed in RIPA Lysis Buffer (Beyotime) supplemented with protease inhibitor cocktails and PhosStop (Roche). Protein was mixed with loading buffer, boiled and subjected to SDS-PAGE electrophoresis and transferred to a PVDF membrane (Millipore) according to a standard protocol. After blocking with 5% non-fat milk in TBST (50 mM Tris-HCl (Sigma-Aldrich) at pH 8.0, 150 mM NaCl (SinoPharm), 0.1% Tween 20 (Sigma-Aldrich)), the membrane was incubated with non-fat milk or BSA-TBST-diluted primary and secondary antibodies. Signals were detected with ECL detection reagent (Vazyme) using AI600 (GE). The antibodies used in this study were: anti-GAPDH (Proteintech, 10494-1-AP), anti-Pim-1 (D8D7Y) Rabbit mAb (Cell Signaling, 54523 S), anti-c-Myc antibody [Y69] (Abcam, ab32072), anti-c-Myc (phospho S62) antibody (Abcam, ab51156), anti-4E-BP1 (53H11) (Cell signaling, 9644 T), anti-p-4E-BP1(T32/46) (Cell signaling, 2855 T), Anti-rabbit IgG, HRP-linked Antibody (Beyotime, A0208).

## Immunofluorescence

Cells were washed twice with PBS and fixed with 4% paraformaldehyde (PFA) (Sigma-Aldrich) for 15 min at room temperature (RT) followed by permeabilization with 0.25% Triton X-100 (Sigma-Aldrich) in 3% BSA (Solarbio) for 30 min at RT. 5% BSA (Solarbio) was used for blocking followed by incubation of primary antibodies overnight at 4 °C. After being washed three times with PBST, cells were stained with Cy3-conjugated secondary antibodies. Images were visualized by Zeiss Z2 (Zeiss). The antibodies used in this study were: anti-E-cadherin (Invitrogen, 13-1900), anti-Human Albumin cross adsorbed (Bethyl, A80-229A), anti-Albumin (GeneTex, GTX102419), Alexa Fluor™ 488 Donkey anti-Rat IgG (H + L) (Invitrogen, A-21208), Alexa Fluor™ 568 Donkey anti-Rabbit IgG (H + L) (Invitrogen, A10042), anti-Human Albumin cross adsorbed (Bethy, A80-229A), Cy3-conjugated AffiniPure Donkey Anti-Rabbit IgG (H + L) (min X Bov,Ck,Gt,GP,Sy Hms,Hrs,Hu,Ms,-Rat,Shp Sr Prot) (Jackson,711-165-152).

## Flow cytometry

For cell death analysis, cells were collected at day 5 after FHH or GFP transfection. Dead cells were detected by Annexin V-PE/7-AAD Apoptosis Detection Kit (Yeasen) according to the manufacturer's instructions and analyzed by LSRFortessa (BD). Data were analyzed with Flowjo.

For intracellular staining of ALBUMIN, $1 \times 10^6$ cells were harvested and fixed with 4% PFA (Sigma-Aldrich) for 30 min, and then permeabilized in with 0.25% Triton X-100 (Sigma-Aldrich) in 3% BSA for 10 min. The cells were then incubated with 3% BSA 1 hour at room temperature followed by incubating with Human Serum Albumin APC-conjugated Antibody (R&D) for 30 min in 3% BSA. Then, the cells were washed three times with cold PBS and the results were analyzed by the LSRFortessa (BD). The antibodies used in this study were: Human Serum Albumin APC-conjugated Antibody (R&D, IC1455A).

## Cell proliferation analysis

Cells were incubated with 10 μM EdU culture medium (Reagent A, RIBOBIO, c10327-1, 1:5000 diluted in culture medium) for 24 h at 37 °C. Then, the cells were washed with PBS two times (5 min/wash) and fixed with 4% PFA for 30 min at room temperature, followed by incubation in 2 mg/ml glycine for 5 min on bleaching shaker. The cells were then washed 3 times in PBS (5 min/wash), and permeated cell membrane by 0.5%-Triton in PBS and incubate at room temperature for 10 min. Then, the cells were washed in PBS and incubated with 1× Apollo® dyeing reaction solution (Reagent B, C, D, E, RIBOBIO, c10327-1) for 30 minutes in the dark, room temperature, and bleaching shaker. After

discarding the dyeing reaction solution, Cells were incubated with 0.5%-Triton in PBS two times in dark (10 min/incubate). Then, the cells were incubated with 1× Hoechst 33342 (Reagent F, RIBOBIO, c10327-1, 1:100 diluted in distilled water) for 30 min in the dark, room temperature, and bleaching shaker. After washing cells in PBS three times (5 min/wash), the cells were visualized by Zeiss Z2 (Zeiss).

## NADP/NADPH analysis

The NADP/NADPH analysis were formed using a NADP/NADPH assay kit (Promega) following manufacturer's instruction. Cells were collected and resuspended in 1 mL cold PBS. Then, counted cells ($2 \times 10^6$ cells) were resuspended in 60 μL Lysis Buffer and incubated at room temperature for 15 minutes. Lysate was then centrifuged at 1500 rpm for 5 minutes and use supernatant for the assay. The 12.5 μL of NADPH/NADP Extraction Solution was added into the NADPH/NADP sample wells and the mix solution was incubated at room temperature for 10–15 minutes. Then, 12.5 μL NADP/NADPH Extraction Solution was added to neutralize NADPH/NADP extracts for 10–15 minutes. 37.5 μL NADP/NADPH reaction mixture was then added to the mix and incubated at RT for 45 min. Monitor fluorescence intensity (Ex/Em = 540/590 nm) by SpectraMax i3 (MD).

## Glutathione assay

The cells were washed once with PBS and collected by centrifugation, and the supernatant was aspirated. Then, 150 μL 5% 5-sulfosalicylic acid (SSA) solution was added to the cell pellet, fully Vortex. The sample was then subjected to three rapid freeze-thaw cycles using liquid nitrogen and a 37 °C water bath. Then the lysate was leave in ice for 5 minutes and then centrifuged at 10,000 g for 10 minutes at 4 °C. The supernatant was taken for determination of total glutathione. The reaction scheme was set up according to manufacturer table (Sigma-Aldrich, CS0260-1KT) and performed every test in five duplicate wells. The first 2 wells should contain only 10 μL of the 5% 5-Sulfosalicylic Acid Solution as a reagent blank. Duplicate 10 μL samples of the prepared Glutathione Standard Solutions were added into separate wells of the plate. 10 μL sample were added into 5 duplicate separate wells. Then, 150 μL of the Working Mixture was added to each well with pipette and mix them by pipetting up and down. The wells were incubated 5 minutes at room temperature and then add 50 μL of the diluted NADPH Solution with a multichannel pipette. The mixture was mixed by pipetting up and down and the absorbance in each well was measured by SpectraMax i3 (MD).

## RNA extraction and RNA-seq

Cell samples were ground with Trizol Reagent (Invitrogen) in liquid $N_2$, then incubated at room temperature. The homogenates were centrifuged at 12,000 g 4 °C and supernatant was thoroughly mixed with chloroform (Sinopharm). After centrifugation of 12,000 g, aqueous phase was transferred and mixed with isopropanol (Sinopharm). Finally, RNA sediment was precipitated two times with 75% ethanol and obtained the total RNA.

Before sequencing, the quantity of total RNA was measured by Nanodrop (Thermo, USA), and the quality was assessed with electrophoresis. For the construction of RNA library, TruSeq® RNA LT Sample Prep Kit v2 (Illumina, USA) was used to treat 2 μg RNA for each sample. Then, mRNA was purified from total RNA with RNA Purification Beads, and fragmented with Elute, Prime, Fragment Mix (EPF). First and second strand cDNAs were synthesized based on random primers. End repair was performed with End Repair Mix 2. After Adenylating 3' ends, adaptors were ligated to RNA fragments. And then, the cDNA was amplified with PCR and quantified with Qubit (Invitrogen, USA). For the cluster generation of index-coded samples, TruSeq PE Cluster Kit v3-cBot-HS (Illumina) was used on the cBot Cluster Generation System (Illumina). Library was sequenced on Illumina Hiseq 3000 platform.

## Protein extraction, isolation of peptides and TMT labeling

Cell sample was mixed with lysis buffer (8 M urea (Sinopharm), 1% protease inhibitor (ThermoFisher) and phosphatase inhibitor (ThermoFisher), and 2 mM EDTA (Sigma-Aldrich), at pH 8.0 with ice-bath) and homogenized by sonication at 0 °C for 3 min (cycle: sonication for 3 s in -180 W power, and stop for 5 s to cool down). After the centrifugation of lysate at 20,000 g at 4 °C for 10 min, the supernatant was transferred and collected. Finally, the protein concentration was measured with 2-D Quant kit (Cytiva) and adjusted to be consistent for all samples.

For the digestion of proteins, protein solution was first reduced with 5 mM dithiothreitol (DTT, pH 8.0) (Sigma-Aldrich) at 56 °C for 30 min, and then incubated with 11 mM iodoacetamide (pH 8.0) (Sigma-Aldrich) for 15 min at room temperature in the dark. To reduce the concentration of urea to less than 2 M, 100 mM triethylammonium bicarbonate (TEAB) (Sigma-Aldrich)was added to the protein sample. Two times trypsin digestions were processed, with the mass ratio of 1:50 trypsin-to-protein for 37 °C overnight treatment and 1:100 for 4 h, respectively. Finally, trifluoroacetic acid (TFA) (Sigma-Aldrich) were added to adjust pH to 2–3 for quenching the digestion.

Before the TMT labeling of peptides, peptides were vacuum-dried after the desalting with Strata X C18 SPE column (Phenomenex), and reconstituted with 0.5 M TEAB (Sigma-Aldrich). Following the manufacturer's protocol with minor modifications, the TMT kit (ThermoFisher Scientific) was constituted with acetonitrile (Fisher Chemical), and the equal peptides based on the absorbance at 280 nm were subsequently processed with TMT kit (pH 8.5), and incubated with labeling reagent (Batch 1: MH [126, 127, and 128] and CLiP-Hep [129, 130, and 131], and Batch 2: GFP [131], FHH-2.25d [129], and FHH-5d [130]) at room temperature for 2 h to ensure a better efficiency of labelling. Finally, the labeled peptides were equally pooled based on the absorbance at 280 nm, desalted, and dried by vacuum.

To measure the TMT labeling efficiency, msConvert[47] was used to convert the data format of raw MS/MS data into mascot generic format (MGF) files. The monoisotopic reporter masses of different TMT 6-plex labels were downloaded from MaxQuant (v.1.4.1.2)[20], including 126 (126.127726 Da), 127 (127.124761 Da), 128 (128.134436 Da), 129 (129.131471 Da), 130 (130.141145 Da), and 131 (131.138180 Da). Using 0.005 Da as the reporter mass tolerance, for each batch, the total number of MS/MS spectra in MGF files was defined as $T$, while the number of MS/MS spectra labeled with at least one detected TMT 6-plex reagent was defined as $L$. Then, the TMT labeling efficiency $E$ of each batch was calculated as below:

$$E = \frac{L}{T} \tag{1}$$

By this approach, we obtained 96.17% and 98.68% efficiencies for mouse hepatocyte maturation and human hepatic reprogramming samples, respectively.

During revision, we additionally prepared two batches of samples for proteomic profiling, without further phosphopeptide enrichment. The efficiencies of TMT labelling were 94.88% and 90.69% for additional samples of mouse hepatocyte maturation (Batch 1: MH [126, 127, 128] and CLiP-Hep [129, 130, 131]) and human hepatic reprogramming (Batch 2: GFP [126], FHH-2.25d [127], and FHH-5d [128]), respectively.

## Phosphopeptide enrichment

The tryptic peptides were fractionated into fractions with the high pH reverse-phase high-performance liquid chromatography (HPLC), based on the Thermo Betasil C18 column. The peptides were separated into 60 fractions with a gradient of 8% to 32% acetonitrile (pH 9.0) (Fisher Chemical) over 60 min and then pooled into 4 and 11 fractions by combining with equal time interval for mouse hepatocyte

maturation and human hepatic reprogramming, followed by the vacuum drying.

For the phosphopeptide enrichment, the loading buffer (50% acetonitrile & 6% TFA (Sigma-Aldrich)) with Ti[4+]-immobilized metal ion affinity chromatography (Ti[4+]-IMAC) microspheres was used to incubate the peptide mixtures for 1 h at room temperature (30 rpm). Next, the IMAC microspheres with enriched phosphopeptides were collected after centrifugation. Then, 50% acetonitrile with 6% TFA and 30% acetonitrile with 0.1% TFA were used to wash the microsphere for 10 min at room temperature (30 rpm) sequentially, to remove the nonspecifically adsorbed peptides. From the elution of phosphopeptides with microspheres by adding elution buffer containing 10% NH$_4$OH for 10 min at room temperature (30 rpm), the supernatant containing phosphopeptides was collected and lyophilized for further analysis.

## LC-MS/MS analysis

The liquid phase A (0.1% formic acid (Fluka)) was used to dissolve the enriched phosphopeptides and load onto a home-made reversed-phase analytical column (length: 15 cm, i.d.: 75 μm), and separated by EASY-nLC 1000 ultra-performance liquid chromatography (UPLC) system. Liquid phase B contains 0.1% formic acid in 90% acetonitrile (Fisher Chemical). The flow rate was maintained at 300 nL/min and the liquid phase gradient setting was as follows: 0–35 min, 4–16% B; 35–65 min, 16–24% B; 65–80 min, 24–40% B; 80–82 min, 40–80% B; 82–90 min, 80% B.

Peptides were subjected to NSI ion source (electrospray voltage: 2.0 kV) to ionize followed by MS/MS in Q Exactive™ Plus (Thermo), and Orbitrap was used for detection and analysis. The scan range of primary MS was 350 to 1800 m/z at a resolution of 70,000. Then, peptides were selected for MS/MS using NCE setting as 30 and detected with scan range starting at 100 m/z at a resolution of 17,500. A data-dependent procedure that alternated between one MS scan followed by 20 MS/MS scans was applied with 15.0 s dynamic exclusion. Automatic gain control (AGC) was set at 1E5.

## Re-analysis of the trans-omic data in drug-resistant cancer cells

For the transcriptomes of DOX resistance, we downloaded two read count files of U-2 OS cells with or without DOX treatment, from Gene Expression Omnibus (GEO)[48], a public resource for maintaining gene expression data (https://www.ncbi.nlm.nih.gov/geo/, accession number: "GSE84863")[11]. Then, the DEMs regulated by DOX against control were calculated with edgeR (version 3.28.1, $p < 0.01$), a frequently used R package to analyze RNA-seq data[49]. Also, we downloaded the transcriptomic data of genistein resistance from GEO (accession number: "GSE56066")[13], which contained four read count files of MCF-7 breast adenocarcinoma cells with or without genistein treatment for 24 h. Then, the DEMs of genistein vs. control were directly computed with edgeR ($p < 0.01$)[49].

For the phosphoproteomic data, we first obtained the searched results of TMT-based quantitative phosphoproteomes of DMSO- or DOX-treated U-2 OS cells with three technical replicates from PRIDE[50], a comprehensive resource for maintaining the proteomic data (https://www.ebi.ac.uk/pride/, accession number: "PXD007145")[12]. In addition, the search results of TMT-based phosphoproteomes with two biological replicates for MDA-MB-231 triple-negative breast cancer cells with or without genistein treatment for 24 h were also downloaded from PRIDE (accession number: "PXD002735")[14].

## Standard database search

For raw MS/MS data of phosphoproteomes, MaxQuant (v.1.4.1.2)[20] was used for standard database search. The MS/MS spectra files of three biological replicates of MH and CLiP-Hep samples were searched against the mouse proteome set downloaded from UniProt (Version 202002)[51], which contained 21,982 unique protein sequences in *Mus*

*musculus*. The MS/MS spectra files of GFP, FHH-2.25d, and FHH-5d samples were searched against the human proteome set obtained from UniProt (Version 201401)[51], which contained 20,274 unique protein sequences in *Homo sapiens*. Trypsin/P was chosen as the cleavage enzyme allowing up to 2 missing cleavages. The fixed modification was set as Carbamidomethyl (C), which Oxidation (M), Acetyl (Protein N-term), and Phospho (STY) were the variable modifications. The minimum peptide length was set as 7, and the mass tolerance for fragment ions was set as 0.02 Da. The false discovery rates (FDRs) of the peptide-spectrum match (PSM) and protein decoy fractions were all set to < 1%, and the minimum score for modified peptides was set to >40.

For raw MS/MS data of mouse or human proteomes, the same reference proteome sets mentioned above were adopted for spectral library searching with MaxQuant (v.1.4.1.2)[20]. Similarly, we chosen Trypsin/P as the cleavage enzyme and 2 as the maximum missing cleavages. The carbamidomethyl (C) was selected as fixed modification and Oxidation (M) and Acetyl (Protein N-term) were taken as the variable modifications. The minimum peptide length and mass tolerance were set as 7 and 0.02 Da, respectively. FDRs of PSM and protein decoy fractions were set to < 1%.

From the MaxQuant results, the raw reporter intensities were taken as the quantification values of p-sites or proteins.

## Detection of DEMs

We analyzed the transcriptomes of mouse hepatocyte maturation, by mapping raw reads to the mouse reference genome, which was downloaded from Ensembl (release version 99, http://www.ensembl.org/)[52]. Six BAM files of two samples each with three biological replicates, were individually produced by two software packages of Bowtie 2 (version 2.2.4)[53] and TopHat (version 2.2.1)[54]. Then, we used Cufflinks (version 2.2.1)[55] to assemble reads and calculate FPKM values of mapped mRNAs for the estimation of their expression levels. The Cuffdiff program in Cufflinks was adopted to detect DEMs for each replicate of CLiP-Hep vs. MH ($p < 0.01$).

For the transcriptomes of GFP, FHH-2.25d, and FHH-5d, we mapped raw reads to the human reference genome, which was downloaded from Ensembl (release version 85, http://www.ensembl.org/)[52]. Three BAM files were processed as described above, and DEMs were detected for the three time points ($p < 0.01$), respectively. The same procedure was also used for the detection of DEMs of HDFs undergoing hepatic transdifferentiation in the context of PIM1 overexpression against GFP transfection.

## Public phosphorylation data resources

We collected experimentally characterized human and mouse p-sites from 9 public phosphorylation databases, including UniProt[51], SysPTM 2.0[56], PhosphoSitePlus[57], PhosphoPep 2.0[58], PHOSIDA[59], HPRD 9[60], Phospho.ELM 9.0[61], dbPTM 3.0[62], and dbPAF[63]. As previously described[63], ambiguous p-sites annotated with "By similarity", "Potential" or "Probable" were excluded from the UniProt database. After the redundancy clearance, we obtained 244,034 known p-sites including 144,116 pS (59.06%), 61,231 pT (25.09%), and 38,687 pY (15.85%) resides in 18,773 human phosphoproteins, and 119,328 p-sites including 85,774 pS (71.88%), 23,594 pT (19.77%), and 9960 pY (8.35%) in 14,044 mouse phosphoproteins.

## Functional enrichment analysis

We performed hypergeometric tests for GO- and KEGG-based enrichment analyses of DEMs and differentially regulated p-sites (DRPs, ≥ 2-fold or ≤0.5-fold change), respectively. First, we downloaded GO annotations from the QuickGO (https://www.ebi.ac.uk/QuickGO/, on 21 October 2017)[19], which contained 19,476 human and 21,552 mouse proteins annotated with at least one GO biological process term. We also downloaded KEGG annotations from the ftp server of KEGG (ftp://ftp.bioinformatics.jp/, released on 15 October 2017)[18], containing

10,042 human and 11,521 mouse proteins annotated with at least one KEGG pathway. To identify GO biological processes that were significantly over- or under-represented in DEMs, here we defined:

$N$ = number of mapped human genes annotated with at least one GO term

$n$ = number of DEMs annotated with at least one GO term

$M$ = number of mapped human genes annotated with the GO term t

$m$ = number of DEMs annotated with the GO term t

Then, the enrichment ratio (E-ratio) was computed, and the $p$ value was calculated with the hypergeometric distribution as below:

$$E - ratio = \frac{\frac{m}{M}}{\frac{n}{N}} \tag{2}$$

$$p = \sum_{m'=m}^{n} \frac{\binom{M}{m'}\binom{N-M}{n-m'}}{\binom{N}{n}} (E - ratio \geq 1), \tag{3}$$

or

$$p = \sum_{m'=0}^{m} \frac{\binom{M}{m'}\binom{N-M}{n-m'}}{\binom{N}{n}} (E - ratio < 1) \tag{4}$$

The same hypergeometric test was also conducted for the enrichment of DRPs. GO-based enrichment analyses were performed for DEMs and DRPs of FHH-2.25d vs. GFP, FHH-5d vs. GFP, and FHH-5d vs. FHH-2.25d during hepatic reprogramming. GO- and KEGG-based enrichment analyses were also conducted for DEMs of CLiP-Hep against MH, as well as DEMs of HDFs undergoing hepatic transdifferentiation in the context of PIM1 overexpression against GFP transfection.

## The CKI algorithm

First, we obtained standard gene names and protein sequences of 524 human and 539 mouse PK genes from a previously developed database of iEKPD 2.0 (http://iekpd.biocuckoo.org/), which contained 109,912 known and predicted PKs in 164 eukaryotes[64]. For the human or mouse transcriptomic data, we identified potential DEPKs by directly mapping DEMs of a pairwise comparison to PKs using gene names. The $p < 0.01$ was selected as the threshold for DOX resistance, mouse hepatocyte maturation, and hepatic reprogramming, while a more stringent threshold of $p < 0.0001$ was selected for genistein resistance.

For identified p-sites in this study, their upstream regulatory PKs were first predicted by iGPS (http://igps.biocuckoo.org/)[10]. All p-sites were prepared in the tab-delimited Phospho.ELM (ELM for short) format[61], including UniProt accession numbers[51], full protein sequences, phosphorylation positions, and residue types. For each sample, the "Batch Predictor" option in iGPS was used, and the ELM file containing all p-sites was directly loaded for a prediction. The probability of a p-site modified by each PK was individually scored, and only results with scores higher than the pre-determined cut-off values were reserved and outputted. For using iGPS, we selected the default parameters of the "Low threshold" and "Experiment/STRING PPI".

Then, we designed two approaches to identify potentially central PKs from the phosphoproteomic data, including an intensity-based method and a network-based method. For the former, we first defined the total substrate intensity (*TSI*) of a PK by adding square root values

(*SRV*s) of TMT intensities of p-sites in their corresponding substrates. For each phosphoproteomic data set, the *TSI* value of a PK $i$ in the sample $A$ was calculated as below:

$$TSI_A(i) = \sum_{j=1}^{n} SRV_j \tag{5}$$

here, $n$ is the number of substrate p-sites of the PK $i$. Thus, the *TSI* score of $m$ PKs in the sample $A$ was computed as below:

$$TSI_A = \sum_{i=1}^{m} TSI_A(i) \tag{6}$$

We performed the Yate's chi-squared test of a pairwise comparison, by calculating a Chi-squared 2 × 2 contingency table:

| | Sample $B$ | Sample $B$ | Total |
|---|---|---|---|
| PK $i$ | $a = TSI_A(i)$ | $b = TSI_B(i)$ | $T_i = a + b$ |
| Other PKs | $c = TSI_A - TSI_A(i)$ | $d = TSI_B - TSI_B(i)$ | $T_o = c + d$ |
| | $T_A = a + c$ | $T_B = b + d$ | $T = a + b + c + d$ |

The $\chi^2$ was determined as below:

$$\chi^2 = \frac{T\left[\max\left(0, |ad - bc| - \frac{T}{2}\right)\right]^2}{T_i T_o T_A T_B} \tag{7}$$

The function chisqrprob(degree, $\chi^2$) of Perl module Statistics::Distributions was adopted to calculate $p$ values, whereas the degree was set as 1. In this method, the $p < 1.0 \times 10^{-5}$ was adopted as the threshold to predict potentially central PKs for DOX resistance, mouse hepatocyte maturation, and hepatic reprogramming, while a relaxed threshold of $p < 0.05$ was selected for genistein resistance.

In the network-based method, we defined the kinase network index (*KNI*) to denote the network state of a PK. First, we reconstructed a site-specific PK-substrate network from predicted ssKSRs mutually quantified in a pair of samples, e.g., treatment vs. control. In the network, the nodes denoted individual p-sites or their upstream regulatory PKs, and the orientations of edges were defined as PKs -> p-sites. Then, single PK networks were individually retrieved to only contain the PK and its substrate p-sites.

To compare sample $A$ and $B$, the weight value of the network edge, the relative intensity ratio (*RIR*) of a p-site in $A$ vs. $B$ was calculated as below:

$$RIR_{p-site} = \frac{TMT\ intensity_A}{TMT\ intensity_B} \tag{8}$$

Based on *RIR* values of p-sites, we split each single PK network into an up-regulated sub-network (*RIR* > 1) and a down-regulated sub-network (*RIR* < 1). For a PK $i$, its up-regulated $KNI_U(i)$ value could be computed by adding *RIR*s of $m$ p-sites in the up-regulated sub-network, whereas its down-regulated $KNI_D(i)$ score could be inferred from *RIR*s of $n$ p-sites in the down-regulated sub-network as below:

$$KNI_U(i) = \sum_{j=1}^{m} RIR_j \tag{9}$$

$$KNI_D(i) = \sum_{k=1}^{n} 1/RIR_k \tag{10}$$

Then, the total $KNI_U$ and $KNI_D$ of $A$ vs. $B$ for $l$ PKs were calculated as below:

$$KNI_U = \sum_{i=1}^{l} KNI_U(i) \qquad (11)$$

$$KNI_D = \sum_{i=1}^{l} KNI_D(i) \qquad (12)$$

Again, the Yate's chi-squared test was used to identify potentially up- or down-regulated PKs with the following Chi-square $2 \times 2$ contingency table:

|  | Up-regulated | Down-regulated | Total |
|---|---|---|---|
| PK $i$ | $a = KNI_U(i)$ | $b = KNI_D(i)$ | $T_i = a + b$ |
| Other PKs | $c = KNI_U - KNI_U(i)$ | $d = KNI_D - KNI_D(i)$ | $T_o = c + d$ |
|  | $T_U = a + c$ | $T_D = b + d$ | $T = a + b + c + d$ |

The $\chi^2$ was computed as below:

$$\chi^2 = \frac{T\left[\max\left(0, |ad - bc| - \frac{T}{2}\right)\right]^2}{T_i T_o T_U T_D} \qquad (13)$$

The $p$ values were also computed with the function chisqrprob(degree, $\chi^2$) of perl module Statistics::Distributions, and the degree was set as 1. In this method, the $p < 1.0 \times 10^{-5}$ was adopted as the threshold to predict potentially central PKs for DOX resistance and mouse hepatocyte maturation, while a relaxed threshold of $p < 0.01$ was selected for genistein resistance and hepatic programming.

Finally, the number of positive hits was counted for each PK from all pairwise comparisons. For DOX resistance[11,12], the threshold in CKI for prioritization of potentially central PKs were set as ≥14 of the 19 pairwise comparisons (1 for mRNA expression, $3 \times 3$ for substrate p-site intensity, and $3 \times 3$ for kinase-substrate network). For genistein resistance[13,14], the threshold was ≥ 6 of the 12 pairwise comparisons ($2 \times 2$ for mRNA expression, $2 \times 2$ for substrate p-site intensity, and $2 \times 2$ for kinase-substrate network). For hepatocyte maturation, the threshold was ≥ 15 of the 27 pairwise comparisons ($3 \times 3$ for mRNA expression, $3 \times 3$ for substrate p-site intensity, and $3 \times 3$ for kinase-substrate network). For hepatic programming, the threshold was ≥ 5 of the 9 pairwise comparisons (3 for mRNA expression, 3 for substrate p-site intensity, and 3 for kinase-substrate network).

### KSEA analysis
For the human phosphoproteomic data, the R package KSEAapp[16] (version 0.99.0) was directly used to identify potentially central PKs ($p < 0.05$). For the mouse phosphoproteomic data, we obtained known ssKSRs by mapping the p-sites to PhosphoSitePlus[57] as previously described[15]. Then KSEAapp was used for predictions of potentially important PKs using mapped ssKSRs ($p < 0.05$).

### Performance evaluation
For the DOX or genistein resistance, known regulatory PKs curated from the literature were taken as the positive data, while other remaining PKs predicted with at least one ssKSR were taken as the negative data. To evaluate the accuracy of CKI and other methods, true positive ($TP$), true negative ($TN$), false positive ($FP$), and false negative ($FN$) values were counted, and we calculated 2 commonly-used measurements, including sensitivity ($Sn$) and specificity ($Sp$) as below:

$$Sn = \frac{TP}{TP + FN}, \qquad (14)$$

$$Sp = \frac{TN}{TN + FP} \qquad (15)$$

The ROC curves were illustrated based on $Sn$ and $1\text{-}Sp$ values, and the corresponding AUC values were calculated. A higher AUC value denotes a higher accuracy of a predictive model in general.

Then, the confusion matrix under a selected threshold of pairwise comparisons was calculated as below:

|  |  | Actual | |
|---|---|---|---|
|  |  | Known | Unknown |
| Predicted | Known | $\frac{TP}{TP+FP}$ | $\frac{FP}{TP+FP}$ |
|  | Unknown | $\frac{FN}{FN+TN}$ | $\frac{TN}{FN+TN}$ |

For each of known or unknown PKs, the number of positive hits and the sum of all minus-log transformed (-lg) $p$ values were counted for PCA analysis.

### Correlation analysis of mRNA expression levels of PKs and their substrate p-site intensities
The correlations between mRNA expression levels of PKs and their corresponding substrate p-site intensities were measured based on the ssKSRs, using the Spearman's correlation. For each ssKSR, the Spearman's correlation coefficient ($\rho$) between the mRNA expression level of the upstream PK and the intensity of the downstream p-site was calculated across different samples or replicates.

### Correlation and clustering of samples
The correlations of biological replicates and samples were measured with the Spearman's correlation, based on transcriptomic or phosphoproteomic data. The Spearman's correlation coefficient was calculated and used for 2-way hierarchical clustering.

### Trans-omic modeling of the hepatic transdifferentiation-associated TPCW
First, we manually collected 41 experimentally identified hepatic lineage genes in *H. sapiens* from the scientific literature. Besides annotated human genes in the KEGG pathway of ferroptosis (KEGG ID: hsa04216), we further curated 37 known ferroptosis-associated genes from PubMed. For cell cycle, all human genes in the KEGG pathway of cell cycle (KEGG ID: hsa04110) were directly used. All curated genes of hepatic lineage, ferroptosis, and cell cycle were shown in Supplementary Data 6. Before the construction of TPCW, we reserved the curated genes transcriptionally up-regulated by FHH with or without PIM1 overexpression ($p < 0.01$). Also, we reserved the curated genes, of which protein products contained at least one DRP. Moreover, we validated an additionally hepatic lineage gene CYP3A4 regulated by FHH in the context of PIM1, and this gene was also included.

In the transcriptional level, we first downloaded pre-calculated TF-binding sites (TFBSs) of all available TFs from 'Genome Tracks' of JASPAR database (http://expdata.cmmt.ubc.ca/JASPAR/downloads/UCSC_tracks/2018/hg38/)[65]. The locus of human genes were extracted from the Gene Annotation Format (GTF) file of Ensembl[52]. Highly potential TFBSs ($p < 0.001$) located within the upstream 2000 bp and downstream 500 bp of each gene locus were reserved to determine potential TF-target relations. From the transcriptomic data, we obtained 24 potentially FHH-regulated TFs transcriptionally up-regulated ($p < 0.01$) in the pairwise comparison of FHH-2.25d vs. GFP, FHH-5d vs. GFP, or FHH-5d vs. FHH-2.25d (Supplementary Data 6). Then, predicted TF-target relations among FHH, the 24 TFs and other target genes were reserved. Also, we downloaded known TF-target relations in *H. sapiens* from the database of TRRUST (v2, http://www.grnpedia.org/trrust/), which contained 8444 and 6552 TF-target relations for 800 human TFs

and 828 mouse TFs curated from the literature, respectively[66]. Standard gene names were used to map the TRRUST data to our data set. To avoid missing any known data, we also searched the PubMed with all pairs of FHH and 24 FHH-regulated TFs with other genes, and obtained additionally known TF-target relations. We merged the three types of data sets, and in total 48 known and 1228 predicted TF-target relations between 27 TFs and 89 target genes were integrated (Supplementary Data 6).

In the phosphorylation level, the experimentally characterized ssKSRs between the 2 central PKs and other proteins were directly taken from two public databases including PhosphoSitePlus[57] and UniProt[51], as well as an additional literature curation. Also, iGPS was used to predict all potential ssKSRs between the 2 central PKs and other proteins, with the default parameters of the "Low threshold" and "Experiment/STRING PPI". In total, 12 known and 459 predicted ssKSRs were obtained between 2 central PKs and 30 other proteins (Supplementary Data 6).

Finally, we merged the TF-target network and the kinase-substrate network together to obtain an integrative TPCW, which was illustrated by Cytoscape, a tool for visualizing complex networks[67]. In the TPCW, there were 27 TFs including FHH and 24 additional FHH-regulated TFs, 2 potentially central PKs, and 60 curated genes of hepatic lineage, ferroptosis, and cell cycle were adopted for TPCW modeling. A hepatic lineage gene, *TF* (Serotransferrin), is also associated with ferroptosis (Supplementary Data 6). In the TF-target network, the nodes denoted TFs or their downstream targets, and the orientations of the edges were defined as TF -> target. In the kinase-substrate network, the nodes denoted PKs or their downstream substrate proteins, and the orientations of the edges were defined as PK -> substrate protein. In the final TPCW, the edge weight was not considered. All known and predicted TF-target relations and ssKSRs were shown in Supplementary Data 6.

### Statistics and reproducibility

All experiments are performed at least in three biological replicates. No statistical method is used to predetermine sample size and no data are excluded from the analyses. All statistical data are presented as the mean + standard deviation (SD). Statistical significance of the difference is determined using Student's *t* test. Differences are considered significant at the $p < 0.05$.

### Reporting summary

Further information on research design is available in the Nature Research Reporting Summary linked to this article.

## Data availability

The RNA-seq data has been deposited into NCBI GEO[48] (https://www.ncbi.nlm.nih.gov/geo/) with the dataset identifier "GSE169702". The mass spectrometry phosphoproteomic and proteomic data including the annotated mass spectra have been deposited into the integrated proteome resources (iProX, http://www.iprox.org/)[68] with the dataset identifier "PXD035829 [https://www.iprox.cn/page/project.html?id=IPX0001681000]". All the other data supporting the findings of this study are available within the article and its supplementary information files, or from the corresponding authors upon reasonable request. Source data are provided with this paper.

## Code availability

The source code of CKI has been uploaded to GitHub (https://github.com/BioCUCKOO/CKI) with the DOI identifier (https://doi.org/10.5281/zenodo.7017943)[69].

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

## Acknowledgements

The authors thank Dr. Cheng Chang (BPRC), for his helpful suggestions on calculation of the TMT labeling efficiency. P.H. is funded by the Ministry of Science and Technology of China (MOST; 2019YFA0801501, 2016YFA0100500), CAMS Innovation Fund for Medical Sciences (2021-I2M-1-058, 2022-I2M-2-003), Tianjin Natural Science Foundation for Distinguished Young Scholars (21JCJQJC00030) and NSFC grant (31970687). Y.X. is funded by National Key R & D Program of China (2021YFF0702000, 2021ZD0201300), NSFC grants (31930021, 31970633), Hubei Natural Science Foundation for Innovative Research Groups (2021CFA005). Y.S. is funded by NSFC grants (82171148, 81822011) and National Key R & D Program of China (2020YFA0908201). X.Z. is funded by Shanghai Sailing Plan for the Young Scientific Talents (19YF1434000). C.W. is funded by Chinese Postdoctoral Science Foundation (2020M682395). Genesis Technology Communication (Beijing) Co., ltd. reviewed the manuscript prior to submission.

## Author contributions

Y.X. and P.H. designed the project. Y.Y., P.H., X.Z., M.L., L.L. and L.M. performed the experiments and analyzed the data. Y.X., C.W., and S.L. developed the CKI algorithm. S.L., C.W., W.D., J.Z. and Y.X. analyzed the transcriptomic and phosphoproteomic data. P.H., W.P., J.C. Q.W. and Y.S. generated human fibroblasts, and contributed to the data analysis. P.H., Y.X., Y.Y., X.Z., S.L. and Y.S. wrote the manuscript. All authors critically revised and approved the manuscript.

## Competing interests

The Authors declare the following competing interests: Huazhong University of Science and Technology and ShanghaiTech University on behalf of the authors Y.X., P.H., S.L. and Y.Y. have filed a Chinese patent application (201811561138.3) related to the CKI methodology. This patent has been granted and published by China National Intellectual Property Administration. All other authors (C.W., X.Z., M.L., W.D., J.Z., L.L., L.M., W.P., J.C., Q.W. and Y.S.) report no competing interests.
