## [Peer Review File · Nature Communications]

REVIEWER COMMENTS

Reviewer #1 (Remarks to the Author):

In this manuscript, Yuan et al., describe an integrated 'omics'-based approach to identify critical protein kinases underlying a given phenotype. The manuscript is presented as a development of an ensemble method. This method is first applied to existing datasets and then to newly generated data on hepatic reprogramming.

Strengths

-The manuscript deals with a significant area of reverse engineering protein kinases and kinase-associated networks.

-The authors experimentally validated specific findings using both overexpression and knockdown studies and uncovered new kinases that are essential for hepatic reprogramming.

Weaknesses

-The first two sections in the manuscript that describe the development of CKI seem disjointed (Figure 1), and the methodology is poorly defined. It is unclear

Specific comments:

1. Phosphorylation events can be a signal for both activation or inhibition. Without perturbation or prior functional data, it is challenging to assign directionality to phosphorylation events. The authors do not describe whether the directionality of the phosphorylation sites was taken into account and, if so, how.
2. Although they cite iGPS server, the details of how the site-specific kinase-substrate relations were predicted were not provided.
3. For doxorubicin resistance in U2OS cells and lpc-Hep datasets, all of the kinases predicted by CKI were shown to display an increase in phosphorylation. It is unclear whether these kinases could simply be predicted from the mass spec data or how do they rank among all phosphosites? It would be more powerful to show the true positive/false positive rate of prediction kinases from this dataset.
4. Bulk RNAseq data for hepatic cell fate transition contains single sample data points. There's single measurement for GFP, FHH-2.25d, and FHH-5d. It is not possible to determine DEG or enrichment from single measurements accurately.

5. Minor comments:

Figure 1 a, and b are not informative

Reviewer #2 (Remarks to the Author):

In their manuscript “A trans-omics-based method reveals a central kinome that controls hepatic reprogramming” Yuan et al. use quantitative transcriptomics and phospho-proteomics of human fibroblasts overexpressing FOXA3, HNF1A and HNF4A (FHH) to predict 15 central protein kinases possibly involved in hepatic reprogramming. For this, they apply an algorithm they developed coined central kinase inference. In following experiments, they functionally validate 7 of these kinases. They further describe the kinase PIM1 to promote conversion into hepatocytes and to protect fibroblasts from ferroptosis and cell cycle arrest during this process. In conclusion they claim to describe the central kinome regulating hepatic reprogramming and emphasize the use of this approach for analysis of key protein kinases in other biological processes.

The manuscript is in large parts well written - the discussion for example - but it needs to undergo major revision before it can be considered for publication. The accession numbers to RNAseq data did not allow data access

<https://www.ncbi.nlm.nih.gov/sra/?term=PRJNA553504>

and the proteomics dataset identifier IPX0001218000 provided
<https://www.iprox.org//page/project.html?id=IPX0001218000>

leads to drosophila proteomics data of a 2020 Nat Commun. paper. Providing reviewer access to deposited data is important for studies involving collection of large high throughput datasets. Please next time provide correct reviewer access information together with a metadata table that allows to map instrument raw data files to specific experiments.

Figure legends of the generally clear and illustrative figures are often inadequate and should be extensively worked over to describe what the figures show. Also, the methods section is often inadequate as it does not provide essential information on buffers used and chemical reaction conditions. Please see detailed comments below.

An important point to address in the manuscript is to set the performance of the applied “central kinase inference” applied in this study in context to other similar approaches in terms of performance and predictive power. This can be done with existing datasets or needs to be discussed at least. Please in any case discuss the advantages and possible disadvantages and caveats of the approach taken here. What are the limitations of this approach? Did the approach miss known targets?

What is a little surprising is that this study presents phospho-proteomics data obtained after enrichment of phospho-peptides, however the corresponding proteomics data obtained from unphosphorylated

peptides is missing. What is the reason for this? If these data were recorded, they should be included in the manuscript and integrated into data analysis.

Some experiments in different figures follow the same experimental setup. Are the data displayed in figures 3d, 3e and 5e from the same or different experiments? Please indicate.

Specific questions:

Why were some phosphorylated protein kinases (Prkaa1 and MAPK4) IDed by proteomics, but not via mRNA? Why were they omitted from further analysis?

Please provide 2-way hierarchical data clustering in supplementary figures or figures for high-throughput data. How do different experimental conditions cluster?

Could the study benefit from flow cytometry experiments? (experiments Fig. 4d and e)

Some parts of the text between introduction and discussion could benefit from editing to be more concise. Some discussion in the results section is lengthy.

The authors give a general overview of how CKI using RNA-seq and phospho-proteomics with TMT quantification. I am a bit unclear what they mean by network state in line 117. Is network state informed by p-site intensity, RNA-seq, or both? If so, is this really a third type of change or is it just a change inferred by 2 types of changes? Needs further clarification.

Are there any caveats to their prediction algorithm? What do they mean by “excellent performance”?

Are ALB mRNA levels the only indicator for hepatic conversion or is there further characterization? This question arose when reading this part. Suggest to anticipate later characterization here already.

Authors induced hepatic reprogramming with FHH lentivirus and liver-specific genes were induced 2 days post-infection. Broad-spectrum kinase and pY kinase inhibitors suppressed reprogramming. Samples were analyzed by RNA-seq and phospho-proteomics at 2.25 days and at 5 days post-infection. HDFs infected with GFP was a control. Compared FHH-2.25 vs GFP, FHH-5d vs GFP, and FHH-5d vs FHH-2.25d. Where there any comparison with day zero? Can you compare FHH-5d vs GFP since GFP was only at 2.25d?

Please make clear in the text which timepoint after induction of conversion is displayed and if GFP has an effect (relationship of GFP data Fig 3d to 3c?). Why is Fig 3e a separate panel? Is it the same experiment as in 3d?

“it could be found that most of the comparison contributed to the final predictions...”. If comparisons don’t contribute in some way to the algorithm then what does? Please clarify.

How is the enrichment of “cell cycle” and “ferroptosis” interpreted as they are emphasized? Further drug-induced ferroptosis experiments under PIM1 overexpressing conditions are needed in this study (for drug-induced ferroptosis experimental setup please see Liu et al. Nat Cell Biol 22, 476–486 (2020) for example).

Page 18 Please describe comprehensive TPCW more comprehensively.

Page 18 MAFF&HLF found in study. What about the others?

Figure legends:

General comment: Often figure legends only provide a conclusion, not what is shown. Legends should describe the figures adequately standing alone.

Fig1c&e: Please indicate how the kinases displayed by name were selected

Fig1f: Only conclusion provided here. Error bar statistics needs to be indicated. Please also follow this for subsequent figures.

Fig2: Please see comment for Fig1 and change accordingly

Fig2e: How was ALBUMIN stained here?

Fig3: Description of what is displayed needed.

Fig4: Significance criteria?

Fig4c: Statistical test information?

Fig4d: Images not described

Fig4f: Phospho-forms of proteins probed?

Fig5a: Magnification? Scale bar?

Fig6: What do we see here? – inadequate figure legend. What is the color coding? Need cross-reference for 7 central kinases from Fig.1.

Fig6b: Indicate, that this is a subpanel of Fig. 6a

Methods section:

Method section is generally incomplete/inadequate. Some procedures seem similar to previous papers of the same authors, but are clearer described there. Please reference or use similar wording, however still provide full information also partly lacking in previous papers. For example, in this manuscript the pH, salt concentrations and buffering agents are not described for very pH sensitive reactions (reductive alkylation of cell lysate, pH of lysis buffer not provided, pH of TMT labelling reaction not provided). Was the alkylation reaction quenched? Furthermore, TMT labelling reaction is not described. It was stated that it follows the manufacturer's protocol with a labelling time of 2h. The manufacturer's protocol calls for 1h labelling. Please provide all reaction conditions adequately. Digestion conditions are unclear, please clarify. Was the efficiency of TMT labelling determined before further sample processing? What is the TMT labelling efficiency of the peptides measured in the data? Please provide the % of N-terminally TMT labelled peptides vs. unlabeled peptides in the data as indication of TMT labelling efficiency. At which ratios were TMT labelled samples mixed? How were 60 fractions combined into 11?

Please describe IMAC microspheres (unusual term) and enrichment procedure. Titanium? Iron? Enrichment after C18 peptide cleanup? What is the phospho-peptide enrichment %? Was the phospho-peptide enrichment performed after fractionation? If so, why? Was the flowthru of the phospho-enrichment discarded?

Proteins (lysates) were sonicated in the presence of 8M urea. Did the samples get heated or did they remain cool? Sonication in urea is problematic as elevated temperatures lead to protein (and derived peptide) modification. Please comment on this if samples could have been affected and modified in this study. Provide details of sonication.

What is the rate of missed tryptic cleavages in the data? Phosphorylated peptides often have high degrees of missed tryptic cleavage sites. Is the observed missed cleavage rate problematic for TMT quantification? Would there be a benefit to include peptides with missed tryptic cleavage sites >2 for phospho-peptides?

Typos/small edits:

Line 143: change surprised to surprising.

Line 145: suggest "good performance" instead

Line 170: Please change "dramatically" to for example "markedly"

Line 171: “expressions” is usually used in singular. Please also follow in several subsequent instances like Line 189 and following

Line 176: no comma

Line 177: Please explain stability

Line 182: change “evidences have” to “evidence has”. The entire sentence is a run-on and should be a bit re-worded for clarity. This is also in line 310. I think these sentences are actually the same. They should not be copied and pasted. Nat Comm papers should be concise and not repetitive.

Line 221: “Bona fide” isn’t really an appropriate phrase here. “Bona fide” means real, which isn’t quite right since you describe it as having high confidence.

Line 223: “Pairwisely” is not a word

Line 226: Oxford comma needed after GFP. Check that all lists have appropriate oxford commas since they can completely change the meaning of a sentence (also in line 230 and 117, but there could be more)

Line 231: Try not to start a sentence with a number unless it is spelled out.

Figure 2: b) label Venn Diagrams and make lines clearer, c) numbers don’t quite line up with squares on left

Line 232: This sentence can be further clarified. Were 15 PKs randomly selected from those that were detected in ≥ 5 comparisons or were there only 15 PKs that were detected in ≥ 5 comparisons. It isn’t clear as written.

Line 236: change “were” to “are”

Line 249: Please rephrase for clarity

Line 255: can delete “here”

Line 256: Change “its effects” to “their effects” (multiple candidate PKs being discussed)

Line 258: change “expressions of ALB” to “ALB expression”, same for “regulations”, see comment above

Line 261: “dramatically”

Line 264 on ALB mRNA levels instead of hepatic conversion

Line 275: “dramatically”

Line 277: “dramatically”

Line 285: “dramatic” – there are 6 instances of “dramatic” in the manuscript. Maybe not use it at all or only rarely?

Line 301: “supported” to “supports”

Line 324: change “promotes” to “promote”

Line 332: Should this in “In consistent” or “Inconsistent”?

Line 352: Should be expression, not expressions

Line 416: Change “in” to “on”

Line 418: “promoter region” – where is this shown in Fig. 6b?

Line 420: remove “the”

Line 422: TPCW did not demonstrate phosphorylation. It indicated/predicted etc. it.

Line 443: change “phosphorylations” to “phosphorylation”

Line 460: “on the contrary” doesn’t have quite the right meaning and suggests that the previous statement is not true. I would just delete it and start the sentence with “PIM1...”

Overall, I think this is an important study with very interesting and exciting results and a broadly useful approach beyond the specific field. If the authors address the points raised above, I think the study is likely suited for publication in Nature Communications. However as, so far, no access to deposited data has been provided and, more importantly, essential minimal standards for description of figures, statistical analysis and experimental methods were not met, my recommendation is for the manuscript to undergo major revision and for the editor to accept a revised version for submission as new initial submission for peer review at the same journal.

Reviewer #3 (Remarks to the Author):

In this study, the authors combined the data of transcriptome and phosphoproteome to find protein kinases (PKs) that critically regulate the direct reprogramming of human fibroblasts to hepatocyte-like cells (iHep cells). Finally, the authors found PIM1 that contributes to the hepatic reprogramming by regulating cell proliferation and cell survival. Although the role of PIM1 in the hepatic reprogramming is interesting, the present data are insufficient to support their claim, and previous studies may weaken the impact of this study.

Major concerns

1. Fig. 1a: There may be a correlation between the levels of protein phosphorylation and a kinase-substrate network. However, I am wondering whether there is a correlation between the expression levels of PK genes and the phosphorylation levels of PK-target proteins. Normally, the phosphorylation levels of PK-target proteins change in the result of activation/inactivation of cell intrinsic/extrinsic signals regardless of the expression levels of PK genes. Thus, this study may not need a trans-omic analysis, but only need a phosphoproteome analysis. Fig. 1c and e actually support this opinion.

2. Fig. 1b-e: The authors conducted a simulation using public data to show a predictive performance of the CKI algorithm, and the number of reported PKs among the resulting candidates is shown in the text, which is an important argument to estimate the performance of CKI. However, to attain objective evaluation of the prediction method, the number of false negatives, which were not predicted in the CKI algorithm, should also be summarized and disclosed.
3. Fig. 1d: What are hepatic progenitor cells used here? Show the details about those cells, and also show what characterizes them as hepatic progenitor cells.
4. Fig. 2a: The authors previously reported that SV40-LT was required for proliferation of human iHep. Show the details about the method of iHep induction used in this study. If SV40-LT was used, the authors should carefully examine an unexpected effect of SV40-LT in the analysis. If it was not used, iHep cultures may contain many fibroblasts, and thus cell heterogeneity is a problem for analysis. The authors should isolate iHep before the analysis.
5. Fig. 2d-f: Do these data indicate an increase or decrease of ALB expression level in each cell, an increase or decrease of reprogramming efficiency, or an increase or decrease of proliferation of ALB-positive cells? The authors need to answer this question with suitable quantitative data.
6. Fig. 2d-f: In addition to ALB expression, other hepatic marker expression and hepatic function should be examined in the screening. Only ALB expression is not sufficient for proper assessment.
7. Fig. 2d: As I mentioned above, an increase of PK gene expression and that of target protein phosphorylation may be less relevant. Thus, the screening should be done by gene knockdown/knockout experiments, but not by an overexpression strategy. In fact, Fig. 2f shows a significant effect of the knockdown of PLK1, PLK4 or NEK2 on the induction of ALB gene expression in fibroblasts.
8. Fig. 2d: In this study, the authors focused on PKs that upregulate ALB expression, such as PIM1 and PIM2. However, PKs suppressing ALB expression, such as TRIB2 and TSSK3, could also be predicted as the candidates of central PKs in hepatic reprogramming. To evaluate the CKI as a “versatile and comprehensive prediction method for PKs”, the authors should also focus on the PKs that suppress ALB expression and reveal the roles of these PKs, similar to PIM1.
9. Fig. 3a: PIM1 gene expression was increased after 5 days post-infection. However, hepatic gene expression was increased after 2 days post-infection (Supplementary Fig. 2b). Thus, it is suggested that PIM1 affects an increase or maintenance of ALB gene expression in iHep, but not involves in an initial event of hepatic reprogramming. The authors should clarify this important point with additional data.
10. Fig. 3c: qPCR data suggest that shPIM1 specifically affects ALB gene expression, but not involves in the hepatic reprogramming. The authors should do more detailed analysis to examine whether PIM1 is critical for hepatic reprogramming or not. qPCR analysis is not enough, but other in vitro and in vivo functional analysis of induced cells with shPIM1 are required.
11. Fig. 4b: Why were other pathways not analyzed in detail? The data shown in Fig. 5g indicate that ferroptosis inhibition has only a limited effect on the hepatic reprogramming. The authors should verify whether there are other critical pathways.

12. Fig. 4d, e: PIM1 is known as a cell growth inducer. So, it is not surprising that EdU-positive cells increased in fibroblasts overexpressing PIM1 with FHH after 3 days post-infection. Importantly, the endogenous gene expression of PIM1 increased after 5 days post-infection (Fig. 3a). Thus, to reveal the role of PIM1 in the hepatic reprogramming, the authors should do gene knockdown/knockout experiments of PIM1 after 5 days post-infection and examine whether endogenous PIM1 is critical for cell proliferation during reprogramming.

13. Fig. 4f: The authors should examine whether these signaling pathways are activated with an upregulation of endogenous PIM1 expression and inhibited by PIM1 knockdown/knockout during hepatic reprogramming.

14. Fig. 5d: In addition to qPCR data, the authors should show the images and the percentages of dead cells in shPIM1- or control shRNA-expressing cells after 5 days post-infection to confirm the inhibition of ferroptosis by PIM1 during hepatic reprogramming.

15. Fig. 5e: The images and the percentages of dead cells in PIM1- or control GFP-expressing cells should be shown.

16. Fig. 5f suggests that only ferroptosis occurs in the FHH-induced cell death. To confirm this, the authors should examine whether other types of cell death (apoptosis, necrosis, pyroptosis, necroptosis, etc.) do not occur during hepatic reprogramming. In addition, cell images during ferroptosis should be shown.

17. Fig. 5g: Do these data indicate an increase of ALB and CYP3A4 expression levels in each cell, an increase of reprogramming efficiency, or an increase of ALB-positive and CYP3A4-positive cells upon inhibition of ferroptosis? The authors need to answer this question with suitable quantitative data.

18. Fig. 5h, i: Again, to reveal the role of PIM1 during hepatic reprogramming, gene knockdown/knockout experiments of PIM1 are required.

19. Fig. 6: The authors summarized the present results and argued that PIM1 and other PKs regulate the hepatic reprogramming through inhibition of ferroptosis and cell cycle arrest by phosphorylating with related proteins in concert with transcriptional control of them with FHH and down-stream transcription factors. However, it is almost impossible to extract any useful information from such a diagram because the edges connecting each factor are not visible. Therefore, the data should be replaced or listed together with quantitatively summarized data that reveals the relationship between factors and the degree of overlap of the regulation of transcription and that of phosphorylation.

Minor concerns

1. The title of this paper should be changed appropriately along with the data obtained in this study.
2. The methods should be improved, because this paper does not go into enough details about the conditions of the experiments.
3. "Data not shown" may not be approved.
4. There are same sentences in the text (for example, line 182-186 and line 310-314).

5. Page 10, Line 205-214: The authors wrote that 5031 unique phosphopeptides were identified through the phosphoproteomics, but they also wrote that 5660 non-redundant p-sites were detected, so why is the number increased?
6. Page 10, Line 210-214: The ratio of detected p-sites was calculated for each modified amino acid. How does this ratio compare to the genome-wide figure for phosphopeptides?
7. Page 14, Line 298-300: Show the data used for selecting 18 genes in a supplementary figure.
8. Fig. 1c, 1e, and 2b and Suppl. Fig. 1g: There are representative PKs shown in the Venn diagrams with the numbers of candidates, but the numbers do not match. So, the name of PKs should be omitted or shown outside the Venn diagrams.
9. Fig. 4a: To create a list of liver- or fetal liver-enriched genes, curated gene lists from sophisticated databases such as MSigDB or KEGG should be used. At least, the data obtained by analyzing the curated gene lists should be included in supplemental data.
10. Fig. 5h, i: Cells expressing only GFP or PIM1 should also be analyzed additionally as controls.
11. Suppl. Fig. 1a-f, 1h, 1j-l, and 3d: These data should be summarized in a Table.
12. Suppl. Fig. 3b and 4b: Resolution of these figures is too low to read information. They should be replaced.
13. Errors and lack of unity are found throughout the paper. The paper should be carefully rewritten.

Reviewer #4 (Remarks to the Author):

This paper makes a methodological contribution and presents a strong validation component. The methodology can enable genome-wide analyses of protein kinase regulations while the biological findings are relevant to address how PKs are regulated to facilitate hepatic reprogramming.

Authors mention: "To investigate the most important PKs that regulate hepatic conversion, we integrated the transcriptomic and phosphoproteomic data, and developed a trans-omics-based algorithm, central kinase inference (CKI), that infer central PKs in a defined biological process."

-- Since one of the primary goals is method design, proper parameter choices with sensitivity analysis and logical reasoning behind the terms used is needed. This is the main drawback of this paper. In the following, I will highlight some of these shortcomings.

- 1) Several of the parameter choices need justification, such as (this is not a comprehensive list):
 - (i) spectral count of 2.89

(ii) class I (LP > 0.75, 86.60%), 674 class II (0.5 < LP ≤ 0.75, 11.91%) and 141 class III (0.25 ≤ LP ≤ 0.5, 2.49%) p-sites

(iii) factor of 2 (≥ 2-fold or ≤ 0.5-fold change) was employed to pairwise compare phosphoproteomes

(iv) The minimum peptide was set as 7, and the mass tolerance for fragment ions was set as 0.02 Da. FDRs of PSM and protein were set to < 1% and minimum score for modified peptides was set to > 40.

(v) To identify DRPs from the phosphoproteomes, a factor of ≥ 2-fold change was employed for each pair of comparison.

2) To ensure data quality, 15 PKs that were simultaneously detected in ≥ 5 comparisons were reserved as highly potential central PKs (Fig. 2b,c and Supplementary Table 4). Furthermore, for the 15 predicted central PKs, their corresponding p-values calculated in changes of mRNA expression, substrate p-site intensity and network state were shown, respectively (Fig. 2c).

-- how does frequency of detection align with p-values? Is there a one-to-one correlation?

3) The Methods in general need more explanation. The bioinformatics procedures are not novel and standard. However, the CKI algorithm needs a lot more details.

(i) It is not clear how the TF-target network was constructed; similarly how was the kinase-substrate network was constructed? How were the two networks integrated considering different edge weights in each network?

(ii) The authors say: "Then, we found the upstream or downstream regions of 24 TF genes contained at least one predicted TFBS of FHH." Firstly, the grammar is incorrect; next this choice (at least one predicted TFBS) seems arbitrary. For each such arbitrary choice a sensitivity analysis is needed to show how varying this threshold changes the results.

(iii) For identifying central PKs, the intensity method was applied. However, many of the terms introduced lack the reasoning. For example, why is TSI a simple sum of square roots of TMT intensities? Is there a logical justification for this?

For the network based method, how was the site-specific PK-substrate network created? How is RIR calculated? For KNI_U(i), why is a linear sum of RIRs sufficient? For KNI_D(i), the expression is logically the harmonic mean of RIRs; is this correct and what is the justification?

Minor:

There are several typos and grammatical errors. I am highlighting some of these, but there can be more; so please proof-read properly.

Since both doxorubicin and genistein are potent DNA damage inducers²²⁻²⁵, it was not surprising that the major DNA damage response

-- surprised  surprising

It could be found that different substrates were preferentially to be mutually phosphorylated by two distinct clusters of central PKs (Supplementary Fig. 3m).

-- preferentially  preferred

PIM1 has been found to promote cell proliferation via various....

-- promotes  promote

The MS/MS spectra files of freshly isolated mouse MH

-- remove extra "of"

Reviewer #1:

In this manuscript, Yuan et al., describe an integrated 'omics'-based approach to identify critical protein kinases underlying a given phenotype. The manuscript is presented as a development of an ensemble method. This method is first applied to existing datasets and then to newly generated data on hepatic reprogramming.

1. Strengths

-The manuscript deals with a significant area of reverse engineering protein kinases and kinase-associated networks.

-The authors experimentally validated specific findings using both overexpression and knockdown studies and uncovered new kinases that are essential for hepatic reprogramming.

Weaknesses

-The first two sections in the manuscript that describe the development of CKI seem disjointed (Figure 1), and the methodology is poorly defined. It is unclear.

Response: We are grateful for the reviewer's comments on our original description of the CKI methodology, which we regret was not presented in a clear manner. To address this concern, the original Fig. 1 has been split into the new Figs. 1, 2, and 3. In Fig. 1, the implementation of each of the three data types used in CKI are illustrated, and corresponding descriptions of the basic hypotheses of these methods are presented in the section "A trans-omics-based method for prediction of central PKs". In Fig. 2, a re-analysis of two reported drug-resistant studies is shown, and the performance of CKI is critically evaluated and compared to other methods. The corresponding descriptions are shown in the section "Evaluation of the accuracy of CKI". Finally, an in-depth analysis of mouse hepatocyte maturation and predictions of potentially central PKs are shown in Fig. 3 and described in the section "CKI reveals cAMP-dependent PKs as key regulators for hepatocyte maturation". We have revised the manuscript as indicated below:

On page 6, paragraph 1, we changed the portions shown in red:

“A trans-omics-based method for prediction of central PKs

To infer **potentially** central PKs in **defined** biological **processes**, we developed CKI, a trans-omics-based computational method **to** analyzed and integrated transcriptomic data derived from RNA sequencing (RNA-seq) and phosphoproteomic data quantified by tandem mass tag (TMT) labeling coupled with liquid chromatography-tandem mass spectrometry (LC-MS/MS) (Fig. 1). **The basic hypothesis behind CKI was that molecular changes at both the transcriptomic and phosphoproteomic levels might be informative in predicting the functional importance of PKs. Comparisons were made of the transcriptomes and phosphoproteomes in paired samples (control vs. treated), taking three types of changes into consideration to synergistically predict central PKs in response to treatment: mRNA expression, substrate p-site intensity, and kinase-substrate network (Fig. 1), as described below.**

PK expression at the transcriptional level is essential for regulating its constitutive activity^{18,19}. Thus, PKs with differentially expressed mRNAs (DEMs) in response to treatment may be involved in orchestrating downstream signaling. In this study, differentially expressed PKs (DEPKs) were directly identified from transcriptomic data (Fig. 1). From the phosphoproteomic data of each sample, directed relations of PKs with p-sites, referred to as site-specific kinase-substrate relations (ssKSRs), were predicted using a previously developed software package called *in vivo* Group-based Prediction System (iGPS). This program integrates sequence and protein-protein interaction (PPI) information for predicting p-sites specifically modified by 408 human and 416 mouse PKs²⁰. Then, we hypothesized that a PK with higher activity might phosphorylate more p-sites with higher modification levels, and *vice versa*. We therefore developed an intensity-based approach to identify potentially central PKs based on differential intensity of substrate p-sites between paired samples, e.g., treatment vs. control (Fig. 1).

An alternative hypothesis was that a PK with higher activity may produce a more positive impact on its regulatory phosphorylation network, and *vice versa*. We therefore developed a network-based method to measure the network change for each PK. From predicted ssKSRs mutually quantified in a pair of samples, a kinase-substrate phosphorylation network was re-constructed with directed relations

of PKs and p-sites. For each PK, its downstream regulatory network was further split into an up-regulated sub-network (up-regulated substrate p-sites) and a down-regulated sub-network (down-regulated substrate p-sites). A statistical test was performed to identify potentially central PKs statistically associated with up- or down-regulated sub-network modules (Fig. 1). From all pairwise comparisons, the number of positive hits were counted for each PK as the only measure to prioritize the final candidate PKs.

Evaluation of the accuracy of CKI

CKI is a model-based method, and no prior data were used for training. To evaluate the performance of CKI, we predicted potentially central PKs using data from two types of previously published drug-resistance studies, DOX resistance and genistein resistance (Fig. 2, Supplementary Fig. 1 and Supplementary Data 1)¹⁴⁻¹⁷.

In 2016, Zanotto-Filho et al. profiled the transcriptomic changes of a human breast cancer cell line (MDA-MB231) and osteosarcoma cell line (U-2 OS) using a number of chemotherapeutic agents including DOX (1 μ M/L, 8 h), and uncovered a regulatory role of nuclear factor erythroid 2-related factor 2 (NFE2L2/NRF2) in chemotherapy resistance¹⁴. In another study, Hoglebe et al. quantified the phosphoproteomes of U-2 OS cells with or without DOX treatment (5 μ M/L, 2 h)¹⁵. From these two studies^{14,15}, we identified 2197 DEMs and obtained quantitative data for 27,590 p-sites in 5592 proteins (Fig. 2a, b). In the phosphoproteomic data, there were 22,744 phosphoserine residues (pS, 82.44%), 4451 phosphothreonine residues (pT, 16.13%), and 395 phosphotyrosine residues (pY, 1.43%) (Fig. 2c). From the literature, known PKs involved in DOX resistance were curated (Supplementary Data 1). Using this independent testing dataset, we evaluated the accuracy of CKI by calculating the AUC score of the method that incorporated all three types of data (mRNA expression, substrate p-site intensity, and kinase-substrate network) and of models built with each data type alone (Fig. 2d). We also compared to a z-score-based method, Kinase-Substrate Enrichment Analysis (KSEA), which predicts potentially important PKs only from phosphoproteomic data^{21,22}. We found that using only data related to changes in substrate p-site intensity or the kinase-substrate network had a comparable accuracy to that of KSEA (AUC = 0.7635, 0.8058, and 0.7871,

respectively) (Fig. 2d). However, when the three types of data were integrated with CKI, the AUC value was higher than the other methods (0.8278) (Fig. 2d). The confusion matrix and principal component analysis (PCA) indicated that known and unknown PKs could be separated in a reasonably accurate manner (Fig. 2e, f). Of the final 22 PKs predicted by CKI, 10 had already been reported as truly important for DOX resistance in U-2 OS cells, including ATM/ATR²³, MAPK13²⁴, and CHEK1²⁵ (Fig. 2g and Supplementary Data 1). In this case, the transcriptomic data were less informative and only contributed to prioritization of one of the 22 final candidate PKs (Fig. 2g and Supplementary Fig. 1a). In addition, we mapped the quantified p-sites to the 22 PKs where available, and found that only one PK activity-associated p-site, ATM S1981²⁶, was up-regulated after DOX treatment (Fig. 2g, Supplementary Fig. 1b and Supplementary Data 1).

Next, we integrated the transcriptomic and phosphoproteomic data of breast cancer cells treated with or without genistein from previous reports^{16,17}. In total, we identified 3264 DEMs and obtained quantitative data for 17,403 p-sites in 7149 proteins (Fig. 2h-j). Using known genistein-associated PKs as the testing data, the AUC value of CKI was calculated as 0.7912, showing a much higher accuracy than all of the other methods tested (Fig. 2k). Again, the results from the confusion matrix and PCA supported CKI could distinguish between known and unknown PKs (Fig. 2l, m). In this case, the transcriptomic data also contributed to prioritizing five of the 21 final candidate PKs (Fig. 2n, Supplementary Fig. 1c and Supplementary Data 1). No PK activity-associated p-sites were identified to be differentially regulated after genistein treatment (Supplementary Fig. 1d and Supplementary Data 1).

Because both DOX and genistein are potent DNA damage inducers¹⁴⁻¹⁷, it was not surprising that the major PK known to respond to DNA damage, ATM, was successfully predicted as a central PK in both cases (Fig. 2g, n). Taken together, the results showed a promising performance of CKI in well-studied drug-resistant cancer cells.

CKI reveals cAMP-dependent PKs as key regulators for hepatocyte maturation

To further validate the accuracy of CKI, we analyzed the hepatocyte maturation process in mouse liver progenitor cells (Fig. 3a). We converted mouse hepatocytes to

proliferative bipotent liver progenitor cells (lpc) using a previously established protocol²⁷. The lpc cells could then differentiate into either CK19+ biliary epithelial cells (lpc-BECs) that can form ductal structures (Fig. 3b, c and e) or ALBUMIN+ hepatocytes (lpc-Heps, Fig. 3d, e). However, lpc-Heps generated with this differentiation protocol are relatively immature²⁷, meaning that further lpc-Hep maturation is required for the application of liver-progenitor-cell-derived hepatocytes. To identify central PKs that potentially promote the maturation of lpc-Hep cells, we profiled the transcriptomes and phosphoproteomes of freshly isolated primary mouse hepatocytes (MHs) and lpc-Hep cells (Fig. 3a).

From the transcriptomic profiling, we detected 2136 DEMs out of 23,558 quantified genes (Fig. 3f-h and Supplementary Data 2). Using pathway and biological process annotations from the Kyoto Encyclopedia of Genes and Genomes (KEGG)²⁸ and Gene Ontology (GO)²⁹, respectively, we performed functional enrichment analyses of the DEMs. lpc-Hep cells were deficient in metabolic pathways compared to MHs (Supplementary Fig. 2a, b). From the phosphoproteomic data, we obtained 10,818 quantified p-sites, including 9478 pS (87.61%), 1251 pT (11.57%), and 89 pY (0.82%) residues in 3575 proteins (Fig. 3i, j). To test the quality of the raw MS/MS data, we found that 4128 (44.41%) phosphopeptides could be traced by ≥ 2 spectral counts, with an average spectral count of 2.19 counts per phosphopeptide (Fig. 3k). Based on the localization probability (LP) score³⁰ derived from MaxQuant³¹, we identified 9110 class I (LP > 0.75, 84.21%), 1452 class II (0.5 < LP \leq 0.75, 13.42%), and 256 class III (0.25 \leq LP \leq 0.5, 2.37%) p-sites (Fig. 3l). From eight public p-site databases, we found that 9839 (90.95%) of the p-sites identified in our data were annotated in at least one database (Fig. 3m). The distribution of fragments per kilobase of exon per million fragments mapped (FPKM) values of mRNAs and TMT intensities of p-sites were similar across different samples (Supplementary Fig. 2c, d), suggesting that neither transcriptomes nor phosphoproteomes markedly changed during hepatocyte maturation. Two-way hierarchical clustering was performed by calculating Spearman's correlation coefficient for the transcriptomic or phosphoproteomic data between pairs of samples. The results indicated that lpc-Hep and MH cells could be unambiguously distinguished (Supplementary Fig. 2e, f). These results demonstrated the high quality of the trans-omic profiling.

Using CKI, we predicted 28 potentially central PKs that may promote the maturation of lpc-Hep cells, and each of the three types of data contributed to the final predictions (Fig. 3n, o and Supplementary Data 2). Again, we found that only one PK activity-associated p-site, Cilk1 T157, was up-regulated in lpc-Hep cells (Supplementary Fig. 2g). Prior to further validations, *Prkaal* was removed from the screen because it was undetectable in lpc-Hep and MH cells, and *Nrk3* was removed because it is too large to be packaged into lentivirus. We then screened the 26 remaining candidates by overexpression of individual PKs in liver progenitor cells before induction of hepatic differentiation. Quantitative real-time PCR (qRT-PCR) showed that *Prkaca*, *Prkacb*, and *Prkx*, all of which encode catalytic subunits of cAMP-dependent PKs, promoted the expression of several liver-enriched metabolic genes (Fig. 3p). Previous reports demonstrated that cAMP is critical for the maturation of hepatocytes³², and our results further identified *Prkaca*, *Prkacb*, and *Prkx* as the key genes in cAMP signaling during hepatocyte maturation. We also analyzed the phosphoproteomic data using KSEA^{21,22}, which predicted 19 potentially functional PKs, among which the three newly identified PKs were not included (Fig. 3o and Supplementary Fig. 2h). The successful identification of cAMP-dependent PKs as key regulators for hepatocyte maturation further validated the reliability and accuracy of CKI.”

2. Specific comments:

Phosphorylation events can be a signal for both activation or inhibition. Without perturbation or prior functional data, it is challenging to assign directionality to phosphorylation events. The authors do not describe whether the directionality of the phosphorylation sites was taken into account and, if so, how.

Response: In this study, we did not consider the directionality (activation or inhibition) of phosphorylation events because the functional impacts of most p-sites were not experimentally characterized. The lack of this information did not weaken the usefulness of CKI; three and two potentially central PKs were experimentally validated to be associated with mouse hepatocyte maturation and human hepatic

reprogramming, respectively. We do believe that including the directionality of phosphorylation events in the near future will undoubtedly increase the accuracy of the method. We indicate this point in the revised manuscript as shown below:

On page 24, paragraph 3, we added:

“In processing the phosphoproteomic data, the directionality of phosphorylation events (activation vs. inhibition) was not considered, because this information was not available for most of the p-sites....”

On page 26, paragraph 2, we added:

“...Second, the directionality of phosphorylation was not considered due to limitations in the available data. Including perturbation or prior functional data to distinguish between activating and inhibiting phosphorylation events is also expected to improve prediction performance....”

3. Although they cite iGPS server, the details of how the site-specific kinase-substrate relations were predicted were not provided.

Response: In the revision, details about the prediction of site-specific kinase-substrate relations are provided in the section “The CKI algorithm” in the Methods. We revised the manuscript as indicated below:

On page 45, paragraph 2, we changed the portions shown in red:

“For identified p-sites in this study, their upstream regulatory PKs were first predicted by iGPS (<http://igps.biocuckoo.org/>)²⁰. All p-sites were prepared in the tab-delimited Phospho.ELM (ELM for short) format⁹¹, including UniProt accession numbers⁸¹, full protein sequences, phosphorylation positions, and residue types. For each sample, the “Batch Predictor” option in iGPS was used, and the ELM file containing all p-sites was directly loaded for a prediction. The probability of a p-site modified by each PK was individually scored, and only results with scores higher than the pre-determined

cut-off values were reserved and outputted. For using iGPS, we selected the default parameters of the “Low threshold” and “Experiment/STRING PPI”.”

4. For doxorubicin resistance in U2OS cells and lpc-Hep datasets, all of the kinases predicted by CKI were shown to display an increase in phosphorylation. It is unclear whether these kinases could simply be predicted from the mass spec data or how do they rank among all phosphosites?

Response: In biochemical experiments, the changes in PK activity can be monitored using the activity-associated p-sites in PKs. In the revision, we present the changes in all p-sites, including activity-associated p-sites quantified in the PKs, if available for DOX resistance (Supplementary Fig. 1b), genistein resistance (Supplementary Fig. 1d), mouse hepatocyte maturation (Supplementary Fig. 2g), and human hepatic reprogramming (Supplementary Fig. 4i) datasets. The results indicated that very few activity-associated p-sites could be quantified due to the relatively low coverage of phosphoproteomic profiling. We added the corresponding descriptions to the revised manuscript as shown below:

On page 9, paragraph 1, we added:

“...In addition, we mapped the quantified p-sites to the 22 PKs where available, and found that only one PK activity-associated p-site, ATM S1981²⁶, was up-regulated after DOX treatment (Fig. 2g, Supplementary Fig. 1b and Supplementary Data 1).”

On page 10, paragraph 1, we added:

“...No PK activity-associated p-sites were identified to be differentially regulated after genistein treatment (Supplementary Fig. 1d and Supplementary Data 1).”

On page 12, paragraph 2, we added:

“...Again, we found that only one PK activity-associated p-site, Cilk1 T157, was up-regulated in lpc-Hep cells (Supplementary Fig. 2g).”

On page 25, paragraph 3, we added:

“...Previously, it was reported that the activity of a PK could be monitored by measuring its activity-associated p-sites^{57,58}. For example, the phosphorylation level of PLK1 at T210 is positively correlated with its PK activity^{57,58}. However, due to the relatively low coverage of phosphoproteomic profiling, we only detected two PK activity-associated p-sites that were up-regulated after DOX treatment (ATM S1981²⁶) or in lpc-Hep cells (Cilk1 T157⁵⁹) (Fig. 1g, Supplementary Fig. 1b, Supplementary Fig. 2g, and Supplementary Data 1). We did not find any PK activity-associated p-sites to be differentially regulated in genistein resistance or hepatic reprogramming....”

In the Supplementary Information, page 14, paragraph 2, we added:

“Moreover, we mapped the quantified p-sites to the 15 PKs if available, and only found two p-sites of S1773 and S1374 in KALRN and ROCK2, respectively, and neither exhibited a ≥ 2 -fold change upon FHH infection (Supplementary Fig. 4j). Of note, none of the two p-sites were associated with their corresponding PK activities. Thus, it was not possible for us to characterize differentially activated or inhibited PKs directly from quantified p-sites....”

5. It would be more powerful to show the true positive/false positive rate of prediction kinases from this dataset.

Response: To address this concern, true positive (*TP*), true negative (*TN*), false positive (*FP*), and false negative (*FN*) values were counted for the DOX resistance and genistein resistance datasets, using known central PKs manually collected from the literature as independent testing datasets. The true positive rate, also referred to as sensitivity (*Sn*), and the false positive rate, which is equal to 1 - specificity (*Sp*), were calculated at different thresholds and used to generate ROC curves. Area under the curve (AUC) values were computed to evaluate and compare the performance of CKI

to other methods (Fig. 2d, k). Confusion matrices and PCA analyses are now also provided (Fig. 2e, f, l, m). We added a new section to the Methods describing this, “Performance evaluation”, as shown below:

On page 49, paragraph 3, we added:

“Performance evaluation

For the DOX or genistein resistance, known regulatory PKs curated from the literature were taken as the positive data, while other remaining PKs predicted with at least one ssKSR were taken as the negative data. To evaluate the accuracy of CKI and other methods, true positive (*TP*), true negative (*TN*), false positive (*FP*), and false negative (*FN*) values were counted, and we calculated 2 commonly-used measurements, including sensitivity (*Sn*) and specificity (*Sp*) as below:

$$Sn = \frac{TP}{TP + FN}, Sp = \frac{TN}{TN + FP}$$

The receiver operating characteristic (ROC) curves were illustrated based on *Sn* and $1-Sp$ values, and the corresponding AUC scores were calculated.

Then, the confusion matrix under a selected threshold of pairwise comparisons was calculated as below:

		Actual	
		Known	Unknown
Predicted	Known	$\frac{TP}{TP + FN}$	$\frac{FN}{TP + FN}$
	Unknown	$\frac{FP}{FP + TN}$	$\frac{TN}{FP + TN}$

For each of known or unknown PKs, the number of positive hits and the sum of all minus-log transformed (-lg) *p* values were counted for PCA analysis.”

6. Bulk RNAseq data for hepatic cell fate transition contains single sample data points. There's single measurement for GFP, FHH-2.25d, and FHH-5d. It is not possible to determine DEG or enrichment from single measurements accurately.

Response: Either biological or technical replicates can be used to evaluate the reproducibility of transcriptomic or phosphoproteomic profiling. However, even if multiple replicates were analyzed, the new findings may still not be accurate merely based on the statistical analysis. We believe that all predictions derived from the omic profiling must be rigorously validated with additional future experiments to ensure the reliability of the new findings. Although only single measurements of transcriptomes and phosphoproteomes were conducted for the GFP, FHH-2.25d, and FHH-5d samples, we still validated two potentially central PKs associated with human hepatic reprogramming from 15 candidate PKs prioritized by CKI. These results indicate that the use of single replicate measurements did not weaken the usefulness of CKI. In Cuffdiff and edgeR, identification of potential DEGs from two-sample comparisons is permissive, likely in a less accurate manner.

7. *Minor comments:*

Figure 1 a, and b are not informative

Response: The original Fig. 1 was split into the new Figs. 1, 2, and 3, and additional details were added. The corresponding descriptions were changed in the main text.

Reviewer #2:

In their manuscript “A trans-omics-based method reveals a central kinome that controls hepatic reprogramming” Yuan et al. use quantitative transcriptomics and phospho-proteomics of human fibroblasts overexpressing FOXA3, HNF1A and HNF4A (FHH) to predict 15 central protein kinases possibly involved in hepatic reprogramming. For this, they apply an algorithm they developed coined central kinase inference. In following experiments, they functionally validate 7 of these kinases. They further describe the kinase PIM1 to promote conversion into hepatocytes and to protect fibroblasts from ferroptosis and cell cycle arrest during this process. In conclusion they claim to describe the central kinome regulating

hepatic reprogramming and emphasize the use of this approach for analysis of key protein kinases in other biological processes.

1. The manuscript is in large parts well written - the discussion for example - but it needs to undergo major revision before it can be considered for publication. The accession numbers to RNAseq data did not allow data access

<https://www.ncbi.nlm.nih.gov/sra/?term=PRJNA553504>

and the proteomics dataset identifier IPX0001218000

provided <https://www.iprox.org//page/project.html?id=IPX0001218000>

leads to drosophila proteomics data of a 2020 Nat Commun. paper. Providing reviewer access to deposited data is important for studies involving collection of large high throughput datasets. Please next time provide correct reviewer access information together with a metadata table that allows to map instrument raw data files to specific experiments.

Response: We have updated the Reporting Summary file to ensure access to the correct files and have provided additional details here related to data accessibility for reviewers:

“The RNA-seq data has been deposited into NCBI Gene Expression Omnibus (GEO, <https://www.ncbi.nlm.nih.gov/geo/>) with the dataset identifier GSE169702 (<https://www.ncbi.nlm.nih.gov/geo/query/acc.cgi?acc=GSE169702>, password: cludksmivraxrqf). The mass spectrometry proteomics data including the annotated mass spectra have been deposited into the integrated proteome resources (iProX, <http://www.iprox.org/>) with the dataset identifier IPX0001681000 (<https://www.iprox.cn/page/PSV023.html?url=1635517905832Pq2I>, password: fohA).”

Reviewers can use the dataset identifier and password to access the transcriptomic or phosphoproteomic data. All of the primary datasets will be freely available for broad use after publication of this study.

2. Figure legends of the generally clear and illustrative figures are often inadequate and should be extensively worked over to describe what the figures show. Also, the methods section is often inadequate as it does not provide essential information on buffers used and chemical reaction conditions. Please see detailed comments below.

Response: All figure legends have been carefully revised to provide additional detail. The Methods section has been significantly updated to provide essential information about buffers used and chemical reaction conditions.

3. An important point to address in the manuscript is to set the performance of the applied “central kinase inference” applied in this study in context to other similar approaches in terms of performance and predictive power. This can be done with existing datasets or needs to be discussed at least.

Response: We agree with this point. From the scientific literature, we manually curated known PKs associated with DOX resistance or genistein resistance as independent datasets to test the accuracy of CKI in comparison to other methods. We split the original Fig. 1 into the new Figs. 1, 2, and 3. Evaluation of CKI using the manually curated testing dataset is presented in Fig. 2. We added a new section entitled “Evaluation of the accuracy of CKI” in the Results to present the details of this new analysis. The corresponding methodology related to performance evaluation is included in a new section entitled “Performance evaluation” in the Methods. We revised the manuscript as indicated below:

On page 8, paragraph 1, we changed the portions shown in red:

“Evaluation of the accuracy of CKI

CKI is a model-based method, and no prior data were used for training. To evaluate the performance of CKI, we predicted potentially central PKs using data from two types of previously published drug-resistance studies, DOX resistance and genistein resistance (Fig. 2, Supplementary Fig. 1 and Supplementary Data 1)¹⁴⁻¹⁷.

In 2016, Zanotto-Filho et al. profiled the transcriptomic changes of a human breast cancer cell line (MDA-MB231) and osteosarcoma cell line (U-2 OS) using a number of chemotherapeutic agents including DOX (1 μ M/L, 8 h), and uncovered a regulatory role of nuclear factor erythroid 2-related factor 2 (NFE2L2/NRF2) in chemotherapy resistance¹⁴. In another study, Hoglebe et al. quantified the phosphoproteomes of U-2 OS cells with or without DOX treatment (5 μ M/L, 2 h)¹⁵. From these two studies^{14,15}, we identified 2197 DEMs and obtained quantitative data for 27,590 p-sites in 5592 proteins (Fig. 2a, b). In the phosphoproteomic data, there were 22,744 phosphoserine residues (pS, 82.44%), 4451 phosphothreonine residues (pT, 16.13%), and 395 phosphotyrosine residues (pY, 1.43%) (Fig. 2c). From the literature, known PKs involved in DOX resistance were curated (Supplementary Data 1). Using this independent testing dataset, we evaluated the accuracy of CKI by calculating the AUC score of the method that incorporated all three types of data (mRNA expression, substrate p-site intensity, and kinase-substrate network) and of models built with each data type alone (Fig. 2d). We also compared to a *z*-score-based method, Kinase-Substrate Enrichment Analysis (KSEA), which predicts potentially important PKs only from phosphoproteomic data^{21,22}. We found that using only data related to changes in substrate p-site intensity or the kinase-substrate network had a comparable accuracy to that of KSEA (AUC = 0.7635, 0.8058, and 0.7871, respectively) (Fig. 2d). However, when the three types of data were integrated with CKI, the AUC value was higher than the other methods (0.8278) (Fig. 2d). The confusion matrix and principal component analysis (PCA) indicated that known and unknown PKs could be separated in a reasonably accurate manner (Fig. 2e, f). Of the final 22 PKs predicted by CKI, 10 had already been reported as truly important for DOX resistance in U-2 OS cells, including ATM/ATR²³, MAPK13²⁴, and CHEK1²⁵ (Fig. 2g and Supplementary Data 1). In this case, the transcriptomic data were less informative and only contributed to prioritization of one of the 22 final candidate PKs (Fig. 2g and Supplementary Fig. 1a). In addition, we mapped the quantified p-sites to the 22 PKs where available, and found that only one PK activity-associated p-site,

ATM S1981²⁶, was up-regulated after DOX treatment (Fig. 2g, Supplementary Fig. 1b and Supplementary Data 1).

Next, we integrated the transcriptomic and phosphoproteomic data of breast cancer cells treated with or without genistein from previous reports^{16,17}. In total, we identified 3264 DEMs and obtained quantitative data for 17,403 p-sites in 7149 proteins (Fig. 2h-j). Using known genistein-associated PKs as the testing data, the AUC value of CKI was calculated as 0.7912, showing a much higher accuracy than all of the other methods tested (Fig. 2k). Again, the results from the confusion matrix and PCA supported CKI could distinguish between known and unknown PKs (Fig. 2l, m). In this case, the transcriptomic data also contributed to prioritizing five of the 21 final candidate PKs (Fig. 2n, Supplementary Fig. 1c and Supplementary Data 1). No PK activity-associated p-sites were identified to be differentially regulated after genistein treatment (Supplementary Fig. 1d and Supplementary Data 1).

Because both DOX and genistein are potent DNA damage inducers¹⁴⁻¹⁷, it was not surprising that the major PK known to respond to DNA damage, ATM, was successfully predicted as a central PK in both cases (Fig. 2g, n). Taken together, the results showed a promising performance of CKI in well-studied drug-resistant cancer cells.”

On page 49, paragraph 3, we added:

“Performance evaluation

For the DOX or genistein resistance, known regulatory PKs curated from the literature were taken as the positive data, while other remaining PKs predicted with at least one ssKSR were taken as the negative data. To evaluate the accuracy of CKI and other methods, true positive (*TP*), true negative (*TN*), false positive (*FP*), and false negative (*FN*) values were counted, and we calculated 2 commonly-used measurements, including sensitivity (*Sn*) and specificity (*Sp*) as below:

$$Sn = \frac{TP}{TP + FN}, Sp = \frac{TN}{TN + FP}$$

The receiver operating characteristic (ROC) curves were illustrated based on *Sn* and $1-Sp$ values, and the corresponding AUC scores were calculated.

Then, the confusion matrix under a selected threshold of pairwise comparisons was calculated as below:

		Actual	
		Known	Unknown
Predicted	Known	$\frac{TP}{TP + FN}$	$\frac{FN}{TP + FN}$
	Unknown	$\frac{FP}{FP + TN}$	$\frac{TN}{FP + TN}$

For each of known or unknown PKs, the number of positive hits and the sum of all minus-log transformed (-lg) *p* values were counted for PCA analysis.”

4. Please in any case discuss the advantages and possible disadvantages and caveats of the approach taken here. What are the limitations of this approach?

Response: We have included a discussion of CKI’s limitations in the revised manuscript as shown below:

On page 28, paragraph 2, we added:

“There are some limitations of this study that should be noted. First, we only considered transcriptomic and phosphoproteomic data for prediction of potentially central PKs. However, protein expression levels are also important for regulating PK activity^{18,19}. Furthermore, epigenetic mechanisms⁷⁰⁻⁷⁵ play a critical role in regulating gene expression in many biological processes, especially cell lineage reprogramming. Thus, it could reasonably be expected that incorporation of quantitative proteomic data, DNA methylomic data, and genome-wide histone modification data would improve model accuracy. Second, the directionality of phosphorylation was not considered due to limitations in the available data. Including perturbation or prior functional data to distinguish between activating and inhibiting phosphorylation events is also expected to improve prediction performance. Finally, we tested the accuracy of CKI in only four biological processes. More processes will be considered in the future to further evaluate the reliability and accuracy of CKI.”

5. *Did the approach miss known targets?*

Response: Yes. In Fig. 2d and 2k, the AUC-ROC values of CKI for prediction of potentially central PKs known to be associated with DOX resistance or genistein resistance are shown: 0.8278 and 0.7912. The confusion matrices shown in Fig. 2e and 2l also indicated that a considerable proportion of known PKs were missed by CKI. However, this limitation did not diminish the usefulness of CKI, because three and two new central PKs were experimentally validated as potentially associated with mouse hepatocyte maturation and human hepatic reprogramming, respectively.

6. *What is a little surprising is that this study presents phospho-proteomics data obtained after enrichment of phospho-peptides, however the corresponding proteomics data obtained from unphosphorylated peptides is missing. What is the reason for this? If these data were recorded, they should be included in the manuscript and integrated into data analysis.*

Response: Two of the corresponding authors were primarily interested in and familiar with analyzing transcriptomic and phosphoproteomic data, which is why these were included in the initial model. We agree that the proteomic data are important and will include these data in our future studies, and have added this point as a limitation of CKI in the last paragraph of the Discussion section.

7. *Some experiments in different figures follow the same experimental setup. Are the data displayed in figures 3d, 3e and 5e from the same or different experiments? Please indicate.*

Response: The data displayed in Fig. 3d, 3e and 5e (revised Fig. 5e, 5f, and 7e) were from the same experiment. We clarified this in the figure legends accordingly.

8. *Specific questions:*

Why were some phosphorylated protein kinases (Prkaa1 and MAPK4) IDed by proteomics, but not via mRNA? Why were they omitted from further analysis?

Response: We apologize for the misleading descriptions in the original manuscript. The corresponding texts have been changed as indicated below:

On page 12, paragraph 2, we changed the portions shown in red:

“...Prior to further validations, *Prkaa1* was removed from the screen because it was undetectable in lpc-Hep and MH cells, and *Nrk3* was removed because it is too large to be packaged into lentivirus. We then screened the 26 remaining candidates by overexpression of individual PKs in liver progenitor cells before induction of hepatic differentiation....”

On page 15, paragraph 3, we changed the portions shown in red:

“Prior to further experimental validation, we excluded *KALRN* and *MAPK4* from the study, because *KALRN* is too large to be packaged into lentivirus and *MAPK4* transcripts were not detected during hepatic reprogramming. We overexpressed each of the remaining 13 candidate PKs in HDFs together with FHH to analyze their effects on hepatic conversion....”

9. *Please provide 2-way hierarchical data clustering in supplementary figures or figures for high-throughput data. How do different experimental conditions cluster?*

Response: We thank the reviewer for this helpful suggestion. In the revision, we included two-way hierarchical data clustering for the transcriptomic data from FHH overexpressed with GFP or PIM1 (Fig. 6), the transcriptomic data for lpc-Hep and MH (Supplementary Fig. 2e), and the phosphoproteomic data for lpc-Hep and MH (Supplementary Fig. 2f). The results indicated that different experimental conditions

could be clearly separated by the omic data. We revised the manuscript as indicated below:

On page 12, paragraph 1, we added:

“...Two-way hierarchical clustering was performed by calculating Spearman’s correlation coefficient for the transcriptomic or phosphoproteomic data between pairs of samples. The results indicated that lpc-Hep and MH cells could be unambiguously distinguished (Supplementary Fig. 2e, f). These results demonstrated the high quality of the trans-omic profiling.”

On page 18, paragraph 1, we changed the portions shown in red:

“...Two-way hierarchical clustering of the transcriptomic data demonstrated that FHH-transduced HDFs with *PIM1* or *GFP* overexpression could be clearly distinguished from one another (Fig. 6d)....”

10. Could the study benefit from flow cytometry experiments? (experiments Fig. 4d and e)

Response: We appreciate the reviewer’s suggestion and have incorporated these experiments into the revision. We performed flow cytometry for Fig. 4d and e, and results are shown in revised Fig. 6i,k and Supplementary Fig. 6b,c. The flow cytometry experiments showed that overexpression of FHH suppressed proliferation of HDFs, and PIM1 rescued FHH-induced suppression of cell proliferation. These new results are consistent with the results we previously obtained with EdU staining.

11. Some parts of the text between introduction and discussion could benefit from editing to be more concise. Some discussion in the results section is lengthy.

Response: The Introduction was edited to be more concise. We added the Supplementary Note 1 entitled “Additional experiments and analyses of hepatic reprogramming” to present more details of the basic data analysis.

12. The authors give a general overview of how CKI using RNA-seq and phospho-proteomics with TMT quantification. I am a bit unclear what they mean by network state in line 117. Is network state informed by p-site intensity, RNA-seq, or both? If so, is this really a third type of change or is it just a change inferred by 2 types of changes? Needs further clarification.

Response: We apologize for the imprecise descriptions in the original manuscript. The “network state” is an intermediate concept to measure changes in the kinase-substrate network. To avoid any ambiguity, we only present this concept in the methodology. We carefully described the basic hypothesis and implementation of each of the 3 methods integrated in CKI in the section “A trans-omics-based method for prediction of central PKs” as indicated below:

On page 6, paragraph 1, we changed the portions shown in red:

“A trans-omics-based method for prediction of central PKs

To infer **potentially** central PKs in **defined** biological **processes**, we developed CKI, a trans-omics-based computational method **to** analyzed and integrated transcriptomic data derived from RNA sequencing (RNA-seq) and phosphoproteomic data quantified by tandem mass tag (TMT) labeling coupled with liquid chromatography-tandem mass spectrometry (LC-MS/MS) (**Fig. 1**). **The basic hypothesis behind CKI was that molecular changes at both the transcriptomic and phosphoproteomic levels might be informative in predicting the functional importance of PKs. Comparisons were made of the transcriptomes and phosphoproteomes in paired samples (control vs. treated), taking three types of changes into consideration to synergistically predict central PKs in response to treatment: mRNA expression, substrate p-site intensity, and kinase-substrate network (Fig. 1), as described below.**

PK expression at the transcriptional level is essential for regulating its constitutive activity^{18,19}. Thus, PKs with differentially expressed mRNAs (DEMs) in response to treatment may be involved in orchestrating downstream signaling. In this study, differentially expressed PKs (DEPKs) were directly identified from transcriptomic data (Fig. 1). From the phosphoproteomic data of each sample, directed relations of PKs with p-sites, referred to as site-specific kinase-substrate relations (ssKSRs), were predicted using a previously developed software package called *in vivo* Group-based Prediction System (iGPS). This program integrates sequence and protein-protein interaction (PPI) information for predicting p-sites specifically modified by 408 human and 416 mouse PKs²⁰. Then, we hypothesized that a PK with higher activity might phosphorylate more p-sites with higher modification levels, and *vice versa*. We therefore developed an intensity-based approach to identify potentially central PKs based on differential intensity of substrate p-sites between paired samples, e.g., treatment vs. control (Fig. 1).

An alternative hypothesis was that a PK with higher activity may produce a more positive impact on its regulatory phosphorylation network, and *vice versa*. We therefore developed a network-based method to measure the network change for each PK. From predicted ssKSRs mutually quantified in a pair of samples, a kinase-substrate phosphorylation network was re-constructed with directed relations of PKs and p-sites. For each PK, its downstream regulatory network was further split into an up-regulated sub-network (up-regulated substrate p-sites) and a down-regulated sub-network (down-regulated substrate p-sites). A statistical test was performed to identify potentially central PKs statistically associated with up- or down-regulated sub-network modules (Fig. 1). From all pairwise comparisons, the number of positive hits were counted for each PK as the only measure to prioritize the final candidate PKs.”

13. Are there any caveats to their prediction algorithm? What do they mean by “excellent performance”?

Response: There are several limitations of CKI; we have added a paragraph to the end of the Discussion section to discuss these (please see our response to your Q4).

To clarify our meaning, we have deleted the word “excellent” throughout the manuscript. By “excellent performance” we meant that the discovery of new central PKs potentially associated with mouse hepatocyte maturation and human hepatic reprogramming supports the usefulness of CKI.

14. Are ALB mRNA levels the only indicator for hepatic conversion or is there further characterization? This question arose when reading this part. Suggest to anticipate later characterization here already.

Response: ALBUMIN is a commonly used marker for hepatocytes. However, as suggested by the reviewer, *ALB* is not the only indicator for hepatic conversion. Thus, we quantified the expression levels of two additional hepatic functional genes (Fig. 4j). Among the candidate PKs, PIM1, PIM2, and PIM3 significantly promoted the induction of hepatic functional genes during hepatic reprogramming.

15. Authors induced hepatic reprogramming with FHH lentivirus and liver-specific genes were induced 2 days post-infection. Broad-spectrum kinase and pY kinase inhibitors suppressed reprogramming. Samples were analyzed by RNA-seq and phospho-proteomics at 2.25 days and at 5 days post-infection. HDFs infected with GFP was a control. Compared FHH-2.25 vs GFP, FHH-5d vs GFP, and FHH-5d vs FHH-2.25d. Where there any comparison with day zero? Can you compare FHH-5d vs GFP since GFP was only at 2.25d?

Response: HDFs infected with GFP were used as a fibroblast cell fate control. In this study, we compared fibroblasts not undergoing hepatic cell fate conversion (FHH-2.25d) with those undergoing hepatic cell fate conversion (FHH-5d). We used HDFs infected with GFP, but not uninfected HDFs (i.e., day zero) to best control for the effects of lentiviral infection.

16. Please make clear in the text which timepoint after induction of conversion is displayed and if GFP has an effect (relationship of GFP data Fig 3d to 3c?). Why is Fig 3e a separate panel? Is it the same experiment as in 3d?

Response: The timepoints are indicated in the figure legends, and we have included the timepoints in the revised figures (Fig. 5d-f, 5h, 6e, 6g-m, 7a-i) to increase clarity.

We did not observe obvious effects of GFP on hepatic cell fate conversion (Response Fig. 1). We have used GFP in our previous studies (Huang, et al., Nature, 2011; Huang, et al., Cell Stem Cell, 2014) to control for the effects of lentiviral infection. The control experiment shown in Fig. 3c (updated Fig. 5c) is HDFs transduced with FHH for 0 days. The control experiment shown in Fig. 3d (updated Fig. 5e) is HDFs transduced with FHH and GFP for 5 days.

Response Fig 1. Relative gene expression levels of HDF infected with indicated virus.

The results shown in Fig. 3d. and Fig. 3e were from the same experiment. Fig. 3d (updated Fig. 5e) suggested that PIM1 promotes the induction of hepatic functional genes during hepatic reprogramming. Fig. 3e (updated Fig. 5f) suggested that PIM1 promotes the induction of endogenous HNF4A (one member of FHH).

17. “it could be found that most of the comparison contributed to the final predictions...”. If comparisons don’t contribute in some way to the algorithm then what does? Please clarify.

Response: We found that the changes in mRNA expression, substrate p-site intensity, and the kinase-substrate network differentially contributed to prioritization of final

candidates in different systems (Fig. 2g, 2n, 3o, 4i). We added descriptions of the contribution of transcriptomic data as indicated below:

On page 9, paragraph 1, we added:

“...In this case, the transcriptomic data were less informative and only contributed to prioritization of one of the 22 final candidate PKs (Fig. 2g and Supplementary Fig. 1a)...”

On page 10, paragraph 1, we added:

“...In this case, the transcriptomic data also contributed to prioritizing five of the 21 final candidate PKs (Fig. 2n, Supplementary Fig. 1c and Supplementary Data 1)...”

On page 14, paragraph 2, we changed the portions shown in red:

“...For the 15 candidate PKs, the corresponding *p* values were calculated for changes in mRNA expression, substrate p-site intensity, and kinase-substrate network (Fig. 4i). The mRNA levels of the 15 PKs were not markedly altered in the early stage of hepatic reprogramming (FHH-2.25d vs. GFP), and did not contribute to the final predictions (Fig. 4i)...”

18. How is the enrichment of “cell cycle” and “ferroptosis” interpreted as they are emphasized? Further drug-induced ferroptosis experiments under PIMI overexpressing conditions are needed in this study (for drug-induced ferroptosis experimental setup please see Liu et al. Nat Cell Biol 22, 476–486 (2020) for example).

Response: We apologize for the imprecise descriptions in the original manuscript. We revised the manuscript as indicated below:

On page 18, paragraph 2, we changed the portions shown in red:

“...Here, we performed **KEGG-** and GO-based enrichment analyses, which demonstrated that pathways enriched in *PIM1* overexpression samples covered several main functions of the liver (Fig. 6f and Supplementary Fig. 6a). Interestingly, we observed that two non-metabolic pathways, cell cycle (**KEGG ID: hsa04110**) and ferroptosis (**KEGG ID: hsa04216**), were enriched **in response to** PIM1 overexpression (Fig. 6f), **which is consistent with a previous report⁴⁰....”**

We agree with the reviewer that drug-induced ferroptosis experiments under PIM1-overexpressing conditions are important for this study. In Liu’s study (Liu et al. Nat Cell Biol, 2020, 22, 476–486), the authors induced ferroptosis by deprivation of cysteine, rather than by treatment with drugs. They used KL-11743 and BAY-876, two GLUT1 inhibitors, to promote cysteine-deprivation-induced ferroptosis. However, the GLUT1 inhibitor alone cannot induce ferroptosis. Thus, in this study, we used RSL3 (GPX4 inhibitor) and erastin (cystine-glutamate antiporter system X_c⁻ inhibitor), the two most commonly used ferroptosis inducers, to induce ferroptosis (e.g., Yang, et al., Trends in Cell Biology, 2016; Chu, et al., Nature Cell Biology, 2019; Dixon, et al., Annual Review of Cancer Biology, 2019). As shown in Supplementary Fig. 7g, PIM1 suppressed RSL3-induced ferroptosis in U2-OS cells and erastin-induced ferroptosis in HDFs. These results show that PIM1 can suppress drug-induced ferroptosis.

19. Page 18 Please describe comprehensive TPCW more comprehensively. Page 18 MAFF&HLF found in study. What about the others?

Response: We largely re-wrote the first paragraphs of the section “Network analysis of an integrative TPCW during hepatic reprogramming” to address this concern. The last paragraph of the section “Trans-omic modeling of the hepatic transdifferentiation-associated TPCW” in the Methods was carefully re-written to provide more details about the TPCW. In addition to *MAFF* and *HLF*, we also described the function of *SOX17*. We revised the manuscript as indicated below:

On page 21, paragraph 1, we changed the portions shown in red:

“Theoretically, enforced **expression** of FHH would transcriptionally up-regulate numerous target genes, including TFs and central PKs, in either **a** direct or indirect manner. Transcriptionally-induced central PKs could be activated to phosphorylate protein substrates, including TFs. Thus, **key genes may** be doubly regulated by TFs and central PKs **at both the transcription and phosphorylation** levels to drive hepatic reprogramming. **Here, we modeled a TPCW that contained** the regulatory relationships among FHH, **24** FHH-regulated TFs, **two** screened central PKs, and **60 curated** genes **related to** hepatic lineage, ferroptosis, and cell cycle **that are** potentially modulated by these regulators (Fig. 8a and Supplementary **Data 6**).

In the TPCW, 24 downstream TFs that were predicted to be directly regulated by FHH were found to be significantly up-regulated during the early stage of hepatic conversion (Fig. 8a). **We counted predicted transcription factor binding sites (TFBSs) in the upstream and downstream regions of the 24 TFs; one TF, MAFF, had only one predicted TFBS (Supplementary Fig. 8a). Of the 24 TFs, MAFF and HLF have previously been reported to contribute to hepatic lineage determination in a massively parallel protein activity assay⁵¹. SOX17 is also a critical TF for specification of the endodermal lineage^{52,53}”**

On page 53, paragraph 2, we changed the portion shown in red:

“...In the TPCW, there were 27 TFs including FHH and 24 additional **FHH-regulated** TFs, **2** potentially central PKs, and **60 curated genes of hepatic lineage, ferroptosis, and cell cycle** were adopted for TPCW modeling. A hepatic lineage gene, *TF* (Serotransferrin), is also associated with ferroptosis (Supplementary **Data 6**). In the TF-target network, the nodes denoted TFs or their downstream targets, and the orientations of the edges were defined as TF -> target. In the kinase-substrate network, the nodes denoted PKs or their downstream substrate proteins, and the orientations of the edges were defined as PK -> substrate protein. In the final TPCW, the edge weight was not considered....”

Response: All of the figure legends were carefully revised to describe the relevant computations or experiments.

21. Fig1c&e: Please indicate how the kinases displayed by name were selected

Response: In the revision, the exact gene names of PKs are no longer shown in the Venn diagrams to avoid any unintended misleading implications.

22. Fig1f: Only conclusion provided here. Error bar statistics needs to be indicated. Please also follow this for subsequent figures.

Response: We apologize for omitting this important information. In the revision, Fig. 1f was replaced with Fig. 3p to show the effects of all screened candidate PKs. The corresponding figure legend was revised to “**p Expression levels of liver metabolic genes in lpc-Hep cells overexpressing individual candidate central PK as quantified by qRT-PCR ($n = 3$).**”

23. Fig2: Please see comment for Fig1 and change accordingly

Response: In the revision, the original Fig. 2 was changed to Fig. 4, and the corresponding figure legend was revised.

24. Fig2e: How was ALBUMIN stained here?

Response: ALBUMIN was stained by immunofluorescence staining. In the revision, the original Fig. 2e was updated to Fig. 4k. The corresponding figure legend was changed as shown in red: “**k Representative image of ALBUMIN immunofluorescence staining in HDFs overexpressing the indicated genes after infection with FHH for 12 days. Scale bars = 100 μ m.**”

25. *Fig3: Description of what is displayed needed.*

Response: In revision, the original Fig. 3 was updated to Fig. 5. The corresponding figure legend was revised as requested.

26. *Fig4: Significance criteria?*

Response: In the revision, the original Fig. 4 was updated to Fig. 6. Data are shown as the mean + standard deviation. $*p < 0.05$, $**p < 0.01$, $***p < 0.001$ (2-sided Student's *t*-test). The corresponding figure legend was revised accordingly.

27. *Fig4c: Statistical test information?*

Response: In the revision, the original Fig. 4c was updated to Fig. 6g. Data are shown as the mean + standard deviation. $**p < 0.01$, $***p < 0.001$ (2-sided Student's *t*-test). The statistical test information was added to the end of figure legend of Fig. 6.

28. *Fig4d: Images not described*

Response: In the revision, the original Fig. 4d was updated to Fig. 6h. We updated the description of the images in the figure legend as shown in red: “**h,i** Representative image (**h**) of EdU positive cells on day 3 of hepatic reprogramming using the EdU staining assay.”

29. *Fig4f: Phospho-forms of proteins probed?*

Response: In the revision, the original Fig. 4f was updated to Fig. 6l. The phospho-forms of proteins were labeled along with their corresponding images. We probed phospho-p70 S6 kinase (Thr389), phospho-MYC (Ser62), and phospho-4E-BP1 (Thr37/46).

30. Fig5a: Magnification? Scale bar?

Response: In the revision, the original Fig. 5a was updated to Fig. 7a. The scale bar is 100 μm , which is included in the revised figure legend.

31. Fig6: What do we see here? – inadequate figure legend. What is the color coding? Need cross-reference for 7 central kinases from Fig.1. Fig6b: Indicate, that this is a subpanel of Fig. 6a

Response: The original Fig. 6 was updated in the revision to Fig. 8. The corresponding figure legend was changed as shown in red: “**Fig. 8 Network analysis of the major molecular landscape in hepatic reprogramming. a** Integrative transcription-phosphorylation collaborative web (TPCW) based on new findings of this study and existing knowledge illustrates the regulatory relationships among FHH, 24 FHH-regulated TFs, two screened central PKs, and 60 curated genes related to hepatic lineage, ferroptosis, and cell cycle progression. **b** PIM1-centered subnetwork during hepatic lineage reprogramming. **c** Working model showing potential roles of PIM1 in hepatic lineage reprogramming.”

32. Methods section:

Method section is generally incomplete/inadequate. Some procedures seem similar to previous papers of the same authors, but are clearer described there. Please reference or use similar wording, however still provide full information also partly lacking in previous papers.

For example, in this manuscript the pH, salt concentrations and buffering agents are not described for very pH sensitive reactions (reductive alkylation of cell lysate, pH of lysis buffer not provided, pH of TMT labelling reaction not provided).

Response: The sections including “Protein extraction, isolation of peptides and TMT labeling” and “Phosphopeptide enrichment” in the Methods were carefully revised to provide details about the pH, salt concentrations, and buffering agents used in this study.

33. Was the alkylation reaction quenched? Furthermore, TMT labelling reaction is not described. It was stated that it follows the manufacturer’s protocol with a labelling time of 2h. The manufacturer’s protocol calls for 1h labelling. Please provide all reaction conditions adequately. Digestion conditions are unclear, please clarify. Was the efficiency of TMT labelling determined before further sample processing? What is the TMT labelling efficiency of the peptides measured in the data? Please provide the % of N-terminally TMT labelled peptides vs. unlabeled peptides in the data as indication of TMT labelling efficiency. At which ratios were TMT labelled samples mixed?

Response: We confirmed that the alkylation reaction was quenched by adding 100 mM triethylammonium bicarbonate (TEAB) into the protein sample. In the revision, we have provided additional detail about protein extraction and digestion protocols and TMT labeling. The efficiencies of TMT labeling were higher than 98% in our phosphoproteomic profiling. These details are now provided in the section “Protein extraction, isolation of peptides and TMT labeling” in the Methods as indicated below:

On page 37, paragraph 3, we changed the portions shown in red:

“For the digestion of proteins, protein solution was first reduced with 5 mM dithiothreitol (DTT, pH 8.0) at 56°C for 30 min, and then incubated with 11 mM iodoacetamide (pH 8.0) for 15 min at room temperature in the dark. To reduce the concentration of urea to less than 2 M, 100 mM triethylammonium bicarbonate (TEAB) was added to the protein sample. Two times trypsin digestions were

processed, with the mass ratio of 1:50 trypsin-to-protein for 37°C overnight treatment and 1:100 for 4 h, respectively. Finally, trifluoroacetic acid (TFA) were added to adjust pH to 2-3 for quenching the digestion.

Before the TMT labeling of peptides, peptides were vacuum-dried after the desalting with Strata X C18 SPE column (Phenomenex), and reconstituted with 0.5 M TEAB. Following the manufacturer's protocol with minor modifications, the TMT kit was constituted with acetonitrile, and the equal peptides based on the absorbance at 280 nm were subsequently processed with TMT kit (pH 8.5), and incubated with labeling reagent (Batch 1: MH [126, 127, and 128] and lpc-Hep [129, 130, and 131], and Batch 2: GFP [131], FHH-2.25d [129], and FHH-5d [130]) at room temperature for 2 h to ensure a better efficiency of labelling. Finally, the labeled peptides were equally pooled based on the absorbance at 280 nm, desalted, and dried by vacuum. The efficiency of TMT labelling was measured based on the numbers of N-terminally labelled peptides and all peptides from the Q Exactive™ Plus (Thermo), and obtained 98.59% and 98.50% efficiencies for mouse hepatocyte maturation and human hepatic reprogramming samples, respectively.”

34. *How were 60 fractions combined into 11?*

Please describe IMAC microspheres (unusual term) and enrichment procedure. Titanium? Iron? Enrichment after C18 peptide cleanup? What is the phospho-peptide enrichment %? Was the phospho-peptide enrichment performed after fractionation? If so, why? Was the flowthru of the phospho-enrichment discarded?

Response: In the revised Methods, we provide details for fraction combinations and the enrichment procedure in the section “Phosphopeptide enrichment” as indicated below:

On page 39, paragraph 2, we changed the portions shown in red:

“**Phosphopeptide enrichment**

The tryptic peptides were fractionated into fractions with the high pH reverse-phase **high-performance liquid chromatography (HPLC)**, based on the Thermo Betasil C18 column. The peptides were separated into 60 fractions with a gradient of 8% to 32% acetonitrile (pH 9.0) over 60 min and then pooled into **4 and 11 fractions by combining with equal time interval for mouse hepatocyte maturation and human hepatic reprogramming**, followed by the vacuum drying.

For the phosphopeptide enrichment, the loading buffer (50% acetonitrile & 6% **TFA**) with **Ti⁴⁺-immobilized metal ion affinity chromatography (Ti⁴⁺-IMAC)** microspheres was used to incubate the peptide mixtures **for 1 h at room temperature (30 rpm)**. Next, the IMAC microspheres with enriched phosphopeptides were collected after centrifugation. Then, 50% acetonitrile with 6% **TFA** and 30% acetonitrile with 0.1% **TFA** were used to wash the microsphere **for 10 min at room temperature (30 rpm)** sequentially, to remove the nonspecifically adsorbed peptides. From the elution of phosphopeptides with microspheres by adding elution buffer containing 10% **NH₄OH for 10 min at room temperature (30 rpm)**, the supernatant containing phosphopeptides was collected and lyophilized for further analysis.”

35. Proteins (lysates) were sonicated in the presence of 8M urea. Did the samples get heated or did they remain cool? Sonication in urea is problematic as elevated temperatures lead to protein (and derived peptide) modification. Please comment on this if samples could have been affected and modified in this study. Provide details of sonication.

Response: In the revision, we have provided additional details about cell lysis, and revised the manuscript as indicated below:

On page 37, paragraph 2, we changed the portions shown in red:

“Cell sample was mixed with lysis buffer (8 M urea, 1% **protease inhibitor and phosphatase inhibitor, and 2 mM EDTA, at pH 8.0 with ice-bath**) and homogenized by sonification **at 0°C for 3 min (cycle: sonication for 3 s in ~180 W power, and stop**

for 5 s to cool down). After the centrifugation of lysate at 20,000 g at 4°C for 10 min, the supernatant was transferred and collected. Finally, the protein concentration was measured with 2-D Quant kit and adjusted to be consistent for all samples.”

36. *What is the rate of missed tryptic cleavages in the data? Phosphorylated peptides often have high degrees of missed tryptic cleavage sites. Is the observed missed cleavage rate problematic for TMT quantification? Would there be a benefit to include peptides with missed tryptic cleavage sites >2 for phospho-peptides?*

Response: The rate of missed tryptic cleavages could not be precisely calculated based on phosphoproteomic data. However, the missed cleavage rate merely affects the TMT quantification, because TMT was quantified based on the intensities of reporter ions from the MS² spectrum. The relaxed parameter of missed cleavages could quantify little additional p-sites, but up to two missing cleavage sites is the default setting and is widely used in searching phosphoproteomic data.

37. *Typos/small edits:*

Line 143: change surprised to surprising.

Response: The corresponding description was changed as shown in red: “...it was not surprising that the major DNA damage response...”.

38. *Line 145: suggest “good performance” instead*

Response: We have changed the description as shown in red: “...a promising performance...”.

39. *Line 170: Please change “dramatically” to for example “markedly”*

Response: We have changed the description as shown in red: "...cAMP-dependent PKs, markedly promoted...".

40. Line 171: "expressions" is usually used in singular. Please also follow in several subsequent instances like Line 189 and following

Response: We have changed "expressions" to "expression" throughout the manuscript.

41. Line 176: no comma

Line 177: Please explain stability

Response: The corresponding description was changed as shown in red: "...The successful identification of cAMP-dependent PKs, as key regulators for hepatocyte maturation, further validated the reliability and accuracy of CKI.".

42. Line 182: change "evidences have" to "evidence has". The entire sentence is a run-on and should be a bit re-worded for clarity. This is also in line 310. I think these sentences are actually the same. They should not be copied and pasted. Nat Comm papers should be concise and not repetitive.

Response: We regret that the original phrasing was misleading. It has been deleted in the revision.

43. Line 221: "Bona fide" isn't really an appropriate phrase here. "Bona fide" means real, which isn't quite right since you describe it as having high confidence.

Response: We deleted "Bona fide" from the manuscript.

44. Line 223: "Pairwisely" is not a word

Response: We deleted “pairwisely” from the manuscript.

45. Line 226: Oxford comma needed after GFP. Check that all lists have appropriate oxford commas since they can completely change the meaning of a sentence (also in line 230 and 117, but there could be more)

Response: We have added Oxford commas throughout the manuscript.

46. Line 231: Try not to start a sentence with a number unless it is spelled out.

Response: Such errors have been corrected throughout the manuscript.

47. Figure 2: b) label Venn Diagrams and make lines clearer, c) numbers don't quite line up with squares on left

Response: The original Fig. 2b has been deleted. Only the numbers of PKs are now shown in Venn diagrams; the gene names of PKs are no longer shown.

48. Line 232: This sentence can be further clarified. Were 15 PKs randomly selected from those that were detected in ≥ 5 comparisons or were there only 15 PKs that were detected in ≥ 5 comparisons. It isn't clear as written.

Response: The original description was changed as shown in red: “...CKI integrated all predictions from the **nine** pairwise comparisons **and prioritized** 15 PKs that were **detected** in ≥ 5 **pairwise** comparisons as the final **candidates** (Fig. **4h** and Supplementary **Data 4**)”.

49. Line 236: change “were” to “are”

Response: This change was made.

50. Line 249: Please rephrase for clarity

Response: We apologize that the original description was misleading and have deleted this sentence in the revision.

51. Line 255: can delete “here”

Response: We have fixed this.

52. Line 256: Change “its effects” to “their effects” (multiple candidate PKs being discussed)

Response: We have changed this as shown in red: “...FHH to analyze **their effects** on hepatic conversion.”.

53. Line 258: change “expressions of ALB” to “ALB expression”, same for “regulations”, see comment above

Response: We have changed this as shown in red: “...of the **ALB expression**, ...”.

54. Line 261: “dramatically”

Response: We have changed this as shown in red: “Moreover, **shRNA-mediated** knockdown of *PIM1* or *PIM2* **markedly** decreased...”.

55. Line 264 on ALB mRNA levels instead of hepatic conversion

Response: The sentence was rephrased as shown in red: "...Overexpression of *PLK1*, *PLK4* or *NEK2*, did not show obvious effect on *ALB* mRNA levels (Fig. 4j).".

56. Line 275: "dramatically"

Line 277: "dramatically"

Response: We have changed instances of the word "dramatically" to other words, e.g., "markedly", throughout the manuscript.

57. Line 285: "dramatic" – there are 6 instances of "dramatic" in the manuscript.
Maybe not use it at all or only rarely?

Response: The original phrase has been changed as shown in red: "...Importantly, increased activation of endogenous *HNF4A*, ...".

58. Line 301: "supported" to "supports"

Response: The original description was changed as shown in red: "...This result further supports the critical role of PIM1 in hepatic reprogramming.".

59. Line 324: change "promotes" to "promote"

Response: We changed this as shown in red: "...PIM1 has been found to promote cell proliferation...".

60. Line 332: Should this in "In consistent" or "Inconsistent"?

Response: The original description was changed as shown in red: "...Consistent with a previous report⁴⁰, we..."

61. Line 352: *Should be expression, not expressions*

Response: We changed the description as shown in red: "...significantly promoted the **expression** of hepatic functional..."

62. Line 416: *Change "in" to "on"*

Response: We changed the description as shown in red: "...*PIM1* was not predicted to be directly regulated by FHH **on** the transcriptional level, ...".

63. Line 418: *"promoter region" – where is this shown in Fig. 6b?*

Response: We added a new Supplementary Fig. 7b to present the upstream and downstream regions of the *PIM1* gene that can be regulated by other transcription factors.

64. Line 420: *remove "the"*

Response: We changed the description as suggested: "...which could possibly explain relatively late activation of *PIM1* by day 5 of hepatic conversion...".

65. Line 422: *TPCW did not demonstrate phosphorylation. It indicated/prediced etc. it.*

Response: We changed the description as shown in red: "...the TPCW **indicated** that *PIM1* **might** reciprocally phosphorylate 10 TFs...".

66. Line 443: *change "phosphorylations" to "phosphorylation"*

Response: The word “phosphorylations” was changed to “phosphorylation” throughout the manuscript.

67. Line 460: “on the contrary” doesn’t have quite the right meaning and suggests that the previous statement is not true. I would just delete it and start the sentence with “PIM1...”

Response: We have changed “on the contrary” to “Also”.

68. Overall, I think this is an important study with very interesting and exciting results and a broadly useful approach beyond the specific field. If the authors address the points raised above, I think the study is likely suited for publication in Nature Communications. However as, so far, no access to deposited data has been provided and, more importantly, essential minimal standards for description of figures, statistical analysis and experimental methods were not met, my recommendation is for the manuscript to undergo major revision and for the editor to accept a revised version for submission as new initial submission for peer review at the same journal.

Response: In the revision, access to the deposited data has been provided. The original figures were significantly improved to provide updated results. The figure legends and experimental methods were carefully revised to include additional details. We are very thankful to the reviewer for their kind and helpful suggestions.

Reviewer #3:

1. In this study, the authors combined the data of transcriptome and phosphoproteome to find protein kinases (PKs) that critically regulate the direct reprogramming of human fibroblasts to hepatocyte-like cells (iHep cells). Finally, the authors found PIM1 that contributes to the hepatic reprogramming by regulating cell proliferation and cell survival. Although the role of PIM1 in the hepatic

reprogramming is interesting, the present data are insufficient to support their claim, and previous studies may weaken the impact of this study.

Response: We appreciate the reviewer's kind feedback about our study and thank the reviewer for pointing out that the present data are insufficient to support our claim. To address the specific concerns raised here, we performed additional experiments to support the claims: 1) We analyzed additional hepatic-lineage genes in the screen of candidate PKs with both gene overexpression and knockdown experiments, which further validated the central role of PIM1 among the candidate PKs in hepatic reprogramming (Figure 4j-l, Supplementary Figure 4l). 2) We performed PAS staining and Dil-ac-LDL uptake assays to validate the necessity of PIM1 in hepatic reprogramming with PIM1 knockdowns (Figure. 5d). 3) We performed flow cytometry experiments to quantify the percentage of proliferating cells to validate the key role of PIM1 in promoting cell proliferation during hepatic reprogramming (Figure 6i,k, Supplementary Figure 6b,c). 4) We used immunoblotting to identify signaling pathways downstream of PIM1 (Figure 6l,m, Supplementary Figure 6e). 5) We quantified cell survival to investigate the role of PIM1 in suppressing cell death during hepatic reprogramming. This was accomplished using both overexpression and knockdown experiments (Supplementary Figure 7b-e). 6) We treated cells undergoing hepatic reprogramming with the necroptosis inhibitor Nec-1s and apoptosis inhibitor z-VAD, and found that inhibition of necroptosis or apoptosis does not suppress cell death during hepatic reprogramming (Supplementary Fig 7f). 7) We investigated the inhibitive ability of PIM1 in commonly used experimental systems of ferroptosis, and found that PIM1 could inhibit ferroptosis in these experimental systems (Supplementary Fig. 7g).

As noted by the reviewer, previous studies have contributed important findings about PIM1. For example, PIM1 has been found to play important roles in the generation of iPS cells and has already been found to promote cell cycle in cancer cells. However, in this manuscript, we present the following important new findings: 1) PIM1 is required for efficient hepatic reprogramming. 2) Hepatic reprogramming

could be significantly optimized by overexpression of PIM1. 3) PIM1 can suppress ferroptosis in hepatic reprogramming and other ferroptosis scenarios. 4) PIM1 suppresses ferroptosis by increasing generation of NADPH and glutathione.

2. Major concerns

Fig. 1a: There may be a correlation between the levels of protein phosphorylation and a kinase-substrate network. However, I am wondering whether there is a correlation between the expression levels of PK genes and the phosphorylation levels of PK-target proteins. Normally, the phosphorylation levels of PK-target proteins change in the result of activation/inactivation of cell intrinsic/extrinsic signals regardless of the expression levels of PK genes. Thus, this study may not need a trans-omic analysis, but only need a phosphoproteome analysis. Fig. 1c and e actually support this opinion.

Response: In the revision, we carefully analyzed the correlations between mRNA expression levels of PKs and the corresponding substrate p-site intensities. The weak correlations between the transcriptomic data and the phosphoproteomic data indicated that the two data types are relatively independent, and therefore a trans-omic analysis can provide new insights by integrating the information from the two data types to reach a high accuracy. We clarified this point by revising the manuscript as indicated below:

On page 26, paragraph 1, we added:

“...Using all pairwise comparisons of the transcriptomic and phosphoproteomic data for each biological process, we analyzed correlations between PK gene expression and the intensity of the corresponding substrate p-sites based on the predicted ssKSRs. The average Spearman’s correlation coefficients were calculated as 0, -0.02, 0.14, and 0.10 for DOX resistance, genistein resistance, hepatocyte maturation, and hepatic reprogramming, respectively (Supplementary Fig. 9a-d), implying a weak correlation between PK gene expression and phosphorylation of the target p-sites.”

On page 50, paragraph 4, we added:

“Correlation analysis of mRNA expression levels of PKs and their substrate p-site intensities

The correlations between mRNA expression levels of PKs and their corresponding substrate p-site intensities were measured based on the ssKSRs, using the Spearman’s correlation. For each ssKSR, the Spearman’s correlation coefficient (ρ) between the mRNA expression level of the upstream PK and the intensity of the downstream p-site was calculated across different samples or replicates.”

3. Fig. 1b-e: The authors conducted a simulation using public data to show a predictive performance of the CKI algorithm, and the number of reported PKs among the resulting candidates is shown in the text, which is an important argument to estimate the performance of CKI. However, to attain objective evaluation of the prediction method, the number of false negatives, which were not predicted in the CKI algorithm, should also be summarized and disclosed.

Response: We agree with the reviewer about the importance of including these data. In the revision, we manually curated known PKs associated with DOX or genistein resistance from the scientific literature. These known PKs were used as an independent testing dataset to measure the accuracy of CKI and allow comparison of CKI to other methods. For the testing dataset, we counted the true positive (*TP*), true negative (*TN*), false positive (*FP*), and false negative (*FN*) values and calculated sensitivity (*Sn*) and specificity (*Sp*). ROC curves were generated for each method and the corresponding AUC values calculated for each curve. We have also included confusion matrices and PCA analyses. The original Fig. 1 was split into the new Figs. 1, 2 and 3. Evaluations of CKI using the DOX and genistein resistance testing data are shown in Fig. 2. We added a new section entitled “Evaluation of the accuracy of CKI” in the Results to present these details. The corresponding methodology for performance evaluation has been added in a new Methods section entitled

“Performance evaluation”. To incorporate these changes, the manuscript has been updated as indicated below:

On page 8, paragraph 1, we changed the portions shown in red:

“Evaluation of the accuracy of CKI

CKI is a model-based method, and no prior data were used for training. To evaluate the performance of CKI, we predicted potentially central PKs using data from two types of previously published drug-resistance studies, DOX resistance and genistein resistance (Fig. 2, Supplementary Fig. 1 and Supplementary Data 1)¹⁴⁻¹⁷.

In 2016, Zanotto-Filho et al. profiled the transcriptomic changes of a human breast cancer cell line (MDA-MB231) and osteosarcoma cell line (U-2 OS) using a number of chemotherapeutic agents including DOX (1 μ M/L, 8 h), and uncovered a regulatory role of nuclear factor erythroid 2-related factor 2 (NFE2L2/NRF2) in chemotherapy resistance¹⁴. In another study, Hoglebe et al. quantified the phosphoproteomes of U-2 OS cells with or without DOX treatment (5 μ M/L, 2 h)¹⁵. From these two studies^{14,15}, we identified 2197 DEMs and obtained quantitative data for 27,590 p-sites in 5592 proteins (Fig. 2a, b). In the phosphoproteomic data, there were 22,744 phosphoserine residues (pS, 82.44%), 4451 phosphothreonine residues (pT, 16.13%), and 395 phosphotyrosine residues (pY, 1.43%) (Fig. 2c). From the literature, known PKs involved in DOX resistance were curated (Supplementary Data 1). Using this independent testing dataset, we evaluated the accuracy of CKI by calculating the AUC score of the method that incorporated all three types of data (mRNA expression, substrate p-site intensity, and kinase-substrate network) and of models built with each data type alone (Fig. 2d). We also compared to a z-score-based method, Kinase-Substrate Enrichment Analysis (KSEA), which predicts potentially important PKs only from phosphoproteomic data^{21,22}. We found that using only data related to changes in substrate p-site intensity or the kinase-substrate network had a comparable accuracy to that of KSEA (AUC = 0.7635, 0.8058, and 0.7871, respectively) (Fig. 2d). However, when the three types of data were integrated with CKI, the AUC value was higher than the other methods (0.8278) (Fig. 2d). The confusion matrix and principal component analysis (PCA) indicated that known and unknown PKs could be separated in a reasonably accurate manner (Fig. 2e, f). Of the

final 22 PKs predicted by CKI, 10 had already been reported as truly important for DOX resistance in U-2 OS cells, including ATM/ATR²³, MAPK13²⁴, and CHEK1²⁵ (Fig. 2g and Supplementary Data 1). In this case, the transcriptomic data were less informative and only contributed to prioritization of one of the 22 final candidate PKs (Fig. 2g and Supplementary Fig. 1a). In addition, we mapped the quantified p-sites to the 22 PKs where available, and found that only one PK activity-associated p-site, ATM S1981²⁶, was up-regulated after DOX treatment (Fig. 2g, Supplementary Fig. 1b and Supplementary Data 1).

Next, we integrated the transcriptomic and phosphoproteomic data of breast cancer cells treated with or without genistein from previous reports^{16,17}. In total, we identified 3264 DEMs and obtained quantitative data for 17,403 p-sites in 7149 proteins (Fig. 2h-j). Using known genistein-associated PKs as the testing data, the AUC value of CKI was calculated as 0.7912, showing a much higher accuracy than all of the other methods tested (Fig. 2k). Again, the results from the confusion matrix and PCA supported CKI could distinguish between known and unknown PKs (Fig. 2l, m). In this case, the transcriptomic data also contributed to prioritizing five of the 21 final candidate PKs (Fig. 2n, Supplementary Fig. 1c and Supplementary Data 1). No PK activity-associated p-sites were identified to be differentially regulated after genistein treatment (Supplementary Fig. 1d and Supplementary Data 1).

Because both DOX and genistein are potent DNA damage inducers¹⁴⁻¹⁷, it was not surprising that the major PK known to respond to DNA damage, ATM, was successfully predicted as a central PK in both cases (Fig. 2g, n). Taken together, the results showed a promising performance of CKI in well-studied drug-resistant cancer cells.”

On page 49, paragraph 3, we added:

“Performance evaluation

For the DOX or genistein resistance, known regulatory PKs curated from the literature were taken as the positive data, while other remaining PKs predicted with at least one ssKSR were taken as the negative data. To evaluate the accuracy of CKI and other methods, true positive (*TP*), true negative (*TN*), false positive (*FP*), and false

negative (*FN*) values were counted, and we calculated 2 commonly-used measurements, including sensitivity (*Sn*) and specificity (*Sp*) as below:

$$Sn = \frac{TP}{TP + FN}, Sp = \frac{TN}{TN + FP}$$

The receiver operating characteristic (ROC) curves were illustrated based on *Sn* and 1-*Sp* values, and the corresponding AUC scores were calculated.

Then, the confusion matrix under a selected threshold of pairwise comparisons was calculated as below:

		Actual	
		Known	Unknown
Predicted	Known	$\frac{TP}{TP + FN}$	$\frac{FN}{TP + FN}$
	Unknown	$\frac{FP}{FP + TN}$	$\frac{TN}{FP + TN}$

For each of known or unknown PKs, the number of positive hits and the sum of all minus-log transformed (-lg) *p* values were counted for PCA analysis.”

4. Fig. 1d: What are hepatic progenitor cells used here? Show the details about those cells, and also show what characterizes them as hepatic progenitor cells.

Response: We regret the omission of this important information in the original manuscript. We generated proliferative bipotent hepatic progenitor cells from primary mouse hepatocytes following a well-established dedifferentiation protocol (Katsuda et al., Cell Stem Cell, 2017). The liver progenitor cells are capable of differentiation into both biliary epithelial cells (lpc-BEC) and hepatocytes (lpc-Hep, Fig. 3b-e). The lpc-BEC cells expressed bile duct marker genes, including *CK19*, *Cftr*, *Aqp1* and *Ae2* (Fig. 3e), and can form ductal structures (Fig. 3b,c). The lpc-Hep cells were ALBUMIN positive (Fig. 3d) and expressed the hepatocyte marker genes *Aat* and *Alb* (Fig. 3e).

To clarify these points, we revised the manuscript as indicated below:

On page 10, paragraph 3, we changed the portions shown in red:

“To further validate the accuracy of CKI, we analyzed the hepatocyte maturation process in mouse liver progenitor cells (Fig. 3a). We converted mouse hepatocytes to proliferative bipotent liver progenitor cells (lpc) using a previously established protocol²⁷. The lpc cells could then differentiate into either CK19+ biliary epithelial cells (lpc-BECs) that can form ductal structures (Fig. 3b, c and e) or ALBUMIN+ hepatocytes (lpc-Heps, Fig. 3d, e). However, lpc-Heps generated with this differentiation protocol are relatively immature²⁷, meaning that further lpc-Hep maturation is required for the application of liver-progenitor-cell-derived hepatocytes. To identify central PKs that potentially promote the maturation of lpc-Hep cells, we profiled the transcriptomes and phosphoproteomes of freshly isolated primary mouse hepatocytes (MHs) and lpc-Hep cells (Fig. 3a).”

5. Fig. 2a: The authors previously reported that SV40-LT was required for proliferation of human iHep. Show the details about the method of iHep induction used in this study. If SV40-LT was used, the authors should carefully examine an unexpected effect of SV40-LT in the analysis. If it was not used, iHep cultures may contain many fibroblasts, and thus cell heterogeneity is a problem for analysis. The authors should isolate iHep before the analysis.

Response: We apologize for the lack of clarity on this point. SV40-LT is required for long-term proliferation of hiHep cells. However, in this study, SV40-LT was not used to avoid unexpected effects of SV40-LT. Without SV40-LT, hiHep cells are capable of division for a limited number of times. We agree with the reviewer that the hiHep cell culture contained many fibroblasts. Although we have tried to purify hiHep cells for several years, we still do not have an efficient method to isolate hiHep cells; we do not have a workable cell surface marker protein that can be used to isolate hiHep cells by FACS sorting, and there is also no workable reporter system that can effectively mark hiHep cells. However, the experimental results supporting our conclusions in the manuscript cannot be explained by fibroblast contamination.

6. *Fig. 2d-f: Do these data indicate an increase or decrease of ALB expression level in each cell, an increase or decrease of reprogramming efficiency, or an increase or decrease of proliferation of ALB-positive cells? The authors need to answer this question with suitable quantitative data.*

Response: We appreciate the reviewer raising this question. In flow cytometry experiments on hiHep cells with ALBUMIN immunofluorescence staining (Supplementary Fig. 5), we observed an increase in ALBUMIN⁺ cells. However, we did not observe an obvious increase in the fluorescence strength of ALBUMIN⁺ cells. Thus, the data more likely indicate an increase in ALB⁺ cell number, rather than an increase of ALB expression in each ALB⁺ cell.

By co-staining ALBUMIN and EdU in hiHep cells treated with EdU for 24 hours, we observed an increase of EdU⁺ in ALB⁺ hiHep cells overexpressing PIM1 (Supplementary Fig. 6c), suggesting increased proliferation capability of hiHep cells in the context of PIM1 overexpression. In addition, we also observed decreased cell death in PIM1-overexpressing HDFs undergoing hepatic reprogramming (Supplementary Fig. 7d,e). Thus, the increased proliferation and decreased cell death of ALB⁺ cells resulted in the increased reprogramming efficiency.

7. *Fig. 2d-f: In addition to ALB expression, other hepatic marker expression and hepatic function should be examined in the screening. Only ALB expression is not sufficient for proper assessment.*

Response: We agree that ALB expression alone is not sufficient for proper assessment. We therefore quantified the expression levels of AAT and ARG, two hepatic functional genes (Fig. 4j,l, Supplementary Fig. 4l). The new results further support our finding that PIM family proteins promote hepatic reprogramming. We also performed ALBUMIN immunofluorescence staining and validated four PKs that showed statistically significant regulation of ALB expression (Fig. 4k).

8. Fig. 2d: As I mentioned above, an increase of PK gene expression and that of target protein phosphorylation may be less relevant. Thus, the screening should be done by gene knockdown/knockout experiments, but not by an overexpression strategy. In fact, Fig. 2f shows a significant effect of the knockdown of *PLK1*, *PLK4* or *NEK2* on the induction of *ALB* gene expression in fibroblasts.

Response: We greatly appreciate the reviewer's constructive suggestion. We agree that overexpression experiments alone are not sufficient for the screening (Fig.2d, updated Fig. 4j). Thus, we performed shRNA-mediated gene knockdown experiments for the PK screening (Fig. 4l, Supplementary Fig. 4l). We found that knockdown of *PIM1* or *PIM2* significantly suppressed the induction of hepatic-lineage genes during hepatic reprogramming, which supported our finding in the overexpression experiments that *PIM1* and *PIM2* can promote hepatic reprogramming. We also found that knockdown of *PLK1*, *PLK2*, or *NEK2* could also inhibit hepatic reprogramming. However, overexpression of *PLK1*, *PLK2*, or *NEK2* showed no significant effect on the induction of hepatic-lineage genes (Fig. 4j,l). Interestingly, although overexpression of *PLK4* or *ROCK2* did not show obvious effects on hepatic reprogramming, knockdown of *PLK4* or *ROCK2* did significantly promote hepatic reprogramming (Fig. 4j,l).

9. Fig. 2d: In this study, the authors focused on PKs that upregulate *ALB* expression, such as *PIM1* and *PIM2*. However, PKs suppressing *ALB* expression, such as *TRIB2* and *TSSK3*, could also be predicted as the candidates of central PKs in hepatic reprogramming. To evaluate the CKI as a "versatile and comprehensive prediction method for PKs", the authors should also focus on the PKs that suppress *ALB* expression and reveal the roles of these PKs, similar to *PIM1*.

Response: We agree that PKs suppressing *ALB* expression, such as *TRIB2* and *TSSK3*, could also be predicted as candidate central PKs in hepatic reprogramming. Thus, we analyzed the expression of *AAT* and *ARG* in the context of *TRIB2* or *TSSK3*

overexpression. We found that overexpression of *TRIB2* had little effect on *AAT* and *ARG* expression. In contrast, although overexpression of *TSSK3* had little effect on *ARG* expression, it significantly suppressed *AAT* expression (Fig. 4j). We also performed shRNA-mediated gene knockdown of *TRIB2* and *TSSK3*. We found that knockdown of *TRIB2* or *TSSK3* had moderate or little effect on hepatic reprogramming (Fig. 4k). These results suggest that *TRIB2* and *TSSK3* could regulate hepatic reprogramming in a moderate manner. Although most of the candidate central PKs showed more or less effects on the regulation of hepatic-lineage genes, two of the 15 predicted central PKs, *PIM1* and *PIM2*, were validated to regulate all examined three hepatic-lineage genes in both overexpression and knockdown experiments.

10. Fig. 3a: PIM1 gene expression was increased after 5 days post-infection. However, hepatic gene expression was increased after 2 days post-infection (Supplementary Fig. 2b). Thus, it is suggested that PIM1 affects an increase or maintenance of ALB gene expression in iHep, but not involves in an initial event of hepatic reprogramming. The authors should clarify this important point with additional data.

Response: As shown in the quantitative PCR and immunoblotting results (Fig. 5a,b), *PIM1* levels slightly increased 2.25 days after infection of FHH. Thus, induction of hepatic gene expression was accompanied by increased *PIM1* expression levels. Moreover, shRNA-mediated knockdown of *PIM1* significantly suppressed induction of hepatic-lineage genes (Fig. 5c), suggesting that *PIM1* is required for the initial event of hepatic reprogramming. In addition, we performed functional analysis for induced cells with sh*PIM1*. As shown in Fig. 5d, *PIM1* knockdown markedly suppressed glycogen storage and ac-LDL uptake in HDF infected with FHH for 5 days, further supported that *PIM1* is required for the initial event of hepatic reprogramming.

11. Fig. 3c: qPCR data suggest that shPIM1 specifically affects ALB gene expression, but not involves in the hepatic reprogramming. The authors should do more detailed analysis to examine whether PIM1 is critical for hepatic reprogramming or not. qPCR analysis is not enough, but other *in vitro* and *in vivo* functional analysis of induced cells with shPIM1 are required.

Response: We understand the reviewer's concern. As detailed in our answer to the previous point, we performed functional analysis for induced cells with shPIM1. As shown in Fig. 5d, PIM1 knockdown markedly suppressed glycogen storage and ac-LDL uptake in HDF infected with FHH for 5 days. Remarkably, PIM1 knockdown significantly increased cell death of HDF infected with FHH (Supplementary Fig. 7b,c). Thus, there are not enough induced cells with shPIM1 for *in vivo* functional analysis. However, currently results using shPIM1 supported that PIM1 is critical for hepatic reprogramming.

12. Fig. 4b: Why were other pathways not analyzed in detail? The data shown in Fig. 5g indicate that ferroptosis inhibition has only a limited effect on the hepatic reprogramming. The authors should verify whether there are other critical pathways.

Response: We appreciate the reviewer's question. As shown in Fig. 4b (updated Fig. 6f), 17 pathways were significantly changed in PIM1 overexpressors. Fifteen were liver-enriched metabolic or functional pathways (e.g., the PPAR signaling pathway, drug metabolism, pentose and glucuronate interconversions, and other similar pathways). Because we found that PIM1 promotes hepatic reprogramming, it is not surprising that pathways related to liver functions were enriched. Among the 17 pathways, we identified two non-metabolic pathways related to cell cycle progression and ferroptosis. Thus, we focused on these two pathways.

We agree that ferroptosis inhibition partially contributes to the effects of PIM1. In Fig. 6g-m, we show that cell cycle also contributes to the effects of PIM1. Based on

our results, we cannot exclude that any other pathways are related to the effects of PIM1.

13. Fig. 4d, e: PIM1 is known as a cell growth inducer. So, it is not surprising that EdU-positive cells increased in fibroblasts overexpressing PIM1 with FHH after 3 days post-infection. Importantly, the endogenous gene expression of PIM1 increased after 5 days post-infection (Fig. 3a). Thus, to reveal the role of PIM1 in the hepatic reprogramming, the authors should do gene knockdown/knockout experiments of PIM1 after 5 days post-infection and examine whether endogenous PIM1 is critical for cell proliferation during reprogramming.

Response: In the revised manuscript, we performed additional experiments to explore the role of PIM1 in hepatic reprogramming. We performed a knockdown experiment with PIM1 shRNA, and found that PIM1 knockdown resulted in increased cell death during hepatic reprogramming (Supplementary Fig. 7b,c). Overexpression of PIM1 suppressed hepatic reprogramming-induced cell death (Supplementary Fig. 7d,e). We also attempted to knockdown PIM1 5 days post-infection. However, this resulted in robust cell death. Thus, we concluded that PIM1 is critical for cell survival during reprogramming.

14. Fig. 4f: The authors should examine whether these signaling pathways are activated with an upregulation of endogenous PIM1 expression and inhibited by PIM1 knockdown/knockout during hepatic reprogramming.

Response: We analyzed the signaling pathways shown in Fig. 4f (updated Fig. 6m) during hepatic reprogramming. We also observed increased phosphorylation of 4D-BP1 and MYC along with upregulation of endogenous PIM1 expression and decreased phosphorylation of MYC in PIM1 knockdowns (Fig. 6l).

15. *Fig. 5d: In addition to qPCR data, the authors should show the images and the percentages of dead cells in shPIM1- or control shRNA-expressing cells after 5 days post-infection to confirm the inhibition of ferroptosis by PIM1 during hepatic reprogramming.*

Response: In light of this suggestion, we have included images showing cell death and quantified the percentages of surviving cells in shPIM1 and shNT cells 5 days post-infection (Supplementary Fig. 7b,c). These results suggest that PIM1 is required for cell survival during hepatic reprogramming.

16. *Fig. 5e: The images and the percentages of dead cells in PIM1- or control GFP-expressing cells should be shown.*

Response: In the revised manuscript we show images of cell death and have quantified the percentages of surviving cells in PIM1- and GFP-overexpressing cells 5 days post-infection (Supplementary Fig. 7d,e). These results further support the function of PIM1 in ferroptosis inhibition during hepatic reprogramming.

17. *Fig. 5f suggests that only ferroptosis occurs in the FHH-induced cell death. To confirm this, the authors should examine whether other types of cell death (apoptosis, necrosis, pyroptosis, necroptosis, etc.) do not occur during hepatic reprogramming. In addition, cell images during ferroptosis should be shown.*

Response: We appreciate the reviewer's constructive comment. As shown in Fig. 5f (updated Fig. 7f), the ferroptosis inhibitors ferrostatin-1 and liproxstatin-1 significantly suppressed FHH-induced cell death. As suggested, we treated cells with Nec-1s (necroptosis inhibitor) or z-VAD (apoptosis and pyroptosis inhibitor), and did not observe obvious effects of these inhibitors on hepatic reprogramming-induced cell death (Supplementary Fig. 7f). In the previous version of the manuscript, we showed images of cell death during hepatic reprogramming in Fig. 5a (updated Fig. 7a).

18. Fig. 5g: Do these data indicate an increase of ALB and CYP3A4 expression levels in each cell, an increase of reprogramming efficiency, or an increase of ALB-positive and CYP3A4-positive cells upon inhibition of ferroptosis? The authors need to answer this question with suitable quantitative data.

Response: To address this question, we performed flow cytometry with HDF+FHH cells. As shown below in Response Fig. 2, we observed an increase in ALBUMIN+ cells after treatment with the ferroptosis inhibitor NAC. However, we did not observe an obvious increase in ALBUMIN immunofluorescence levels in the ALBUMIN+ cells. Thus, it is most probable that the ferroptosis inhibitor promoted reprogramming efficiency or cell survival of ALB+ cells, rather than increasing ALB expression levels in each cell. We did not perform flow cytometry with CYP3A4 staining because none of the CYP3A4 we tested worked sufficiently well for flow cytometry in our hands.

Response Fig. 2 Flow cytometry showing HDF+FHH for ALBUMIN immunofluorescence staining.

19. Fig. 5h, i: Again, to reveal the role of PIM1 during hepatic reprogramming, gene knockdown/knockout experiments of PIM1 are required.

Response: As suggested, we performed gene knockdown experiments to further investigate the role of PIM1 in the generation of NADPH and glutathione during hepatic reprogramming. We found that PIM1 knockdown decreased the NADPH/NADP⁺ ratio and glutathione generation (Supplementary Fig. 7j,k), suggesting that PIM1 is required for the maintenance of the NADPH/NADP⁺ ratio and glutathione levels.

20. *Fig. 6: The authors summarized the present results and argued that PIM1 and other PKs regulate the hepatic reprogramming through inhibition of ferroptosis and cell cycle arrest by phosphorylating with related proteins in concert with transcriptional control of them with FHH and down-stream transcription factors. However, it is almost impossible to extract any useful information from such a diagram because the edges connecting each factor are not visible. Therefore, the data should be replaced or listed together with quantitatively summarized data that reveals the relationship between factors and the degree of overlap of the regulation of transcription and that of phosphorylation.*

Response: We agree with the reviewer's comment; quantitative mRNA expression levels for transcription factors and targets and TMT intensities of p-sites, if available, have been added into two sheets named "TF-target relations" and "ssKSRs" in Supplementary Data 6.

21. *Minor concerns*

The title of this paper should be changed appropriately along with the data obtained in this study.

Response: We changed the title as suggested: "**Trans-omics-based inference of central protein kinases**".

22. *The methods should be improved, because this paper does not go into enough details about the conditions of the experiments.*

Response: In the revision, the descriptions of methods have been significantly improved to present all conditions of the experiments.

23. *“Data not shown” may not be approved.*

Response: This description was misleading and therefore deleted from the manuscript; all experimental data were and are present.

24. *There are same sentences in the text (for example, line 182-186 and line 310-314).*

Response: Duplicated sentences have been removed or revised.

25. *Page 10, Line 205-214: The authors wrote that 5031 unique phosphopeptides were identified through the phosphoproteomics, but they also wrote that 5660 non-redundant p-sites were detected, so why is the number increased?*

Response: The number of non-redundant phosphopeptides is lower than the number of non-redundant p-sites because one phosphopeptide may contain multiple p-sites. We re-checked the analysis and confirmed that these numbers are correct.

26. *Page 10, Line 210-214: The ratio of detected p-sites was calculated for each modified amino acid. How does this ratio compare to the genome-wide figure for phosphopeptides?*

Response: In the revised Methods, we added this important information in the section “Public phosphorylation data resources” as indicated below:

On page 43, paragraph 1, we changed the portions shown in red:

“...After the redundancy clearance, we obtained 244,034 known p-sites including 144,116 pS (59.06%), 61,231 pT (25.09%), and 38,687 pY (15.85%) residues in 18,773 human phosphoproteins, and 119,328 p-sites including 85,774 pS (71.88%), 23,594 pT (19.77%), and 9960 pY (8.35%) in 14,044 mouse phosphoproteins.”

In public databases, the ratio of pY sites is much higher because many pY sites were experimentally identified using pan-specific pY antibodies for phosphopeptide enrichment. Such a procedure enables the high-throughput identification of pY sites by LC-MS/MS. In this study, we used the routine IMAC-based method for phosphopeptide enrichment, which mainly enriches pS/pT phosphopeptides. Thus, the ratio of pY sites is lower in our study, as well as the phosphoproteomic profiling of DOX or genistein resistance. We did not use pan-specific pY antibodies in this work.

27. *Page 14, Line 298-300: Show the data used for selecting 18 genes in a supplementary figure.*

Response: The FPKM values and changes of the 18 selected genes are shown in two sheets labeled “FPKM” and “DEMs” in Supplementary Data 5.

28. *Fig. 1c, 1e, and 2b and Suppl. Fig. 1g: There are representative PKs shown in the Venn diagrams with the numbers of candidates, but the numbers do not match. So, the name of PKs should be omitted or shown outside the Venn diagrams.*

Response: We agree with the reviewer and have removed the gene names of PKs from all Venn diagrams, including Fig. 3n for mouse hepatocyte maturation, Fig. 4h for human hepatic reprogramming, Supplementary Fig. 1a for DOX resistance, and Supplementary Fig. 1c for genistein resistance.

29. *Fig. 4a: To create a list of liver- or fetal liver-enriched genes, curated gene lists from sophisticated databases such as MSigDB or KEGG should be used. At least, the data obtained by analyzing the curated gene lists should be included in supplemental data.*

Response: We appreciate the reviewer's constructive suggestion. We have created a list of liver- or fetal liver-enriched genes from MSigDB and revised Fig. 4a (updated Fig. 6e) accordingly.

30. *Fig. 5h, i: Cells expressing only GFP or PIM1 should also be analyzed additionally as controls.*

Response: As suggested, we analyzed the NADPH/NADP⁺ ratio and total glutathione in HDFs expressing only GFP or PIM1. As shown in Supplementary Fig. 7h,i, overexpression of PIM1 increased the NADPH/NADP⁺ ratio and total glutathione levels.

31. *Suppl. Fig. 1a-f, 1h, 1j-l, and 3d: These data should be summarized in a Table.*

Response: The primary data for the original Supplementary Fig. 1a-f are now presented in Supplementary Data 1; the primary data for the original Supplementary Fig. 1h and 1j-l are now in Supplementary Data 2; and the primary data for the original Supplementary Fig. 3d are shown in Supplementary Data 3.

32. *Suppl. Fig. 3b and 4b: Resolution of these figures is too low to read information. They should be replaced.*

Response: The two images have been replaced with higher resolution images.

33. *Errors and lack of unity are found throughout the paper. The paper should be carefully rewritten.*

Response: In the revision, we significantly improved the presentation by adding additional details about the methodology and comparing CKI to other methods. The primary data for all analyses are provided. The figures were updated and the corresponding figures legends were revised.

Reviewer #4:

This paper makes a methodological contribution and presents a strong validation component. The methodology can enable genome-wide analyses of protein kinase regulations while the biological findings are relevant to address how PKs are regulated to facilitate hepatic reprogramming.

1. Authors mention: "To investigate the most important PKs that regulate hepatic conversion, we integrated the transcriptomic and phosphoproteomic data, and developed a trans-omics-based algorithm, central kinase inference (CKI), that infer central PKs in a defined biological process."

-- Since one of the primary goals is method design, proper parameter choices with sensitivity analysis and logical reasoning behind the terms used is needed. This is the main drawback of this paper. In the following, I will highlight some of these shortcomings.

Response: We agree with the reviewer that all parameter choices should be presented to ensure the reproducibility of the study. In the revision, all of these details have been included. We revised the manuscript as indicated below:

On page 44, paragraph 3, we changed the portions shown in red:

“The CKI algorithm

First, we obtained standard **gene** names and protein sequences of 524 human and 539 mouse PK genes from a previously developed database of iEKPD 2.0 (<http://iekpd.biocuckoo.org/>), which contained 109,912 known and predicted PKs in 164 eukaryotes⁹⁴. For the **human or mouse** transcriptomic data, we **identified potential DEPKs by directly mapping** DEMs of a pairwise comparison to PKs **using gene names**. The $p < 0.01$ was selected as the threshold for DOX resistance, mouse hepatocyte maturation, and hepatic reprogramming, while a more stringent threshold of $p < 0.0001$ was selected for genistein resistance.

For identified p-sites in this study, their upstream regulatory PKs were **first** predicted by **iGPS** (<http://igps.biocuckoo.org/>)²⁰. **All p-sites were prepared in the tab-delimited Phospho.ELM (ELM for short) format⁹¹, including UniProt accession numbers⁸¹, full protein sequences, phosphorylation positions, and residue types**. For each sample, the “Batch Predictor” option in iGPS was used, and the ELM file containing all p-sites was directly loaded for a prediction. The probability of a p-site modified by each PK was individually scored, and only results with scores higher than the pre-determined cut-off values were reserved and outputted. For using iGPS, we selected the default parameters of the “Low threshold” and “Experiment/STRING PPI”.

Then, we designed two approaches to identify **potentially** central PKs from the phosphoproteomic data, including an intensity-based method and a network-based method. For the former, we first defined the total substrate intensity (*TSI*) of a PK by adding square root values (*SRVs*) of TMT intensities of p-sites in their corresponding substrates. For each phosphoproteomic data set, the *TSI* value of a PK *i* in the sample *A* was calculated as below:

$$TSI_A(i) = \sum_{j=1}^n SRV_j$$

Here, *n* is the number of substrate p-sites of the PK *i*. Thus, the *TSI* score of *m* PKs in the sample *A* was computed as below:

$$TSI_A = \sum_{i=1}^m TSI_A(i)$$

We performed the Yate's chi-squared test of a pairwise comparison, by calculating a Chi-squared 2×2 contingency table:

	Sample A	Sample B	Total
PK i	$a = TSI_A(i)$	$b = TSI_B(i)$	$T_i = a + b$
Other PKs	$c = TSI_A - TSI_A(i)$	$d = TSI_B - TSI_B(i)$	$T_o = c + d$
	$T_A = a + c$	$T_B = b + d$	$T = a + b + c + d$

The χ^2 was determined as below:

$$\chi^2 = \frac{T(\max(0, |ad - bc| - T/2))^2}{T_i T_o T_A T_B}$$

The function `chisqprob(degree, χ^2)` of Perl module `Statistics::Distributions` was adopted to calculate p values, whereas the degree was set as 1. In this method, the $p < 1.0 \times 10^{-5}$ was adopted as the threshold to predict potentially central PKs for DOX resistance, mouse hepatocyte maturation, and hepatic reprogramming, while a relaxed threshold of $p < 0.05$ was selected for genistein resistance.

In the network-based method, we defined the kinase network index (*KNI*) to denote the network state of a PK. First, we re-constructed a site-specific PK-substrate network from predicted ssKSRs mutually quantified in a pair of samples, e.g., treatment vs. control. In the network, the nodes denoted individual p-sites or their upstream regulatory PKs, and the orientations of edges were defined as PKs \rightarrow p-sites. Then, single PK networks were individually retrieved to only contain the PK and its substrate p-sites.

To compare sample A and B, the weight value of the network edge, the relative intensity ratio (*RIR*) of a p-site in A vs. B was calculated as below:

$$RIR_{p\text{-site}} = \frac{TMT\ intensity_A}{TMT\ intensity_B}$$

Based on *RIR* values of p-sites, we split each single PK network into an up-regulated sub-network ($RIR > 1$) and a down-regulated sub-network ($RIR < 1$). For a PK i , its up-regulated $KNI_U(i)$ value could be computed by adding *RIRs* of m p-sites in the up-regulated sub-network, whereas its down-regulated $KNI_D(i)$ score

could be inferred from *RIRs* of n p-sites in the down-regulated sub-network as below:

$$KNI_U(i) = \sum_{j=1}^m RIR_j$$

$$KNI_D(i) = \sum_{k=1}^n 1/RIR_k$$

Then, the total KNI_U and KNI_D of *A* vs. *B* for l PKs were calculated as below:

$$KNI_U = \sum_{i=1}^l KNI_U(i)$$

$$KNI_D = \sum_{i=1}^l KNI_D(i)$$

Again, the Yate's chi-squared test was used to identify potentially up- or down-regulated PKs with the following Chi-square 2×2 contingency table:

	Up-regulated	Down-regulated	Total
PK i	$a = KNI_U(i)$	$b = KNI_D(i)$	$T_i = a + b$
Other PKs	$c = KNI_U - KNI_U(i)$	$d = KNI_D - KNI_D(i)$	$T_o = c + d$
	$T_U = a + c$	$T_D = b + d$	$T = a + b + c + d$

The χ^2 was computed as below:

$$\chi^2 = \frac{T(\max(0, |ad - bc| - T/2))^2}{T_i T_o T_U T_D}$$

The p values were also computed with the function `chisqprob(degree, χ^2)` of perl module `Statistics::Distributions`, and the degree was set as 1. In this method, the $p < 1.0 \times 10^{-5}$ was adopted as the threshold to predict potentially central PKs for DOX resistance and mouse hepatocyte maturation, while a relaxed threshold of $p < 0.01$ was selected for genistein resistance and hepatic programming.

Finally, the number of positive hits was counted for each PK from all pairwise comparisons. For DOX resistance^{14,15}, the threshold in CKI for prioritization of potentially central PKs were set as ≥ 14 of the 19 pairwise comparisons (1 for mRNA expression, 3×3 for substrate p-site intensity, and 3×3 for kinase-substrate network). For genistein resistance^{16,17}, the threshold was ≥ 6 of the 12 pairwise comparisons (2×2 for mRNA expression, 2×2 for substrate p-site intensity, and 2×2 for kinase-substrate network). For hepatocyte maturation, the threshold was ≥ 15 of the 27 pairwise comparisons (3×3 for mRNA expression, 3×3 for substrate p-site intensity, and 3×3 for kinase-substrate network). For hepatic programming, the

threshold was ≥ 5 of the 9 pairwise comparisons (3 for mRNA expression, 3 for substrate p-site intensity, and 3 for kinase-substrate network).”

2. 1) Several of the parameter choices need justification, such as (this is not a comprehensive list):

(i) spectral count of 2.89

Response: This number is not a parameter choice. It was calculated from the analysis of our phosphoproteomic data on hepatic reprogramming. We have confirmed that this number is correct.

(ii) class I ($LP > 0.75$, 86.60%), 674 class II ($0.5 < LP \leq 0.75$, 11.91%) and 141 class III ($0.25 \leq LP \leq 0.5$, 2.49%) p-sites

Response: We performed classification of p-sites based on the LP score as described in a previously published study (Humphrey et al., *Cell Metab*, 2013, 17, 1009-20, PMID: 23684622).

(iii) factor of 2 (≥ 2 -fold or ≤ 0.5 -fold change) was employed to pairwise compare phosphoproteomes

Response: A 2-fold change is a widely used threshold to identify differentially regulated p-sites for a two sample comparison without biological or technical replicates.

(iv) The minimum peptide was set as 7, and the mass tolerance for fragment ions was set as 0.02 Da. FDRs of PSM and protein were set to $< 1\%$ and minimum score for modified peptides was set to > 40 .

Response: These parameters are the default configurations for MS/MS identification. We did not change these routine parameters.

(v) To identify DRPs from the phosphoproteomes, a factor of ≥ 2 -fold change was employed for each pair of comparison.

Response: For a two-sample comparison, a 2-fold change is a commonly used threshold to identify DRPs. This parameter choice did not influence the predictions of CKI.

3. To ensure data quality, 15 PKs that were simultaneously detected in ≥ 5 comparisons were reserved as highly potential central PKs (Fig. 2b,c and Supplementary Table 4). Furthermore, for the 15 predicted central PKs, their corresponding p -values calculated in changes of mRNA expression, substrate p -site intensity and network state were shown, respectively (Fig. 2c).

-- how does frequency of detection align with p -values? Is there a one-to-one correlation?

Response: In the revision, the p -values are provided for all comparisons of transcriptomic or phosphoproteomic data of the 15 PKs in Supplementary Data 4.

4. The Methods in general need more explanation. The bioinformatics procedures are not novel and standard. However, the CKI algorithm needs a lot more details.

Response: We are thankful for the reviewer's suggestion. As noted by the reviewer, our routine analyses were not novel, and standard bioinformatic methods were used. Our CKI algorithm is a new method, and we carefully described the basic hypothesis and implementation of each of the three methods in CKI as indicated below:

On page 6, paragraph 1, we changed the portions shown in red:

“A trans-omics-based method for prediction of central PKs

To infer potentially central PKs in defined biological processes, we developed CKI, a trans-omics-based computational method to analyze and integrate transcriptomic data derived from RNA sequencing (RNA-seq) and phosphoproteomic data quantified by tandem mass tag (TMT) labeling coupled with liquid chromatography-tandem mass spectrometry (LC-MS/MS) (Fig. 1). The basic hypothesis behind CKI was that molecular changes at both the transcriptomic and phosphoproteomic levels might be informative in predicting the functional importance of PKs. Comparisons were made of the transcriptomes and phosphoproteomes in paired samples (control vs. treated), taking three types of changes into consideration to synergistically predict central PKs in response to treatment: mRNA expression, substrate p-site intensity, and kinase-substrate network (Fig. 1), as described below.

PK expression at the transcriptional level is essential for regulating its constitutive activity^{18,19}. Thus, PKs with differentially expressed mRNAs (DEMs) in response to treatment may be involved in orchestrating downstream signaling. In this study, differentially expressed PKs (DEPKs) were directly identified from transcriptomic data (Fig. 1). From the phosphoproteomic data of each sample, directed relations of PKs with p-sites, referred to as site-specific kinase-substrate relations (ssKSRs), were predicted using a previously developed software package called *in vivo* Group-based Prediction System (iGPS). This program integrates sequence and protein-protein interaction (PPI) information for predicting p-sites specifically modified by 408 human and 416 mouse PKs²⁰. Then, we hypothesized that a PK with higher activity might phosphorylate more p-sites with higher modification levels, and *vice versa*. We therefore developed an intensity-based approach to identify potentially central PKs based on differential intensity of substrate p-sites between paired samples, e.g., treatment vs. control (Fig. 1).

An alternative hypothesis was that a PK with higher activity may produce a more positive impact on its regulatory phosphorylation network, and *vice versa*. We therefore developed a network-based method to measure the network change for each PK. From predicted ssKSRs mutually quantified in a pair of samples, a kinase-substrate phosphorylation network was re-constructed with directed relations of PKs and p-sites. For each PK, its downstream regulatory network was further split

into an up-regulated sub-network (up-regulated substrate p-sites) and a down-regulated sub-network (down-regulated substrate p-sites). A statistical test was performed to identify potentially central PKs statistically associated with up- or down-regulated sub-network modules (Fig. 1). From all pairwise comparisons, the number of positive hits were counted for each PK as the only measure to prioritize the final candidate PKs.”

5. (i) *It is not clear how the TF-target network was constructed; similarly how was the kinase-substrate network was constructed? How were the two networks integrated considering different edge weights in each network?*

Response: In the revision, we present additional detail about the construction of kinase-substrate networks in the section “A trans-omics-based method for prediction of central PKs” of the Results. We have also included details on the construction of the TF-target network in the section “Trans-omic modeling of the hepatic transdifferentiation-associated TPCW” in the Methods. For simplicity, the different edge weights were not considered for network integration. We revised the manuscript as indicated below:

On page 53, paragraph 2, we added:

“...In the TF-target network, the nodes denoted TFs or their downstream targets, and the orientations of the edges were defined as TF -> target. In the kinase-substrate network, the nodes denoted PKs or their downstream substrate proteins, and the orientations of the edges were defined as PK -> substrate protein. In the final TPCW, the edge weight was not considered...”

6. (ii) *The authors say: "Then, we found the upstream or downstream regions of 24 TF genes contained at least one predicted TFBS of FHH." Firstly, the grammar is incorrect; next this choice (at least one predicted TFBS) seems arbitrary. For each*

such arbitrary choice a sensitivity analysis is needed to show how varying this threshold changes the results.

Response: The original description was accurate, because one important transcription factor, *MAFF*, was predicted to have only one TFBS of FHH. If this threshold was changed, *MAFF* would not be included. We clarified this point by revising the manuscript as indicated below:

On page 21, paragraph 2, we changed the portions shown in red:

“...We counted predicted transcription factor binding sites (TFBSs) in the upstream and downstream regions of the 24 TFs; one TF, *MAFF*, had only one predicted TFBS (Supplementary Fig. 8a). Of the 24 TFs, *MAFF* and *HLF* have previously been reported to contribute to hepatic lineage determination in a massively parallel protein activity assay⁵¹. *SOX17* is also a critical TF for specification of the endodermal lineage^{52,53}...”

7. (iii) *For identifying central PKs, the intensity method was applied. However, many of the terms introduced lack the reasoning. For example, why is TSI a simple sum of square roots of TMT intensities? Is there a logical justification for this?*

Response: TSI is the simple sum of square roots of TMT intensities, a selection we made based not only on logical justification but from exhaustive testing. We tested a number of methods that looked to be more reasonable in theory, but these methods were not ultimately used because they resulted in lower accuracy.

8. *For the network based method, how was the site-specific PK-substrate network created? How is RIR calculated? For $KNI_U(i)$, why is a linear sum of RIRs sufficient? For $KNI_D(i)$, the expression is logically the harmonic mean of RIRs; is this correct and what is the justification?*

Response: The descriptions of the network-based method have been greatly improved in the revision; please see the response to Q1. We made the parameter choices not merely from logical justification, but from exhaustive parameter testing during the development of CKI. A logically reasonable method or parameter choice will be useless if it cannot reach a superior accuracy compared to other methods. The current version the model is useful and accurate, and helped us to identify new central PKs in mouse hepatocyte maturation and human hepatic reprogramming.

9. *Minor:*

There are several typos and grammatical errors. I am highlighting some of these, but there can be more; so please proof-read properly.

Response: We have carefully revised the manuscript to correct all spelling and grammatical errors.

10. *Since both doxorubicin and genistein are potent DNA damage inducers²²⁻²⁵, it was not surprised that the major DNA damage response*
-- surprised  surprising

Response: The corresponding description was changed as shown in red: "...it was not **surprising** that the major DNA damage response...".

11. *It could be found that different substrates were preferentially to be mutually phosphorylated by two distinct clusters of central PKs (Supplementary Fig. 3m).*
-- preferentially  preferred

Response: The original phrasing was imprecise and was removed from the revision.

12. *PIM1 has been found to promotes cell proliferation via various....*
-- promotes  promote

Response: We changed the description as shown in red: "...PIM1 has been found to promote cell proliferation...".

13. The MS/MS spectra files of of freshly isolated mouse MH

-- remove extra "of"

Response: We changed this phrase to: "...The MS/MS spectra files of freshly isolated mouse MH and liver progenitor...".

REVIEWER COMMENTS

Reviewer #1 (Remarks to the Author):

The authors have adequately addressed my concerns. The revised manuscript describes an important method with application in broad areas of biology.

Reviewer #2 (Remarks to the Author):

First of all, I would like to acknowledge the detailed response to all three reviewers' previous comments. The authors undertook a major revision of the manuscript, which overall benefited greatly from these efforts. Most comments were addressed and sequencing and MS raw data were now shared, allowing to review them. Although the manuscript underwent revision already, I see this submission as a new first submission, because raw data review has not been possible before.

That being said, I think interpretation of the MS data recorded in this study still suffers from a major problem: As changes in underlying protein levels were not reported here, it is hardly possible to draw conclusions from measured differences in quantified phosphorylation sites alone. To give one example extracted from the authors' own data: in the hiHep dataset (11 MS runs), because enrichment of phosphorylated peptides was only 42.3%, the majority of peptides allow quantification of several thousand proteins via non-phosphorylated sites in parallel to phosphosite quantification. In the manuscript, non-phosphorylated peptides from these data were not analyzed, I performed my own analysis. One example for a protein with many changed phosphorylation sites in this dataset is AKAP12, of which the authors report 45 sites with large changes (supplementary data table). The level of quantified, non-phosphorylated AKAP12 largely behaves in parallel to the overall change in its phosphorylation. The protein level of AKAP12 is different in the samples analyzed and this is the most important reason for the observed difference in phosphorylation site quantification instead of differential activity of protein kinases. The other MS dataset recorded in this study (MH/lpcHep set, 4 MS runs), has the same fundamental problem: only phosphosites were quantified, not underlying proteins. This problem applies to all phosphorylated proteins analyzed; stating missing protein expression data as a limitation of the CKI (line 555) and quantitative proteomic data possibly improving CKI (line 559) does not suffice here, I am afraid. I think it is an important question if specific changes in phosphorylation that are independent of changes in protein level are better suited for CKI than all changes in phosphosite levels with likely most just reflecting changes in protein level. In other words, measuring the underlying protein level is an important control for the phosphorylation changes and unfortunately this is missing here so far.

TMT labeling efficiencies of both sets were 91.8% (MH) and 93.7% (hiHep) and peptides especially in the hiHep data had a very high overall missed cleavage rate of 34%, with phosphorylated peptides having a missed trypsin cleavage rate of 47.5%, which at some point becomes problematic, even if stated otherwise in the authors' reply. The hiHep dataset does not contain any replicates of the experimental conditions, which I would like to suggest in addition to possible quality control of digests and phospho enrichment during MS sample preparation when the experiment is repeated.

The authors report CILK1 T157 and Y159 phosphorylation sites in the supplementary data table and supplementary figure 2g from the MH/lpcHep set with identical values for both, whereas in the text (line 238) only T157 is referred to. I was not able to obtain a peptide-spectrum match for either T157 or Y159 phosphorylated peptides derived from CILK1 from the raw data. Although I used a different search engine than the authors, this needs to be clarified, even if it requires manual inspection of the peptide-mapped spectrum, because this phospho-peptide is explicitly discussed in the manuscript twice (lines 238 and 552).

Some minor points:

Line 239/240 States that Prkaa1 was undetectable in lpc-Hep and MH cells, whereas supplementary Fig. 2g and the associated data table shows phosphorylated peptides of Prkaa1 from these cells. This point is still not clear to me even if clarified already, sorry for that.

I previously requested "a metadata table that allows to map instrument raw data files to specific experiments". Please provide such a table with the MS raw data on iProX, stating the filenames and which experiment they belong to and which experimental condition is in which TMT channel and so on. This table makes it less tedious to reanalyze the data and should always accompany deposited MS data; the authors use published datasets themselves in this study.

L32 "higher than other methods" – please specify.

L73 AUC scores – Please explain. What is the meaning of these numbers?

L99 "analyze and integrate"

L133 Please specify statistic test.

L234 Please delete (or specify/rephrase).

L289/290 Please delete (or specify/rephrase).

L779 Different from the numbers I see (please see above). Should be clarified.

L849 Please use actual human reference proteome. 2014 version is outdated.

L853 Please also search for unphosphorylated PSMs.

L857 For TMT quantification, was there any filter applied for minimal TMT signal? If so, please specify.

L857 Was there also an FDR correction on the protein level after the FDR correction on the peptide level? Please clarify.

Method section: What is meant with TMT intensity? TMT raw intensity or corrected for noise/background?

Fig.2i p-sites>p-peptides?

Fig.3i p-sites>p-peptides?

Fig.3m Number of p-sites in this study higher than in any other database? My first thought, presentation in the figure is a little confusing. I suggest to edit to show that the numbers show the overlap with the database reported p-sites.

Fig3i-m Rather supplementary figures, although they look nice.

Fig 3n: Reduce circle opacity for better readability, numbers in black for contrast.

Fig. 4b-h ☒ Supplementary figures rather than main figures.

Fig. 6d: Scale from 1-0.99?

Fig 6: Move all panels just characterizing the dataset(s) with little specific information to supplementary figures.

Fig.7f: Please increase contrast of PI signal in pictures.

Overall: Please reduce number of figures - 8 is too extensive - and end with an illustrative, simple panel supporting the discussion putting findings into biological context.

In summary, I would like to reiterate that this study is interesting, important beyond the field and should be published in Nature Communications. I strongly suggest to repeat the two MS experiments - and possibly the associated RNAseq runs if needed -, so that they include protein level measurements as controls for changes in quantified phosphosites and in case of hiHep data biological replicates. This is doable and I would like to encourage the authors to just get to it. The manuscript improved by a lot already and will be a good paper in the end.

Reviewer #3 (Remarks to the Author):

The authors partially answered the questions from this reviewer. However, there are many critical issues involved in the data and the experimental design of this study and some minor concerns, described below.

Critical issues:

Fig. 1a: There may be a correlation between the levels of protein phosphorylation and a kinase-substrate network. However, I am wondering whether there is a correlation between the expression levels of PK genes and the phosphorylation levels of PK-target proteins. Normally, the phosphorylation levels of PK-target proteins change in the result of activation/inactivation of cell intrinsic/extrinsic signals regardless of the expression levels of PK genes. Thus, this study may not need a trans-omic analysis, but only need a phosphoproteome analysis. Fig. 1c and e actually support this opinion.

Response: In the revision, we carefully analyzed the correlations between mRNA expression levels of PKs and the corresponding substrate p-site intensities. The weak correlations between the transcriptomic data and the phosphoproteomic data indicated that the two data types are relatively independent, and therefore a trans-omic analysis can provide new insights by integrating the information from the two data types to reach a high accuracy. We clarified this point by revising the manuscript as indicated below:

On page 26, paragraph 1, we added:

“...Using all pairwise comparisons of the transcriptomic and phosphoproteomic data for each biological process, we analyzed correlations between PK gene expression and the intensity of the corresponding substrate p-sites based on the predicted ssKSRs. The average Spearman’s correlation coefficients were calculated as 0, -0.02, 0.14, and 0.10 for DOX resistance, genistein resistance, hepatocyte maturation, and hepatic reprogramming, respectively (Supplementary Fig. 9a-d), implying a weak correlation between PK gene expression and phosphorylation of the target p-sites.”

On page 50, paragraph 4, we added:

“Correlation analysis of mRNA expression levels of PKs and their substrate p-site intensities

The correlations between mRNA expression levels of PKs and their corresponding substrate p-site intensities were measured based on the ssKSRs, using the Spearman’s correlation. For each ssKSR, the Spearman’s correlation coefficient (ρ) between the mRNA expression level of the upstream PK and the intensity of the downstream p-site was calculated across different samples or replicates.”

Reviewer’s response: The authors state that the transcription levels of PKs and the phosphorylation levels of PK-target proteins are independent, and therefore, the authors claim that the trans-omic analysis with these data can provide new insights. However, ROC curves show that mRNA expression data contributes little to the prediction of central PKs; on the other hand, network data highly

contributes to the prediction (Fig. 2d and 2k). Thus, it seems that a combined analysis with the p-site intensities and kinase-substrate network data is sufficient for the prediction. In other words, mRNA expression data and trans-omic analysis may not be necessary for predicting the CKI algorithm.

Fig. 1b-e: The authors conducted a simulation using public data to show a predictive performance of the CKI algorithm, and the number of reported PKs among the resulting candidates is shown in the text, which is an important argument to estimate the performance of CKI. However, to attain objective evaluation of the prediction method, the number of false negatives, which were not predicted in the CKI algorithm, should also be summarized and disclosed.

Response: We agree with the reviewer about the importance of including these data. In the revision, we manually curated known PKs associated with DOX or genistein resistance from the scientific literature. These known PKs were used as an independent testing dataset to measure the accuracy of CKI and allow comparison of CKI to other methods. For the testing dataset, we counted the true positive (TP), true negative (TN), false positive (FP), and false negative (FN) values and calculated sensitivity (Sn) and specificity (Sp). ROC curves were generated for each method and the corresponding AUC values calculated for each curve. We have also included confusion matrices and PCA analyses. The original Fig. 1 was split into the new Figs. 1, 2 and 3. Evaluations of CKI using the DOX and genistein resistance testing data are shown in Fig. 2. We added a new section entitled "Evaluation of the accuracy of CKI" in the Results to present these details. The corresponding methodology for performance evaluation has been added in a new Methods section entitled "Performance evaluation". To incorporate these changes, the manuscript has been updated as indicated below:

On page 8, paragraph 1, we changed the portions shown in red:

"Evaluation of the accuracy of CKI

CKI is a model-based method, and no prior data were used for training. To evaluate the performance of CKI, we predicted potentially central PKs using data from two types of previously published drug-resistance studies, DOX resistance and genistein resistance (Fig. 2, Supplementary Fig. 1 and Supplementary Data 1)14-17.

In 2016, Zanotto-Filho et al. profiled the transcriptomic changes of a human breast cancer cell line (MDA-MB231) and osteosarcoma cell line (U-2 OS) using a number of chemotherapeutic agents including DOX (1 μ M/L, 8 h), and uncovered a regulatory role of nuclear factor erythroid 2-related factor 2 (NFE2L2/NRF2) in chemotherapy resistance¹⁴. In another study, Hoglebe et al. quantified the phosphoproteomes of U-2 OS cells with or without DOX treatment (5 μ M/L, 2 h)¹⁵. From these two studies^{14,15}, we identified 2197 DEMs and obtained quantitative data for 27,590 p-sites in 5592 proteins (Fig. 2a, b). In the phosphoproteomic data, there were 22,744 phosphoserine residues (pS, 82.44%), 4451 phosphothreonine residues (pT, 16.13%), and 395 phosphotyrosine residues (pY, 1.43%) (Fig. 2c). From the literature, known PKs involved in DOX resistance were curated (Supplementary Data 1). Using this independent testing dataset, we evaluated the accuracy of CKI by calculating the AUC score of the method that incorporated all three types of data (mRNA expression, substrate p-site intensity, and kinase-substrate network) and of models built with each data type alone (Fig. 2d). We also

compared to a z-score-based method, Kinase-Substrate Enrichment Analysis (KSEA), which predicts potentially important PKs only from phosphoproteomic data^{21,22}. We found that using only data related to changes in substrate p-site intensity or the kinase-substrate network had a comparable accuracy to that of KSEA (AUC = 0.7635, 0.8058, and 0.7871, respectively) (Fig. 2d). However, when the three types of data were integrated with CKI, the AUC value was higher than the other methods (0.8278) (Fig. 2d). The confusion matrix and principal component analysis (PCA) indicated that known and unknown PKs could be separated in a reasonably accurate manner (Fig. 2e, f). Of the final 22 PKs predicted by CKI, 10 had already been reported as truly important for DOX resistance in U-2 OS cells, including ATM/ATR23, MAPK1324, and CHEK125 (Fig. 2g and Supplementary Data 1). In this case, the transcriptomic data were less informative and only contributed to prioritization of one of the 22 final candidate PKs (Fig. 2g and Supplementary Fig. 1a). In addition, we mapped the quantified p-sites to the 22 PKs where available, and found that only one PK activity-associated p-site, ATM S198126, was up-regulated after DOX treatment (Fig. 2g, Supplementary Fig. 1b and Supplementary Data 1).

Next, we integrated the transcriptomic and phosphoproteomic data of breast cancer cells treated with or without genistein from previous reports^{16,17}. In total, we identified 3264 DEMs and obtained quantitative data for 17,403 p-sites in 7149 proteins (Fig. 2h-j). Using known genistein-associated PKs as the testing data, the AUC value of CKI was calculated as 0.7912, showing a much higher accuracy than all of the other methods tested (Fig. 2k). Again, the results from the confusion matrix and PCA supported CKI could distinguish between known and unknown PKs (Fig. 2l, m). In this case, the transcriptomic data also contributed to prioritizing five of the 21 final candidate PKs (Fig. 2n, Supplementary Fig. 1c and Supplementary Data 1). No PK activity-associated p-sites were identified to be differentially regulated after genistein treatment (Supplementary Fig. 1d and Supplementary Data 1).

Because both DOX and genistein are potent DNA damage inducers¹⁴⁻¹⁷, it was not surprising that the major PK known to respond to DNA damage, ATM, was successfully predicted as a central PK in both cases (Fig. 2g, n). Taken together, the results showed a promising performance of CKI in well-studied drug-resistant cancer cells.”

On page 49, paragraph 3, we added:

“Performance evaluation

For the DOX or genistein resistance, known regulatory PKs curated from the literature were taken as the positive data, while other remaining PKs predicted with at least one ssKSR were taken as the negative data. To evaluate the accuracy of CKI and other methods, true positive (TP), true negative (TN), false positive (FP), and false negative (FN) values were counted, and we calculated 2 commonly-used measurements, including sensitivity (S_n) and specificity (S_p) as below:

$$S_n = TP / (TP + FN), S_p = TN / (TN + FP)$$

The receiver operating characteristic (ROC) curves were illustrated based on S_n and $1 - S_p$ values, and the corresponding AUC scores were calculated.

Then, the confusion matrix under a selected threshold of pairwise comparisons was calculated as below:

Actual

Known Unknown

Predicted Known TP/(TP+FN) FN/(TP+FN)

Unknown FP/(FP+TN) TN/(FP+TN)

For each of known or unknown PKs, the number of positive hits and the sum of all minus-log transformed (-lg) p values were counted for PCA analysis.”

Reviewer’s response: In the revised manuscript, the authors presented confusion matrixes to evaluate the predictive performance of the CKI algorithm (Fig. 2e and 2l). The matrixes showed very low false-negative ratios of the CKI algorithm (the scores are 0.03 and 0.04 in Fig. 2e and Fig. 2l, respectively), but simultaneously showed high false-positive ratios of the CKI algorithm (the scores are 0.55 and 0.81 in Fig. 2e and Fig. 2l, respectively). These data indicate that the CKI algorithm permits contamination of a large number of false-positives. Thus, the predictive performance of the CKI algorithm is considered to be inadequate for the prediction of central PKs. In addition, descriptions for calculating scores in the confusion matrixes contain mistakes: for example, FN and FP should be reversed.

Fig. 2a: The authors previously reported that SV40-LT was required for proliferation of human iHep. Show the details about the method of iHep induction used in this study. If SV40-LT was used, the authors should carefully examine an unexpected effect of SV40-LT in the analysis. If it was not used, iHep cultures may contain many fibroblasts, and thus cell heterogeneity is a problem for analysis. The authors should isolate iHep before the analysis.

Response: We apologize for the lack of clarity on this point. SV40-LT is required for long-term proliferation of hiHep cells. However, in this study, SV40-LT was not used to avoid unexpected effects of SV40-LT. Without SV40-LT, hiHep cells are capable of division for a limited number of times. We agree with the reviewer that the hiHep cell culture contained many fibroblasts. Although we have tried to purify hiHep cells for several years, we still do not have an efficient method to isolate hiHep cells; we do not have a workable cell surface marker protein that can be used to isolate hiHep cells by FACS sorting, and there is also no workable reporter system that can effectively mark hiHep cells. However, the experimental results supporting our conclusions in the manuscript cannot be explained by fibroblast contamination.

Reviewer’s response: Because many fibroblasts contaminate hiHep cultures, the data are not reflective of the nature of hiHep. Thus, the results in this study cannot be trusted. hiHep could be isolated using markers of epithelial cells, such as E-cadherin, because hiHep are epithelial cells but fibroblasts are mesenchymal cells.

Fig. 2d-f: Do these data indicate an increase or decrease of ALB expression level in each cell, an increase or decrease of reprogramming efficiency, or an increase or decrease of proliferation of ALB-positive cells? The authors need to answer this question with suitable quantitative data.

Response: We appreciate the reviewer raising this question. In flow cytometry experiments on hiHep cells with ALBUMIN immunofluorescence staining (Supplementary Fig. 5), we observed an increase in ALBUMIN+ cells. However, we did not observe an obvious increase in the fluorescence strength of ALBUMIN+ cells. Thus, the data more likely indicate an increase in ALB+ cell number, rather than an increase of ALB expression in each ALB+ cell.

By co-staining ALBUMIN and EdU in hiHep cells treated with EdU for 24 hours, we observed an increase of EdU+ in ALB+ hiHep cells overexpressing PIM1 (Supplementary Fig. 6c), suggesting increased proliferation capability of hiHep cells in the context of PIM1 overexpression. In addition, we also observed decreased cell death in PIM1-overexpressing HDFs undergoing hepatic reprogramming (Supplementary Fig. 7d,e). Thus, the increased proliferation and decreased cell death of ALB+ cells resulted in the increased reprogramming efficiency.

Reviewer's response: Additional data are helpful, but not showing an increase of the reprogramming efficiency. The authors should describe the results correctly: the data suggest that PIM1 overexpression activate proliferation of hiHep and decrease cell death of HDFs.

Fig. 2d-f: In addition to ALB expression, other hepatic marker expression and hepatic function should be examined in the screening. Only ALB expression is not sufficient for proper assessment.

Response: We agree that ALB expression alone is not sufficient for proper assessment. We therefore quantified the expression levels of AAT and ARG, two hepatic functional genes (Fig. 4j,l, Supplementary Fig. 4l). The new results further support our finding that PIM family proteins promote hepatic reprogramming. We also performed ALBUMIN immunofluorescence staining and validated four PKs that showed statistically significant regulation of ALB expression (Fig. 4k).

Reviewer's response: Hepatic function was not analyzed, and there is no statistical data in fig 4k.

Fig. 2d: As I mentioned above, an increase of PK gene expression and that of target protein phosphorylation may be less relevant. Thus, the screening should be done by gene knockdown/knockout experiments, but not by an overexpression strategy. In fact, Fig. 2f shows a significant effect of the knockdown of PLK1, PLK4 or NEK2 on the induction of ALB gene expression in fibroblasts.

Response: We greatly appreciate the reviewer's constructive suggestion. We agree that overexpression experiments alone are not sufficient for the screening (Fig.2d, updated Fig. 4j). Thus, we performed shRNA-mediated gene knockdown experiments for the PK screening (Fig. 4l, Supplementary Fig. 4l). We

found that knockdown of PIM1 or PIM2 significantly suppressed the induction of hepatic-lineage genes during hepatic reprogramming, which supported our finding in the overexpression experiments that PIM1 and PIM2 can promote hepatic reprogramming. We also found that knockdown of PLK1, PLK2, or NEK2 could also inhibit hepatic reprogramming. However, overexpression of PLK1, PLK2, or NEK2 showed no significant effect on the induction of hepatic-lineage genes (Fig. 4j,l). Interestingly, although overexpression of PLK4 or ROCK2 did not show obvious effects on hepatic reprogramming, knockdown of PLK4 or ROCK2 did significantly promote hepatic reprogramming (Fig. 4j,l).

Reviewer's response: Additional data indicate that the results of gene overexpression experiments in this study are unreliable. As this reviewer mentioned previously, an increase of PK gene expression and that of target protein phosphorylation may be less relevant. Thus, the screening should be done by gene knockdown/knockout experiments, but not by an overexpression strategy. This reviewer cannot understand why the authors did not evaluate the function of PLK1, PLK2, and NEK2.

Fig. 2d: In this study, the authors focused on PKs that upregulate ALB expression, such as PIM1 and PIM2. However, PKs suppressing ALB expression, such as TRIB2 and TSSK3, could also be predicted as the candidates of central PKs in hepatic reprogramming. To evaluate the CKI as a "versatile and comprehensive prediction method for PKs", the authors should also focus on the PKs that suppress ALB expression and reveal the roles of these PKs, similar to PIM1.

Response: We agree that PKs suppressing ALB expression, such as TRIB2 and TSSK3, could also be predicted as candidate central PKs in hepatic reprogramming. Thus, we analyzed the expression of AAT and ARG in the context of TRIB2 or TSSK3 overexpression. We found that overexpression of TRIB2 had little effect on AAT and ARG expression. In contrast, although overexpression of TSSK3 had little effect on ARG expression, it significantly suppressed AAT expression (Fig. 4j). We also performed shRNA-mediated gene knockdown of TRIB2 and TSSK3. We found that knockdown of TRIB2 or TSSK3 had moderate or little effect on hepatic reprogramming (Fig. 4k). These results suggest that TRIB2 and TSSK3 could regulate hepatic reprogramming in a moderate manner. Although most of the candidate central PKs showed more or less effects on the regulation of hepatic-lineage genes, two of the 15 predicted central PKs, PIM1 and PIM2, were validated to regulate all examined three hepatic-lineage genes in both overexpression and knockdown experiments.

Reviewer's response: The authors added AAT and ARG expression data under overexpression conditions of candidate PKs (Fig. 4j) and ALB, AAT, and ARG expression data under knockdown conditions of the PKs (Fig. 4l). Overexpression of PIM1 and PIM2 changed the expression of all three hepatic genes, while the knockdown experiments revealed that five PKs (PIM1, PIM2, PLK1, PLK2, and NEK2) also changed the expression of all three hepatic genes. Based on these data, this reviewer cannot understand why the authors did not evaluate the function of PLK1, PLK2, and NEK2. As this reviewer mentioned previously, an increase of PK gene expression and that of target protein phosphorylation may be less relevant. Thus,

the screening should be done by gene knockdown/knockout experiments, but not by an overexpression strategy.

Fig. 5g: Do these data indicate an increase of ALB and CYP3A4 expression levels in each cell, an increase of reprogramming efficiency, or an increase of ALB-positive and CYP3A4-positive cells upon inhibition of ferroptosis? The authors need to answer this question with suitable quantitative data.

Response: To address this question, we performed flow cytometry with HDF+FHH cells. As shown below in Response Fig. 2, we observed an increase in ALBUMIN+ cells after treatment with the ferroptosis inhibitor NAC. However, we did not observe an obvious increase in ALBUMIN immunofluorescence levels in the ALBUMIN+ cells. Thus, it is most probable that the ferroptosis inhibitor promoted reprogramming efficiency or cell survival of ALB+ cells, rather than increasing ALB expression levels in each cell. We did not perform flow cytometry with CYP3A4 staining because none of the CYP3A4 we tested worked sufficiently well for flow cytometry in our hands.

Response Fig. 2 Flow cytometry showing HDF+FHH for ALBUMIN immunofluorescence staining.

Reviewer's response: Considering that PIM1 and ferroptosis inhibitors increase the reprogramming efficiency is simplistic. In the present state of the work, the authors should consider that PIM1 and ferroptosis inhibitors support proliferation of hiHep and decrease cell death of HDFs.

The title of this paper should be changed appropriately along with the data obtained in this study.

Response: We changed the title as suggested: "Trans-omics-based inference of central protein kinases".

Reviewer's response: The new title is inconcrete and overstatement. The findings in this study should be reflected correctly to the title. In the abstract, the authors mentioned that "PIM1 promotes hepatic conversion and protects HDFs from reprogramming-induced ferroptosis and cell cycle arrest" as the conclusion of this study. This sentence may be more suitable as the title than the present one.

Minor concerns:

Fig. 1d: What are hepatic progenitor cells used here? Show the details about those cells, and also show what characterizes them as hepatic progenitor cells.

Response: We regret the omission of this important information in the original manuscript. We generated proliferative bipotent hepatic progenitor cells from primary mouse hepatocytes following a well-established dedifferentiation protocol (Katsuda et al., Cell Stem Cell, 2017). The liver progenitor cells are capable of differentiation into both biliary epithelial cells (lpc-BEC) and hepatocytes (lpc-Hep, Fig. 3b-e). The lpc-BEC cells expressed bile duct marker genes, including CK19, Cftr, Aqp1 and Ae2 (Fig. 3e), and can form ductal structures (Fig. 3b,c). The lpc-Hep cells were ALBUMIN positive (Fig. 3d) and expressed the hepatocyte marker genes Aat and Alb (Fig. 3e).

To clarify these points, we revised the manuscript as indicated below:

On page 10, paragraph 3, we changed the portions shown in red:

“To further validate the accuracy of CKI, we analyzed the hepatocyte maturation process in mouse liver progenitor cells (Fig. 3a). We converted mouse hepatocytes to proliferative bipotent liver progenitor cells (lpc) using a previously established protocol²⁷. The lpc cells could then differentiate into either CK19+ biliary epithelial cells (lpc-BECs) that can form ductal structures (Fig. 3b, c and e) or ALBUMIN+ hepatocytes (lpc-Heps, Fig. 3d, e). However, lpc-Heps generated with this differentiation protocol are relatively immature²⁷, meaning that further lpc-Hep maturation is required for the application of liver-progenitor-cell-derived hepatocytes. To identify central PKs that potentially promote the maturation of lpc-Hep cells, we profiled the transcriptomes and phosphoproteomes of freshly isolated primary mouse hepatocytes (MHs) and lpc-Hep cells (Fig. 3a).”

Reviewer’s response: The authors must describe the name of cells used in this study correctly to avoid misunderstanding. Chemically induced liver progenitors (CLiPs) or dedifferentiated hepatocytes should be used here.

Fig. 3a: PIM1 gene expression was increased after 5 days post-infection. However, hepatic gene expression was increased after 2 days post-infection (Supplementary Fig. 2b). Thus, it is suggested that PIM1 affects an increase or maintenance of ALB gene expression in iHep, but not involves in an initial event of hepatic reprogramming. The authors should clarify this important point with additional data.

Response: As shown in the quantitative PCR and immunoblotting results (Fig. 5a,b), PIM1 levels slightly increased 2.25 days after infection of FHH. Thus, induction of hepatic gene expression was accompanied by increased PIM1 expression levels. Moreover, shRNA-mediated knockdown of PIM1 significantly suppressed induction of hepatic-lineage genes (Fig. 5c), suggesting that PIM1 is required for the initial event of hepatic reprogramming. In addition, we performed functional analysis for induced cells with shPIM1. As shown in Fig. 5d, PIM1 knockdown markedly suppressed glycogen storage and ac-LDL uptake in HDF infected with FHH for 5 days, further supported that PIM1 is required for the initial event of hepatic reprogramming.

Reviewer's response: In fact, PIM1 expression is highly upregulated after 5 days post-infection. Also, PIM1 overexpression activated proliferation of hiHep and decreased cell death of HDFs. The authors should discuss the role of PIM1 in the hepatic reprogramming more precisely.

Fig. 4b: Why were other pathways not analyzed in detail? The data shown in Fig. 5g indicate that ferroptosis inhibition has only a limited effect on the hepatic reprogramming. The authors should verify whether there are other critical pathways.

Response: We appreciate the reviewer's question. As shown in Fig. 4b (updated Fig. 6f), 17 pathways were significantly changed in PIM1 overexpressors. Fifteen were liver-enriched metabolic or functional pathways (e.g., the PPAR signaling pathway, drug metabolism, pentose and glucuronate interconversions, and other similar pathways). Because we found that PIM1 promotes hepatic reprogramming, it is not surprising that pathways related to liver functions were enriched. Among the 17 pathways, we identified two non-metabolic pathways related to cell cycle progression and ferroptosis. Thus, we focused on these two pathways.

We agree that ferroptosis inhibition partially contributes to the effects of PIM1. In Fig. 6g-m, we show that cell cycle also contributes to the effects of PIM1. Based on our results, we cannot exclude that any other pathways are related to the effects of PIM1.

Reviewer's response: The authors should carefully describe this in the discussion.

Fig. 4f: The authors should examine whether these signaling pathways are activated with an upregulation of endogenous PIM1 expression and inhibited by PIM1 knockdown/knockout during hepatic reprogramming.

Response: We analyzed the signaling pathways shown in Fig. 4f (updated Fig. 6m) during hepatic reprogramming. We also observed increased phosphorylation of 4D-BP1 and MYC along with upregulation of endogenous PIM1 expression and decreased phosphorylation of MYC in PIM1 knockdowns (Fig. 6l).

Reviewer's response: Where are the data of PIM1 knockdown in fig 6l? Also, western blot data for total 4E-BP1 protein are missing in fig 6l and 6m.

Reviewer #4 (Remarks to the Author):

My concerns were addressed satisfactorily by the authors.

Detailed Responses to Reviewers' Comments

Reviewer #1:

The authors have adequately addressed my concerns. The revised manuscript describes an important method with application in broad areas of biology.

Reviewer #2:

First of all, I would like to acknowledge the detailed response to all three reviewers' previous comments. The authors undertook a major revision of the manuscript, which overall benefited greatly from these efforts. Most comments were addressed and sequencing and MS raw data were now shared, allowing to review them. Although the manuscript underwent revision already, I see this submission as a new first submission, because raw data review has not been possible before.

1. That being said, I think interpretation of the MS data recorded in this study still suffers from a major problem: As changes in underlying protein levels were not reported here, it is hardly possible to draw conclusions from measured differences in quantified phosphorylation sites alone. To give one example extracted from the authors' own data: in the hiHep dataset (11 MS runs), because enrichment of phosphorylated peptides was only 42.3%, the majority of peptides allow quantification of several thousand proteins via non-phosphorylated sites in parallel to phosphosite quantification. In the manuscript, non-phosphorylated peptides from these data were not analyzed, I performed my own analysis. One example for a protein with many changed phosphorylation sites in this dataset is AKAP12, of which the authors report 45 sites with large changes (supplementary data table). The level of quantified, non-phosphorylated AKAP12 largely behaves in parallel to the overall change in its phosphorylation. The protein level of AKAP12 is different in the samples analyzed and this is the most important reason for the observed difference in phosphorylation site quantification instead of differential activity of protein kinases.

Response: We are thankful for this helpful suggestion from the reviewer. In revision, we performed additional proteomic quantifications for mouse hepatocyte maturation and human hepatic reprogramming, respectively. In order to be consistent with our transcriptomic and phosphoproteomic data, three biological replicates were prepared for mouse MH and CLiP-Hep cells, for mouse hepatocyte maturation. For human hepatic reprogramming, protein samples of GFP, FHH-2.25d and FHH-5d were prepared. Then, we carefully analyzed and compared the predictions of CKI, using p-sites with or without normalization by their corresponding protein expression levels. For AKAP12, we in total detected 44 DRPs from merely phosphoproteomic data. After the p-sites were normalized by proteomic data, we identified 46 DRPs, and 43 of these DRPs were covered by our original results that only used phosphoproteomic data. These new results were present by adding a new Supplementary Note 3 entitled

“Additional analyses of the CKI accuracy”, Supplementary 12 and 13. We revised the manuscript as below:

Main text, page 24, paragraph 2, lines 481-490, added,

“In CKI, the p-site intensity was directly used, without considering the potential impact of protein expression. To evaluate the influence of protein expression on the CKI accuracy, we additionally conducted proteomic quantifications for mouse hepatocyte maturation and human hepatic reprogramming. Then, the CKI predictions were compared, using p-sites with or without normalization by their corresponding protein expression level (Supplementary Note 3, Supplementary Fig. 12, 13). From the results, we found that only one validated PK, PIM1, could still be predicted if normalized by proteomic data (Supplementary Fig. 12f, 13f), indicating proteomic normalization didn't increase the CKI accuracy for our study.”

Supplementary Information, page 7, paragraph 2, added,

“Supplementary Note 3. Additional analyses of the CKI accuracy

To evaluate the accuracy of CKI using p-sites with or without normalization by their corresponding protein expression levels, we additionally conducted quantitative proteomic profiling for mouse hepatocyte maturation and human hepatic reprogramming, respectively. For the former, three biological replicates were prepared for immature hepatocytes generated from liver progenitor cells (CLiP-Hep) and mature hepatocytes (MH) isolated from mouse liver, respectively. For the latter, human dermal fibroblasts (HDFs) transduced with GFP for 2.25 days (GFP), or 2.25 days (FHH-2.25d) and 5 days (FHH-5d) after transduction of *FOXA3*, *HNF1A*, and *HNF4A* (FHH). During database search of peptides, proteins were also simultaneously quantified by MaxQuant (v.1.4.1.2)¹² that summed up eXtracted Ion Current (XIC) of all isotopic clusters associated with each mappable protein sequence. Then, the global centering (GC) method¹³ was used to normalize proteomic data for each sample, in which the average intensity value of all proteins was normalized to 1 (Mean=1).

From the proteomic data of mouse hepatocyte maturation, we identified 31,318 peptides and 6250 proteins from 6 samples (Supplementary Fig. 12a). The average spectral count of mappable peptides was 2.19, and 15,586 (49.77%) peptides were matched with ≥ 2 spectral counts (Supplementary Fig. 12b). The average number of peptides for mappable proteins was 5.34, and only 1276 (20.42%) proteins were quantified with one matched peptide (Supplementary Fig. 12c). Two-way hierarchical clustering was performed by calculating the Spearman's correlation coefficient for the proteomic data between pairs of samples, and the results indicated that CLiP-Hep and MH cells could be unambiguously distinguished (Supplementary Fig. 12d). From the proteomic data, 16 of 28 predicted central PKs were not detected at all, and only one newly identified PK, *Prkx*, was significantly up-regulated in CLiP-Hep cells (Supplementary Fig. 12e).

Using the p-sites normalized by their corresponding protein expression levels, we re-analyzed the trans-omic data, and predicted 36 potentially central PKs, in which the three newly identified PKs, *Prkaca*, *Prkacb*, and *Prkx*, were not included

(Supplementary Fig. 12f). In addition, we evaluated the accuracy of CKI with or without transcriptomic data. By excluding transcriptomic data, 31 potentially central PKs were predicted merely from phosphoproteomic data, including all the three newly validated PKs (Supplementary Fig. 12g), indicating that including transcriptomic data or not did not influence the final validations.

For human hepatic reprogramming, we identified 28,651 peptides and 5446 proteins from proteomic data (Supplementary Fig. 13a). The average spectral count of mappable peptides was 2.15, and 13,638 (47.60%) peptides were matched with ≥ 2 spectral counts (Supplementary Fig. 13b). The average number of peptides for mappable proteins was 5.52, and only 976 (17.92%) proteins were quantified with one matched peptide (Supplementary Fig. 13c). To evaluate the potential influence of protein expression on the quantification of p-sites, DRPs were identified and compared, using p-sites with or without normalization by their corresponding protein expression levels. From the results, it could be found that including proteomic data or not did not markedly change the phosphoproteomics-based results. For example, 44 DRPs were identified on A-kinase anchor protein 12 (AKAP12) from phosphoproteomic data, and 43 of these p-sites remained to be DRPs after normalization of p-site intensities by proteomic data (Supplementary Fig. 13d). From the proteomic data, only 3 of 15 predicted central PKs, including MAPK4, ROCK2, and *KALRN*, were detected, and only MAPK4 exhibited a > 2 -fold change at FHH-5d against FHH-2.25d or GFP (Supplementary Fig. 13e).

Again, we re-analyzed the transcriptomic data and phosphoproteomic data using the p-sites normalized by their corresponding protein expression levels, and predicted 19 potentially central PKs involved in hepatic reprogramming. Only one validated PK, PIM1, was included (Supplementary Fig. 13f). By excluding transcriptomic data, 13 potentially central PKs including a validated PK, PIM2, were predicted merely from phosphoproteomic data. However, the most important finding of this study, PIM1, could not be covered any longer (Supplementary Fig. 13g).

Our study on human hepatic reprogramming has been started since 2014, and at that time we used the human proteome set from UniProt (Version 201401)¹⁴ for database search. To test whether the update of UniProt will influence the p-site identification, we downloaded the recently released human reference proteome set from UniProt (Version 202203)¹⁴. It could be found that different versions of reference proteome sets did not significantly change the phosphoproteomic results, and only 305 (5.4%) p-sites identified from UniProt Version 201401 were not covered by UniProt Version 202203 (Supplementary Fig. 13h). Using UniProt Version 202203, we re-analyzed the trans-omic data, and CKI could still recall 13 of 15 previously predicted PKs, including the two validated PKs, PIM1 and PIM2 (Supplementary Fig. 13i).”

2. The other MS dataset recorded in this study (MH/lpcHep set, 4 MS runs), has the same fundamental problem: only phosphosites were quantified, not underlying proteins. This problem applies to all phosphorylated proteins analyzed; stating missing protein expression data as a limitation of the CKI (line 555) and quantitative proteomic data possibly improving CKI (line 559) does not suffice here, I am afraid. I think it is an

important question if specific changes in phosphorylation that are independent of changes in protein level are better suited for CKI than all changes in phosphosite levels with likely most just reflecting changes in protein level. In other words, measuring the underlying protein level is an important control for the phosphorylation changes and unfortunately this is missing here so far.

Response: In revision, we additionally performed quantitative proteomic profiling for mouse hepatocyte maturation and human hepatic reprogramming, respectively. After the p-site intensities were normalized using proteomic data, we found that only one of the 5 validated PKs, PIM1, could still be correctly predicted. This result indicated that using p-sites normalized by proteomic data did not increase the accuracy of CKI for our further validations. However, from the proteomic data, we found that one validated PK, *Prkx*, was significantly up-regulated in CLiP-Hep against MH cells, at the protein level. Thus, this result supported the usefulness of proteomic data. In our future study, we will test the efficiency of the method that integrates the proteomic change, but not the normalization of p-sites using proteomic data. The comparison of CKI predictions with or without proteomic normalization was present in above answer. Here we indicated the usefulness of proteomic data as below:

Main text, page 26, paragraph 2, lines 517-520, added,

“...Indeed, we found that one validated PK, *Prkx*, was significantly up-regulated in CLiP-Hep against MH cells, at the proteomic expression level (Supplementary Fig. 12e, $p < 0.01$), supporting the usefulness of proteomic data....”

3. TMT labeling efficiencies of both sets were 91.8% (MH) and 93.7% (hiHep) and peptides especially in the hiHep data had a very high overall missed cleavage rate of 34%, with phosphorylated peptides having a missed trypsin cleavage rate of 47.5%, which at some point becomes problematic, even if stated otherwise in the authors' reply. The hiHep dataset does not contain any replicates of the experimental conditions, which I would like to suggest in addition to possible quality control of digests and phospho enrichment during MS sample preparation when the experiment is repeated.

Response: Based on your concern, we re-calculated the TMT labeling efficiencies, and obtained 96.17% and 98.68% efficiencies for mouse hepatocyte maturation and human hepatic reprogramming samples, respectively. In revision, we also prepared two batches of samples for proteomic quantification, and 94.88% and 90.69% for the former and latter, respectively. We understand that the TMT labeling efficiency might be slightly different using different methods. Thus, we provided the details on the calculation of TMT labeling efficiencies to ensure the reproducibility of our study. We revised the manuscript as below:

Main text, page 36, paragraph 2, lines 736-755, changed,

“To measure the TMT labeling efficiency, msConvert⁷³ was used to convert the data format of raw MS/MS data into mascot generic format (MGF) files. The monoisotopic

reporter masses of different TMT 6-plex labels were downloaded from MaxQuant²⁷, including 126 (126.127726 Da), 127 (127.124761 Da), 128 (128.134436 Da), 129 (129.131471 Da), 130 (130.141145 Da), and 131 (131.138180 Da). Using 0.005 Da as the reporter mass tolerance, for each batch, the total number of MS/MS spectra in MGF files was defined as T , while the number of MS/MS spectra labeled with at least one detected TMT 6-plex reagent was defined as L . Then, the TMT labeling efficiency E of each batch was calculated as below:

$$E = \frac{L}{T}$$

By this approach, we obtained 96.17% and 98.68% efficiencies for mouse hepatocyte maturation and human hepatic reprogramming samples, respectively.

During revision, we additionally prepared two batches of samples for proteomic profiling, without further phosphopeptide enrichment. The efficiencies of TMT labelling were 94.88% and 90.69% for additional samples of mouse hepatocyte maturation (Batch 1: MH [126, 127, 128] and CLiP-Hep [129, 130, 131]) and human hepatic reprogramming (Batch 2: GFP [126], FHH-2.25d [127], and FHH-5d [128]), respectively.”

For your concern on the hiHep dataset without any biological or technical replicates, our viewpoints are: 1) Either biological or technical replicates can only evaluate the reproducibility of omic data in statistics. They cannot ensure the reliability of new findings in biology. 2) We believe that various and additional experimental assays should be performed to validate the predictions derived from omics data. 3) Our major findings of 5 central PKs are rooted from both computational analyses and further experiments, and will not be influenced with or without any replicates. In our future studies, we will try our best to include replicates if budget permits.

4. The authors report CILK1 T157 and Y159 phosphorylation sites in the supplementary data table and supplementary figure 2g from the MH/lpcHep set with identical values for both, whereas in the text (line 238) only T157 is referred to. I was not able to obtain a peptide-spectrum match for either T157 or Y159 phosphorylated peptides derived from CILK1 from the raw data. Although I used a different search engine than the authors, this needs to be clarified, even if it requires manual inspection of the peptide-mapped spectrum, because this phospho-peptide is explicitly discussed in the manuscript twice (lines 238 and 552).

Response: The T157 site has been reported to be a PK activity-associated p-site of Cilk1, and a higher phosphorylation level of T157 denotes a higher PK activity of Cilk1. However, the biological function of Y159 is unclear. In our analysis of mouse hepatocyte maturation, we indeed found that the phosphorylation level of Cilk1 T157 was increased in CLiP-Hep against MH, indicating that the PK activity of Cilk1 was enhanced in CLiP-Hep cells. However, the changes of PK activities do not indicate the corresponding PKs must be functionally important, and our further experimental validations didn't validate Cilk1 to be an essential PK for mouse hepatocyte maturation.

For the peptide-spectrum match of Cilk1 T157 and Y159, we checked our phosphoproteomic results and added a new Supplementary Fig. 3m. In revision, we added a description in the legend of Supplementary Fig. 3 as “...**m The MaxQuant MS/MS spectrum of the phosphopeptide for T157 and Y159 in Cilk1...**”

5. *Some minor points:*

Line 239/240 States that Prkaa1 was undetectable in lpc-Hep and MH cells, whereas supplementary Fig. 2g and the associated data table shows phosphorylated peptides of Prkaa1 from these cells. This point is still not clear to me even if clarified already, sorry for that.

Response: We performed quantitative PCR to analyze the expression of *Prkaa1* in lpc-Hep (currently named as CLiP-Hep cells) and MH cells. However, we did not detect the expression of *Prkaa1*. To avoid confusion, we removed this description from the manuscript. In addition, we performed quantitative PCR experiment to analyze the effect of *Prkaa1* in CLiP-Hep maturation (Figure 2k). As we shown in Figure 2k, *Prkaa1* has little effect on hepatocyte maturation.

6. I previously requested “a metadata table that allows to map instrument raw data files to specific experiments”. Please provide such a table with the MS raw data on iProX, stating the filenames and which experiment they belong to and which experimental condition is in which TMT channel and so on. This table makes it less tedious to reanalyze the data and should always accompany deposited MS data; the authors use published datasets themselves in this study.

Response: We have submitted the metadata table on iProX, including the description of experiments, experimental conditions, TMT channels and files.

7. L32 “higher than other methods” – please specify.

Response: In revision, we provided the details on comparison of the CKI accuracy to other methods as below:

Main text, page 1, paragraph 1, lines 33-34, changed,

“...Using known PKs associated with anti-cancer drug resistance, **the accuracy of CKI denoted by area under the curve (AUC) is 5.2% to 29.5% higher** than other methods...”

Main text, page 8, paragraph 1, lines 138-145, added,

“CKI is a model-based method, and no prior data were used for training. To evaluate the performance of CKI, we predicted potentially central PKs using data from two types of previously published drug-resistance studies, DOX resistance and genistein resistance (Supplementary Note 1, Supplementary Fig. 1, 2 and Supplementary Data 1)¹⁴⁻¹⁷. Compared to other methods such as Kinase-Substrate Enrichment Analysis (KSEA)^{21,22}, the AUC values of CKI were 5.2% (0.8278 vs. 0.7871) and 29.5% (0.7912

vs. 0.6112) higher for DOX and genistein resistance, respectively (Supplementary Fig. 1d, k).”

8. L73 AUC scores – Please explain. What is the meaning of these numbers?

Response: In machine learning, the standard definition of area under the curve (AUC) is the probability that a predictive classifier will rank a randomly chosen positive example higher than that of a randomly chosen negative example. The higher the AUC value, the better the performance of the computational model at distinguishing between the positive and negative samples. Frankly, we think such a definition will not be easily understood by biologists. Thus, we revised the manuscript by adding a much simpler description as below:

Main text, page 49, paragraph 2, lines 1002-1003, added,

“...A higher AUC value denotes a higher accuracy of a predictive model in general.”

9. L99 “analyze and integrate”

Response: We have changed the corresponding description as “...a trans-omics-based computational method to **analyze** and **integrate** transcriptomic data...”

10. L133 Please specify statistic test.

Response: We have changed the description as “...The Yate’s chi-squared test was performed to identify ...”

11. L234 Please delete (or specify/rephrase).

L289/290 Please delete (or specify/rephrase).

Response: We deleted these sentences in this revision.

12. L779 Different from the numbers I see (please see above). Should be clarified.

Response: We re-calculated the TMT labelling efficiencies, and addressed your concern at Q3.

13. L849 Please use actual human reference proteome. 2014 version is outdated.

Response: The trans-omic study of human hepatic reprogramming was started in 2014. At that time, the human reference proteome set of UniProt Version 201401 was not outdated. We understand that the results might be not identical if different versions of reference proteome sets were used. Thus, we analyzed the phosphoproteomic data using the human reference proteome set of UniProt Version 202203, and found that the

validated PKs were not changed. We present the new results in the last paragraph of Supplementary Note 3 as below:

Supplementary Information, page 9, paragraph 2, added,

“Our study on human hepatic reprogramming has been started since 2014, and at that time we used the human proteome set from UniProt (Version 201401)¹⁴ for database search. To test whether the update of UniProt will influence the p-site identification, we downloaded the recently released human reference proteome set from UniProt (Version 202203)¹⁴. It could be found that different versions of reference proteome sets did not significantly change the phosphoproteomic results, and only 305 (5.4%) p-sites identified from UniProt Version 201401 were not covered by UniProt Version 202203 (Supplementary Fig. 13h). Using UniProt Version 202203, we re-analyzed the transomic data, and CKI could still recall 13 of 15 previously predicted PKs, including the two validated PKs, PIM1 and PIM2 (Supplementary Fig. 13i).”

14. L853 Please also search for unphosphorylated PSMs.

Response: Our phosphoproteomic data sets were not appropriate for accurate quantification of non-phosphorylated proteins, because during phosphopeptide enrichment, some but not all negatively changed non-phosphorylated peptides might also be enriched. Thus, analysis of unphosphorylated PSMs cannot accurately provide the protein quantification information. To resolve this problem, we performed additional proteomic quantifications of mouse hepatocyte maturation and human hepatic reprogramming, without phosphopeptide enrichment. And the detailed analyses of proteomic data were present in Supplementary Note 3.

15. L857 For TMT quantification, was there any filter applied for minimal TMT signal? If so, please specify.

Response: The default parameters of MaxQuant were set for TMT quantification. We did not add more filters.

16. L857 Was there also an FDR correction on the protein level after the FDR correction on the peptide level? Please clarify.

Response: Thank you very much for your concern. In revision, we provided the details on FDR corrections as below:

Main text, page 40, paragraph 2, lines 828-838, added,

“...The false discovery rates (FDRs) of the peptide-spectrum match (PSM) and protein decoy fractions were all set to < 1%, and the minimum score for modified peptides was set to > 40.

For raw MS/MS data of mouse or human proteomes, the same reference proteome sets mentioned above were adopted for spectral library searching with MaxQuant

(v.1.4.1.2)²⁷. Similarly, we chosen Trypsin/P as the cleavage enzyme and 2 as the maximum missing cleavages. The carbamidomethyl (C) was selected as fixed modification and Oxidation (M) and Acetyl (Protein N-term) were taken as the variable modifications. The minimum peptide length and mass tolerance were set as 7 and 0.02 Da, respectively. FDRs of PSM and protein decoy fractions were set to < 1%.”

17. Method section: What is meant with TMT intensity? TMT raw intensity or corrected for noise/background?

Response: From the MaxQuant results, we directly used the raw TMT intensities from the column “Reporter intensity” without any correction. We indicated this point by revising the manuscript as below:

Main text, page 41, paragraph 2, lines 839-840, added,

“From the MaxQuant results, the raw reporter intensities were taken as the quantification values of p-sites or proteins.”

18. Fig.2i p-sites>p-peptides?

Fig.3i p-sites>p-peptides?

Response: In our phosphoproteomic data, the number of non-redundant phosphopeptides is lower than the number of non-redundant p-sites, because multiple p-sites could be identified from one phosphopeptide. We re-checked the analysis and confirmed these numbers to be correct.

19. Fig.3m Number of p-sites in this study higher than in any other database? My first thought, presentation in the figure is a little confusing. I suggest to edit to show that the numbers show the overlap with the database reported p-sites.

Response: We apologize for this inconvenience, and have changed the label of the X axis to “Number of overlap”, indicating the number of identified p-sites that are also covered by other public database. The original Supplementary Fig. 3g and Fig. 5i have been updated.

20. Fig3i-m Rather supplementary figures, although they look nice.

Response: The original Fig. 3i-m were put into Supplementary Fig. 3c-g, as well as the corresponding descriptions.

21. Fig 3n: Reduce circle opacity for better readability, numbers in black for contrast.

Response: Based on your suggestion, we have updated the Fig. 2i (Original Fig. 3n) and Supplementary Fig. 5i (Original Fig. 4h) for better readability.

22. *Fig. 4b-h ∅ Supplementary figures rather than main figures.*

Response: We have put Fig. 4b-h into Supplementary Fig. 5c, e-i, and q, together with the corresponding descriptions.

23. *Fig. 6d: Scale from 1-0.99?*

Response: We re-checked the analysis and confirmed that these numbers are correct, the spearman's correlation coefficients between different biological replicates ranged from 0.993-0.994, while the coefficients between different samples ranged from 0.988-0.990.

24. *Fig 6: Move all panels just characterizing the dataset(s) with little specific information to supplementary figures.*

Response: The original Fig. 6b-d have been put into Supplementary Fig. 8a-c, together with the corresponding descriptions.

25. *Fig. 7f: Please increase contrast of PI signal in pictures.*

Response: We increased the contrast of PI signals in Fig. 6f (Original Fig. 7f) accordingly.

26. *Overall: Please reduce number of figures - 8 is too extensive - and end with an illustrative, simple panel supporting the discussion putting findings into biological context.*

Response: We have put the original Fig. 2 into Supplementary Fig. 1, and the corresponding Section entitled "Evaluation of the accuracy of CKI" was moved into Supplementary Note 1. In the current version, there are only 7 figures in the main text.

27. *In summary, I would like to reiterate that this study is interesting, important beyond the field and should be published in Nature Communications. I strongly suggest to repeat the two MS experiments - and possibly the associated RNAseq runs if needed -, so that they include protein level measurements as controls for changes in quantified phosphosites and in case of hiHep data biological replicates. This is doable and I would like to encourage the authors to just get to it. The manuscript improved by a lot already and will be a good paper in the end.*

Response: We are very grateful for your helpful advice, and performed additional quantitative proteomic profiling of mouse hepatocyte maturation and human hepatic reprogramming, respectively. We have carefully addressed each of your concerns.

Reviewer #3:

The authors partially answered the questions from this reviewer. However, there are many critical issues involved in the data and the experimental design of this study and some minor concerns, described below.

1. Critical issues:

Fig. 1a: There may be a correlation between the levels of protein phosphorylation and a kinase-substrate network. However, I am wondering whether there is a correlation between the expression levels of PK genes and the phosphorylation levels of PK-target proteins. Normally, the phosphorylation levels of PK-target proteins change in the result of activation/inactivation of cell intrinsic/extrinsic signals regardless of the expression levels of PK genes. Thus, this study may not need a trans-omic analysis, but only need a phosphoproteome analysis. Fig. 1c and e actually support this opinion.

Response: In the revision, we carefully analyzed the correlations between mRNA expression levels of PKs and the corresponding substrate p-site intensities. The weak correlations between the transcriptomic data and the phosphoproteomic data indicated that the two data types are relatively independent, and therefore a trans-omic analysis can provide new insights by integrating the information from the two data types to reach a high accuracy. We clarified this point by revising the manuscript as indicated below:

On page 26, paragraph 1, we added:

“...Using all pairwise comparisons of the transcriptomic and phosphoproteomic data for each biological process, we analyzed correlations between PK gene expression and the intensity of the corresponding substrate p-sites based on the predicted ssKSRs. The average Spearman’s correlation coefficients were calculated as 0, -0.02, 0.14, and 0.10 for DOX resistance, genistein resistance, hepatocyte maturation, and hepatic reprogramming, respectively (Supplementary Fig. 9a-d), implying a weak correlation between PK gene expression and phosphorylation of the target p-sites.”

On page 50, paragraph 4, we added:

“Correlation analysis of mRNA expression levels of PKs and their substrate p-site intensities

The correlations between mRNA expression levels of PKs and their corresponding substrate p-site intensities were measured based on the ssKSRs, using the Spearman’s correlation. For each ssKSR, the Spearman’s correlation coefficient (ρ) between the mRNA expression level of the upstream PK and the intensity of the downstream p-site was calculated across different samples or replicates.”

Reviewer’s response: The authors state that the transcription levels of PKs and the phosphorylation levels of PK-target proteins are independent, and therefore, the authors claim that the trans-omic analysis with these data can provide new insights. However, ROC curves show that mRNA expression data contributes little to the

prediction of central PKs; on the other hand, network data highly contributes to the prediction (Fig. 2d and 2k). Thus, it seems that a combined analysis with the p-site intensities and kinase-substrate network data is sufficient for the prediction. In other words, mRNA expression data and trans-omic analysis may not be necessary for predicting the CKI algorithm.

Response: When we started the study of human hepatic reprogramming in 2014, we did not know whether both phosphoproteomic and transcriptomic data were informative for prediction of potentially central PKs. By experimental validations, we identified two PKs, PIM1 and PIM2, to be essential in human hepatic reprogramming. Our major finding, PIM1, cannot be correctly predicted any longer, if transcriptomic data is excluded. This result supports the usefulness of transcriptomic data and our trans-omic analysis. The details on prediction of potentially central PKs with or without transcriptomic data were shown in below:

Supplementary Information, page 8, paragraph 2, added,

“...In addition, we evaluated the accuracy of CKI with or without transcriptomic data. By excluding transcriptomic data, 31 potentially central PKs were predicted merely from phosphoproteomic data, including all the three newly validated PKs (Supplementary Fig. 12g), indicating that including transcriptomic data or not did not influence the final validations.”

Supplementary Information, page 9, paragraph 1, added,

“...By excluding transcriptomic data, 13 potentially central PKs including a validated PK, PIM2, were predicted merely from phosphoproteomic data. However, the most important finding of this study, PIM1, could not be covered any longer (Supplementary Fig. 13g).”

2. Fig. 1b-e: The authors conducted a simulation using public data to show a predictive performance of the CKI algorithm, and the number of reported PKs among the resulting candidates is shown in the text, which is an important argument to estimate the performance of CKI. However, to attain objective evaluation of the prediction method, the number of false negatives, which were not predicted in the CKI algorithm, should also be summarized and disclosed.

Response: We agree with the reviewer about the importance of including these data. In the revision, we manually curated known PKs associated with DOX or genistein resistance from the scientific literature. These known PKs were used as an independent testing dataset to measure the accuracy of CKI and allow comparison of CKI to other methods. For the testing dataset, we counted the true positive (TP), true negative (TN), false positive (FP), and false negative (FN) values and calculated sensitivity (Sn) and specificity (Sp). ROC curves were generated for each method and the corresponding AUC values calculated for each curve. We have also included confusion matrices and PCA analyses. The original Fig. 1 was split into the new Figs. 1, 2 and 3. Evaluations

of CKI using the DOX and genistein resistance testing data are shown in Fig. 2. We added a new section entitled “Evaluation of the accuracy of CKI” in the Results to present these details. The corresponding methodology for performance evaluation has been added in a new Methods section entitled “Performance evaluation”. To incorporate these changes, the manuscript has been updated as indicated below:

On page 8, paragraph 1, we changed the portions shown in red:

“Evaluation of the accuracy of CKI

CKI is a model-based method, and no prior data were used for training. To evaluate the performance of CKI, we predicted potentially central PKs using data from two types of previously published drug-resistance studies, DOX resistance and genistein resistance (Fig. 2, Supplementary Fig. 1 and Supplementary Data 1)¹⁴⁻¹⁷.

In 2016, Zannotto-Filho et al. profiled the transcriptomic changes of a human breast cancer cell line (MDA-MB231) and osteosarcoma cell line (U-2 OS) using a number of chemotherapeutic agents including DOX (1 μ M/L, 8 h), and uncovered a regulatory role of nuclear factor erythroid 2-related factor 2 (NFE2L2/NRF2) in chemotherapy resistance¹⁴. In another study, Hogrebe et al. quantified the phosphoproteomes of U-2 OS cells with or without DOX treatment (5 μ M/L, 2 h)¹⁵. From these two studies^{14,15}, we identified 2197 DEMs and obtained quantitative data for 27,590 p-sites in 5592 proteins (Fig. 2a, b). In the phosphoproteomic data, there were 22,744 phosphoserine residues (pS, 82.44%), 4451 phosphothreonine residues (pT, 16.13%), and 395 phosphotyrosine residues (pY, 1.43%) (Fig. 2c). From the literature, known PKs involved in DOX resistance were curated (Supplementary Data 1). Using this independent testing dataset, we evaluated the accuracy of CKI by calculating the AUC score of

the method that incorporated all three types of data (mRNA expression, substrate p-site intensity, and kinase-substrate network) and of models built with each data type alone (Fig. 2d). We also compared to a z-score-based method, Kinase-Substrate Enrichment Analysis (KSEA), which predicts potentially important PKs only from phosphoproteomic data^{21,22}. We found that using only data related to changes in substrate p-site intensity or the kinase-substrate network had a comparable accuracy to that of KSEA (AUC = 0.7635, 0.8058, and 0.7871, respectively) (Fig. 2d). However, when the three types of data were integrated with CKI, the AUC value was higher than the other methods (0.8278) (Fig. 2d). The confusion matrix and principal component analysis (PCA) indicated that known and unknown PKs could be separated in a reasonably accurate manner (Fig. 2e, f). Of the final 22 PKs predicted by CKI, 10 had already been reported as truly important for DOX resistance in U-2 OS cells, including ATM/ATR²³, MAPK1³²⁴, and CHEK1²⁵ (Fig. 2g and Supplementary Data 1). In this case, the transcriptomic data were less informative and only contributed to prioritization of one of the 22 final candidate PKs (Fig. 2g and Supplementary Fig. 1a). In addition, we mapped the quantified p-sites to the 22 PKs where available, and found that only one PK activity-associated p-site, ATM S1981²⁶, was up-regulated after DOX treatment (Fig. 2g, Supplementary Fig. 1b and Supplementary Data 1).

Next, we integrated the transcriptomic and phosphoproteomic data of breast cancer cells treated with or without genistein from previous reports^{16,17}. In total, we identified 3264 DEMs and obtained quantitative data for 17,403 p-sites in 7149 proteins (Fig. 2h-j). Using known genistein-associated PKs as the testing data, the AUC value of CKI was calculated as 0.7912, showing a much higher accuracy than all of the other methods tested (Fig. 2k). Again, the results from the confusion matrix and PCA supported CKI could distinguish between known and unknown PKs (Fig. 2l, m). In this case, the transcriptomic data also contributed to prioritizing five of the 21 final candidate PKs (Fig. 2n, Supplementary Fig. 1c and Supplementary Data 1). No PK activity-associated p-sites were identified to be differentially regulated after genistein treatment (Supplementary Fig. 1d and Supplementary Data 1).

Because both DOX and genistein are potent DNA damage inducers¹⁴⁻¹⁷, it was not surprising that the major PK known to respond to DNA damage, ATM, was successfully predicted as a central PK in both cases (Fig. 2g, n). Taken together, the results showed a promising performance of CKI in well-studied drug-resistant cancer cells.”

On page 49, paragraph 3, we added:

“Performance evaluation

For the DOX or genistein resistance, known regulatory PKs curated from the literature were taken as the positive data, while other remaining PKs predicted with at least one ssKSR were taken as the negative data. To evaluate the accuracy of CKI and other methods, true positive (TP), true negative (TN), false positive (FP), and false negative (FN) values were counted, and we calculated 2 commonly-used measurements, including sensitivity (Sn) and specificity (Sp) as below:

$$Sn = TP / (TP + FN), Sp = TN / (TN + FP)$$

The receiver operating characteristic (ROC) curves were illustrated based on Sn and 1-Sp values, and the corresponding AUC scores were calculated.

Then, the confusion matrix under a selected threshold of pairwise comparisons was calculated as below:

Actual

Known Unknown

Predicted Known TP/(TP+FN) FN/(TP+FN)

Unknown FP/(FP+TN) TN/(FP+TN)

For each of known or unknown PKs, the number of positive hits and the sum of all minus-log transformed (-lg) p values were counted for PCA analysis.”

Reviewer’s response: In the revised manuscript, the authors presented confusion matrixes to evaluate the predictive performance of the CKI algorithm (Fig. 2e and 2l). The matrixes showed very low false-negative ratios of the CKI algorithm (the scores are 0.03 and 0.04 in Fig. 2e and Fig. 2l, respectively), but simultaneously showed high false-positive ratios of the CKI algorithm (the scores are 0.55 and 0.81 in Fig. 2e and Fig. 2l, respectively). These data indicate that the CKI algorithm permits contamination of a large number of false-positives. Thus, the predictive performance of the CKI algorithm is considered to be inadequate for the prediction of central PKs.

In addition, descriptions for calculating scores in the confusion matrixes contain mistakes: for example, FN and FP should be reversed.

Response: In bioinformatics, we need to balance the prediction accuracy and originality of biological findings. If many PKs have been identified to be involved in human hepatic reprogramming, we will use these PKs as the benchmark data set to train a much accurate model, termed “model-free method” in bioinformatics. However, the originality of such a study will be greatly diminished for biologists. When we started this study, we clearly understood that no PKs were identified previously, and our findings would be the first report to identify central PKs in regulating human hepatic reprogramming. Because no training data set was available, the conventional model-free methods were not applicable. Thus, CKI is a model-based method, and no prior data were used for training. Undoubtedly, its accuracy will be lower than any model-free methods. However, the discovery of 5 central PKs supported the accuracy and efficiency of CKI.

For your concern on the confusion matrices, we revised the manuscript as below:

Main text, page 49, paragraph 3, changed,

“Then, the confusion matrix under a selected threshold of pairwise comparisons was calculated as below:

		Actual	
		Known	Unknown
Predicted	Known	$\frac{TP}{TP + FP}$	$\frac{FP}{TP + FP}$
	Unknown	$\frac{FN}{FN + TN}$	$\frac{TN}{FN + TN}$

3. Fig. 2a: The authors previously reported that SV40-LT was required for proliferation of human iHep. Show the details about the method of iHep induction used in this study. If SV40-LT was used, the authors should carefully examine an unexpected effect of SV40-LT in the analysis. If it was not used, iHep cultures may contain many fibroblasts, and thus cell heterogeneity is a problem for analysis. The authors should isolate iHep before the analysis.

Response: We apologize for the lack of clarity on this point. SV40-LT is required for long-term proliferation of hiHep cells. However, in this study, SV40-LT was not used to avoid unexpected effects of SV40-LT. Without SV40-LT, hiHep cells are capable of division for a limited number of times. We agree with the reviewer that the hiHep cell culture contained many fibroblasts. Although we have tried to purify hiHep cells for several years, we still do not have an efficient method to isolate hiHep cells; we do not have a workable cell surface marker protein that can be used to isolate hiHep cells by

FACS sorting, and there is also no workable reporter system that can effectively mark hiHep cells. However, the experimental results supporting our conclusions in the manuscript cannot be explained by fibroblast contamination.

Reviewer's response: Because many fibroblasts contaminate hiHep cultures, the data are not reflective of the nature of hiHep. Thus, the results in this study cannot be trusted. hiHep could be isolated using markers of epithelial cells, such as E-cadherin, because hiHep are epithelial cells but fibroblasts are mesenchymal cells.

Response: In this study, we focused on the early events during hepatic reprogramming. Thus, we performed CKI analysis on HDFs undergoing hepatic conversion for 2.25 days and 5 days. Though the expression of hepatic genes increased after 2.5 days induction (Supplementary Figure 4b), the expression levels of ALBUMIN and E-cadherin in HDF+FHH (5d) are far below those in HDF+FHH (14d). The protein of ALBUMIN and E-cadherin are not detectable by immunofluorescence staining (Response Figure 1). Thus, it is impossible to isolate hiHep cells by FACS sorting during the early stage of hepatic conversion.

The contamination of fibroblasts is a great problem for the study of early stage of cell fate reprogramming. During the early stage of cell fate reprogramming, it is difficult to distinguish cells that will be successfully reprogrammed from the bulk cells. The contamination of fibroblasts reduced the changes of signaling pathways during cell fate reprogramming. Thus, a more systematic method to integrate multiple omic data, such as CKI, could be applied to predict the potentially important signaling pathways. In this study, we predicted 15 potentially important hepatic reprogramming-associated PKs based on CKI. We also experimentally validated the functions of individual central PK candidate, and showed the important functions of PIM1, PIM2, TRIB2, TSSK3, PLK1, PLK2 and MEK2 during hepatic conversion. Our results supported the usefulness of CKI for prediction of central PKs during cell fate conversion, especially at the early stage when purification of successfully converted cells is not applicable.

Response Figure 1. Representative image of ALBUMIN and E-cadherin staining in HDFs overexpression FHH or LT+FHH for indicated days.

4. Fig. 2d-f: Do these data indicate an increase or decrease of ALB expression level in each cell, an increase or decrease of reprogramming efficiency, or an increase or

decrease of proliferation of ALB-positive cells? The authors need to answer this question with suitable quantitative data.

Response: We appreciate the reviewer raising this question. In flow cytometry experiments on hiHep cells with ALBUMIN immunofluorescence staining (Supplementary Fig. 5), we observed an increase in ALBUMIN⁺ cells. However, we did not observe an obvious increase in the fluorescence strength of ALBUMIN⁺ cells. Thus, the data more likely indicate an increase in ALB⁺ cell number, rather than an increase of ALB expression in each ALB⁺ cell.

By co-staining ALBUMIN and EdU in hiHep cells treated with EdU for 24 hours, we observed an increase of EdU⁺ in ALB⁺ hiHep cells overexpressing PIM1 (Supplementary Fig. 6c), suggesting increased proliferation capability of hiHep cells in the context of PIM1 overexpression. In addition, we also observed decreased cell death in PIM1-overexpressing HDFs undergoing hepatic reprogramming (Supplementary Fig. 7d,e). Thus, the increased proliferation and decreased cell death of ALB⁺ cells resulted in the increased reprogramming efficiency.

Reviewer's response: Additional data are helpful, but not showing an increase of the reprogramming efficiency. The authors should describe the results correctly: the data suggest that PIM1 overexpression activate proliferation of hiHep and decrease cell death of HDFs.

Response: We appreciate the reviewer for this constructive suggestion. We improved the description accordingly. We did not mention “*PIM1 overexpression activate proliferation of hiHep and decrease cell death of HDFs*” at this part (manuscript text related to Figure 4), because we verified the functions of PIM1 in activation of proliferation and suppression of cell deaths in Figure 5.

On page 14, Line 274, we changed the portions shown in red:

“PIM1 promotes hepatic reprogramming”

On page 15, Line 292, we changed the portions shown in red:

“...Immunofluorescence staining also showed that the proportion of ALBUMIN positive cells was increased in response to PIM1 overexpression (Fig. 4g and Supplementary Fig. 7)...”

On page 15, Line 296, we changed the portions shown in red:

“...These results demonstrated that PIM1 efficiently promotes hepatic reprogramming.”

5. Fig. 2d-f: In addition to ALB expression, other hepatic marker expression and hepatic function should be examined in the screening. Only ALB expression is not sufficient for proper assessment.

Response: We agree that ALB expression alone is not sufficient for proper assessment. We therefore quantified the expression levels of AAT and ARG, two hepatic functional genes (Fig. 4j,l, Supplementary Fig. 4l). The new results further support our finding that PIM family proteins promote hepatic reprogramming. We also performed ALBUMIN immunofluorescence staining and validated four PKs that showed statistically significant regulation of ALB expression (Fig. 4k).

Reviewer's response: Hepatic function was not analyzed, and there is no statistical data in fig 4k.

Response: In the revised manuscript, we performed functional assays to evaluate the functions of PLK1, PLK4 and NEK2 by gene knockdown experiments (Supplementary Figure 6b-d). As we showed in Supplementary Figure 6b-d, knockdown of PLK1, PLK4 or NEK2 significantly suppressed hepatic reprogramming.

In addition, as suggested by this reviewer, we showed the statistical data in Fig. 3d (originally Fig. 4k).

On page 13, Line 265, we changed the portions shown in red:

“...Overexpression of *PLK1*, *PLK4* or *NEK2* did not have an obvious effect on the expression of hepatic-lineage genes (Fig. 3c). However, knockdown of *PLK1*, *PLK4* or *NEK2* led to significant inhibition of hepatic conversion (Fig. 3e and Supplementary Fig. 6a-d)....”

6. Fig. 2d: As I mentioned above, an increase of PK gene expression and that of target protein phosphorylation may be less relevant. Thus, the screening should be done by gene knockdown/knockout experiments, but not by an overexpression strategy. In fact, Fig. 2f shows a significant effect of the knockdown of PLK1, PLK4 or NEK2 on the induction of ALB gene expression in fibroblasts.

Response: We greatly appreciate the reviewer's constructive suggestion. We agree that overexpression experiments alone are not sufficient for the screening (Fig.2d, updated Fig. 4j). Thus, we performed shRNA-mediated gene knockdown experiments for the PK screening (Fig. 4l, Supplementary Fig. 4l). We found that knockdown of PIM1 or PIM2 significantly suppressed the induction of hepatic-lineage genes during hepatic reprogramming, which supported our finding in the overexpression experiments that PIM1 and PIM2 can promote hepatic reprogramming. We also found that knockdown of PLK1, PLK2, or NEK2 could also inhibit hepatic reprogramming. However, overexpression of PLK1, PLK2, or NEK2 showed no significant effect on the induction of hepatic-lineage genes (Fig. 4j,l). Interestingly, although overexpression of PLK4 or ROCK2 did not show obvious effects on hepatic reprogramming, knockdown of PLK4 or ROCK2 did significantly promote hepatic reprogramming (Fig. 4j,l).

Reviewer's response: Additional data indicate that the results of gene overexpression experiments in this study are unreliable. As this reviewer mentioned previously, an

increase of PK gene expression and that of target protein phosphorylation may be less relevant. Thus, the screening should be done by gene knockdown/knockout experiments, but not by an overexpression strategy. This reviewer cannot understand why the authors did not evaluate the function of PLK1, PLK2, and NEK2.

Response: We thank this reviewer for indicating the importance of gene knockdown experiments. In the revised manuscript, we performed additional experiments to evaluate the functions of *PLK1*, *PLK2* and *NEK2* by gene knockdown (Supplementary Figure 6). As we showed in Supplementary Figure 6b-d, knockdown of *PLK1*, *PLK4* or *NEK2* significantly suppressed hepatic reprogramming.

On page 13, Line 265, we changed the portions shown in red:

“...Overexpression of *PLK1*, *PLK4* or *NEK2* did not have an obvious effect on the expression of hepatic-lineage genes (Fig. 3c). However, knockdown of *PLK1*, *PLK4* or *NEK2* led to significant inhibition of hepatic conversion (Fig. 3e and Supplementary Fig. 6a-d)....”

7. Fig. 2d: *In this study, the authors focused on PKs that upregulate ALB expression, such as PIM1 and PIM2. However, PKs suppressing ALB expression, such as TRIB2 and TSSK3, could also be predicted as the candidates of central PKs in hepatic reprogramming. To evaluate the CKI as a “versatile and comprehensive prediction method for PKs”, the authors should also focus on the PKs that suppress ALB expression and reveal the roles of these PKs, similar to PIM1.*

Response: We agree that PKs suppressing ALB expression, such as TRIB2 and TSSK3, could also be predicted as candidate central PKs in hepatic reprogramming. Thus, we analyzed the expression of AAT and ARG in the context of TRIB2 or TSSK3 overexpression. We found that overexpression of TRIB2 had little effect on AAT and ARG expression. In contrast, although overexpression of TSSK3 had little effect on ARG expression, it significantly suppressed AAT expression (Fig. 4j). We also performed shRNA-mediated gene knockdown of TRIB2 and TSSK3. We found that knockdown of TRIB2 or TSSK3 had moderate or little effect on hepatic reprogramming (Fig. 4k). These results suggest that TRIB2 and TSSK3 could regulate hepatic reprogramming in a moderate manner. Although most of the candidate central PKs showed more or less effects on the regulation of hepatic-lineage genes, two of the 15 predicted central PKs, PIM1 and PIM2, were validated to regulate all examined three hepatic-lineage genes in both overexpression and knockdown experiments.

Reviewer’s response: The authors added AAT and ARG expression data under overexpression conditions of candidate PKs (Fig. 4j) and ALB, AAT, and ARG expression data under knockdown conditions of the PKs (Fig. 4l). Overexpression of PIM1 and PIM2 changed the expression of all three hepatic genes, while the knockdown experiments revealed that five PKs (PIM1, PIM2, PLK1, PLK2, and NEK2) also changed the expression of all three hepatic genes. Based on these data, this

reviewer cannot understand why the authors did not evaluate the function of PLK1, PLK2, and NEK2. As this reviewer mentioned previously, an increase of PK gene expression and that of target protein phosphorylation may be less relevant. Thus, the screening should be done by gene knockdown/knockout experiments, but not by an overexpression strategy.

Response: We thank this reviewer again for suggesting the evaluation of the functions of *PLK1*, *PLK2* and *NEK2* by gene knockdown experiments. As we mentioned above, in the revised manuscript, we performed additional experiments to evaluate the functions of *PLK1*, *PLK2* and *NEK2* by gene knockdown (Supplementary Figure 6). As we showed in Supplementary Figure 6b-d, knockdown of *PLK1*, *PLK4* or *NEK2* significantly suppressed hepatic reprogramming.

8. Fig. 5g: *Do these data indicate an increase of ALB and CYP3A4 expression levels in each cell, an increase of reprogramming efficiency, or an increase of ALB-positive and CYP3A4-positive cells upon inhibition of ferroptosis? The authors need to answer this question with suitable quantitative data.*

Response: To address this question, we performed flow cytometry with HDF+FHH cells. As shown below in Response Fig. 2, we observed an increase in ALBUMIN+ cells after treatment with the ferroptosis inhibitor NAC. However, we did not observe an obvious increase in ALBUMIN immunofluorescence levels in the ALBUMIN+ cells. Thus, it is most probable that the ferroptosis inhibitor promoted reprogramming efficiency or cell survival of ALB+ cells, rather than increasing ALB expression levels in each cell. We did not perform flow cytometry with CYP3A4 staining because none of the CYP3A4 we tested worked sufficiently well for flow cytometry in our hands.

Response Fig. 2 Flow cytometry showing HDF+FHH for ALBUMIN immunofluorescence staining.

Reviewer's response: Considering that PIM1 and ferroptosis inhibitors increase the reprogramming efficiency is simplistic. In the present state of the work, the authors should consider that PIM1 and ferroptosis inhibitors support proliferation of hiHep and decrease cell death of HDFs.

Response: We appreciate this reviewer for this comment. We have revised the manuscript according.

On page 14, Line 274, we changed the portions shown in red:
“**PIM1 promotes hepatic reprogramming**”

On page 15, Line 292, we changed the portions shown in red:

“...Immunofluorescence staining also showed that the **proportion of ALBUMIN positive cells** was increased in response to *PIMI* overexpression (Fig. 4g and Supplementary Fig. 7)...”

On page 15, Line 296, we changed the portions shown in red:

“...These results demonstrated that *PIMI* **efficiently promotes** hepatic reprogramming.”

9. The title of this paper should be changed appropriately along with the data obtained in this study.

Response: We changed the title as suggested: “Trans-omics-based inference of central protein kinases”.

Reviewer’s response: The new title is inconcrete and overstatement. The findings in this study should be reflected correctly to the title. In the abstract, the authors mentioned that “PIMI promotes hepatic conversion and protects HDFs from reprogramming-induced ferroptosis and cell cycle arrest” as the conclusion of this study. This sentence may be more suitable as the title than the present one.

Response: We have changed the title as suggested: “PIMI promotes hepatic conversion by overcoming reprogramming-induced ferroptosis and cell cycle arrest.”

10. Minor concerns:

Fig. 1d: What are hepatic progenitor cells used here? Show the details about those cells, and also show what characterizes them as hepatic progenitor cells.

Response: We regret the omission of this important information in the original manuscript. We generated proliferative bipotent hepatic progenitor cells from primary mouse hepatocytes following a well-established dedifferentiation protocol (Katsuda et al., Cell Stem Cell, 2017). The liver progenitor cells are capable of differentiation into both biliary epithelial cells (lpc-BEC) and hepatocytes (lpc-Hep, Fig. 3b-e). The lpc-BEC cells expressed bile duct marker genes, including CK19, Cfr, Aqp1 and Ae2 (Fig. 3e), and can form ductal structures (Fig. 3b,c). The lpc-Hep cells were ALBUMIN positive (Fig. 3d) and expressed the hepatocyte marker genes Aat and Alb (Fig. 3e).

To clarify these points, we revised the manuscript as indicated below:

On page 10, paragraph 3, we changed the portions shown in red:

“To further validate the accuracy of CKI, we analyzed the hepatocyte maturation process in mouse liver progenitor cells (Fig. 3a). We converted mouse hepatocytes to proliferative bipotent liver progenitor cells (lpc) using a previously established protocol²⁷. The lpc cells could then differentiate into either CK19⁺ biliary epithelial cells (lpc-BECs) that can form ductal structures (Fig. 3b, c and e) or ALBUMIN⁺ hepatocytes (lpc-Heps, Fig. 3d, e). However, lpc-Heps generated with this differentiation protocol are relatively immature²⁷, meaning that further lpc-Hep maturation is required for the application of liver-progenitor-cell-derived hepatocytes.

To identify central PKs that potentially promote the maturation of lpc-Hep cells, we profiled the transcriptomes and phosphoproteomes of freshly isolated primary mouse hepatocytes (MHs) and lpc-Hep cells (Fig. 3a)."

Reviewer's response: The authors must describe the name of cells used in this study correctly to avoid misunderstanding. Chemically induced liver progenitors (CLiPs) or dedifferentiated hepatocytes should be used here.

Response: We used CLiPs for the name of chemically induced liver progenitor cells all through the manuscript accordingly.

11. Fig. 3a: PIM1 gene expression was increased after 5 days post-infection. However, hepatic gene expression was increased after 2 days post-infection (Supplementary Fig. 2b). Thus, it is suggested that PIM1 affects an increase or maintenance of ALB gene expression in iHep, but not involves in an initial event of hepatic reprogramming. The authors should clarify this important point with additional data.

Response: As shown in the quantitative PCR and immunoblotting results (Fig. 5a,b), PIM1 levels slightly increased 2.25 days after infection of FHH. Thus, induction of hepatic gene expression was accompanied by increased PIM1 expression levels. Moreover, shRNA-mediated knockdown of PIM1 significantly suppressed induction of hepatic-lineage genes (Fig. 5c), suggesting that PIM1 is required for the initial event of hepatic reprogramming. In addition, we performed functional analysis for induced cells with shPIM1. As shown in Fig. 5d, PIM1 knockdown markedly suppressed glycogen storage and ac-LDL uptake in HDF infected with FHH for 5 days, further supported that PIM1 is required for the initial event of hepatic reprogramming.

Reviewer's response: In fact, PIM1 expression is highly upregulated after 5 days post-infection. Also, PIM1 overexpression activated proliferation of hiHep and decreased cell death of HDFs. The authors should discuss the role of PIM1 in the hepatic reprogramming more precisely.

Response:

We appreciate this reviewer for this comment. We have revised the manuscript according.

On page 14, Line 274, we changed the portions shown in red:

"PIM1 promotes hepatic reprogramming"

On page 15, Line 292, we changed the portions shown in red:

"...Immunofluorescence staining also showed that the proportion of ALBUMIN positive cells was increased in response to PIM1 overexpression (Fig. 4g and Supplementary Fig. 7)..."

On page 15, Line 296, we changed the portions shown in red:

“...These results demonstrated that *PIM1* efficiently promotes hepatic reprogramming.”

12. Fig. 4b: Why were other pathways not analyzed in detail? The data shown in Fig. 5g indicate that ferroptosis inhibition has only a limited effect on the hepatic reprogramming. The authors should verify whether there are other critical pathways.

Response: We appreciate the reviewer's question. As shown in Fig. 4b (updated Fig. 6f), 17 pathways were significantly changed in *PIM1* overexpressors. Fifteen were liver-enriched metabolic or functional pathways (e.g., the PPAR signaling pathway, drug metabolism, pentose and glucuronate interconversions, and other similar pathways). Because we found that *PIM1* promotes hepatic reprogramming, it is not surprising that pathways related to liver functions were enriched. Among the 17 pathways, we identified two non-metabolic pathways related to cell cycle progression and ferroptosis. Thus, we focused on these two pathways.

We agree that ferroptosis inhibition partially contributes to the effects of *PIM1*. In Fig. 6g-m, we show that cell cycle also contributes to the effects of *PIM1*. Based on our results, we cannot exclude that any other pathways are related to the effects of *PIM1*.

Reviewer's response: The authors should carefully describe this in the discussion.

Response: We discussed this in below:

On page 25, Line 506, we changed the portions shown in red:

“In this study, 17 pathways were enriched upon *PIM1* overexpressing during hepatic reprogramming (Fig. 5c). We only validated two of them, including “cell cycle” and “ferroptosis”. Thus, there could be other downstream signaling pathways for *PIM1* during hepatic reprogramming....”

13. Fig. 4f: The authors should examine whether these signaling pathways are activated with an upregulation of endogenous *PIM1* expression and inhibited by *PIM1* knockdown/knockout during hepatic reprogramming.

Response: We analyzed the signaling pathways shown in Fig. 4f (updated Fig. 6m) during hepatic reprogramming. We also observed increased phosphorylation of 4D-BP1 and MYC along with upregulation of endogenous *PIM1* expression and decreased phosphorylation of MYC in *PIM1* knockdowns (Fig. 6l).

Reviewer's response: Where are the data of *PIM1* knockdown in fig 6l? Also, western blot data for total 4E-BP1 protein are missing in fig 6l and 6m.

Response: The data of *PIM1* knockdown for Fig. 5i (Original Fig. 6l) was shown in Supplementary Figure 8h. As this reviewer suggested, we performed western blot experiment of total 4E-BP1 in Fig. 5i and 5j.

Reviewer #4:

My concerns were addressed satisfactorily by the authors.

REVIEWER COMMENTS

Reviewer #3 (Remarks to the Author):

The authors should correctly respond to the issues related to their previous answers.

1. The authors' claim that "the CKI is a powerful method that can be used for discovery of regulatory PKs" is unreasonable, because false-positive values in the confusion matrix indicate low accuracy of the CKI (Supplementary Figure 1e and 1l). Unless this point is correctly revalued, it is difficult to accept the claims made in this paper regarding the usefulness of the CKI.
2. In Supplementary Figure 12g, the authors themselves show that transcriptome data are not necessarily essential for the detection of PK predicted from CKI. Therefore, description of the evaluation of CKI in this paper is an overstatement. The authors should correctly rewrite the abstract, introduction, and other parts of manuscript that refer to the universal usefulness of CKI.
3. The difference in AUC values is obtained by comparing CKI and "other methods", which is described in the abstract and introduction, is the comparison of CKI with KSEA. Thus, the authors should change "other methods" to KSEA.
4. The authors' comment, such as "In this study, we focused on the early events during hepatic reprogramming. Thus, we performed CKI analysis on HDFs undergoing hepatic conversion for 2.25 days and 5 days. Though the expression of hepatic genes increased after 2.5 days induction (Supplementary Figure 4b), the expression levels of ALBUMIN and Ecadherin in HDF+FHH (5d) are far below those in HDF+FHH (14d). The protein of ALBUMIN and E-cadherin are not detectable by immunofluorescence staining (Response Figure 1). Thus, it is impossible to isolate hiHep cells by FACS sorting during the early stage of hepatic conversion.", and Response Figure 1 should be included in the revised paper, both of which are important for understanding the experimental system.
5. shNEK2 may be correct in Supplementary Figure 6b-d. The authors should carefully check misspelling in the text, figures and supplementary figures.
6. In addition to the comment, such as "...Overexpression of PLK1, PLK4 or NEK2 did not have an obvious effect on the expression of hepatic-lineage genes (Fig. 3c). However, knockdown of PLK1, PLK4 or NEK2 led to significant inhibition of hepatic conversion (Fig. 3e and Supplementary Fig. 6a-d)....", the authors should correctly discuss the importance of PLK1, PLK2, and NEK2 in the hepatic conversion. Similar to PIM1, at least PLK1, PLK2, and NEK2 could be considered as protein kinases involved in the hepatic conversion.
7. The title may be better than old one, but "overcoming" is too strong to correctly show the findings in this study. "... by suppressing ..." may be suitable.

8. The authors' comment, such as "In this study, 17 pathways were enriched upon PIM1 overexpressing during hepatic reprogramming (Fig. 5c). We only validated two of them, including "cell cycle" and "ferroptosis". Thus, there could be other downstream signaling pathways for PIM1 during hepatic reprogramming..." is insufficient. The previous answer comment of authors, such as "As shown in Fig. 4b (updated Fig. 6f), 17 pathways were significantly changed in PIM1 overexpressors. Fifteen were liver-enriched metabolic or functional pathways (e.g., the PPAR signaling pathway, drug metabolism, pentose and glucuronate interconversions, and other similar pathways). Because we found that PIM1 promotes hepatic reprogramming, it is not surprising that pathways related to liver functions were enriched. Among the 17 pathways, we identified two non-metabolic pathways related to cell cycle progression and ferroptosis. Thus, we focused on these two pathways. We agree that ferroptosis inhibition partially contributes to the effects of PIM1. In Fig. 6g-m, we show that cell cycle also contributes to the effects of PIM1. Based on our results, we cannot exclude that any other pathways are related to the effects of PIM1." is important and thus should be included in the Results.

9. The amount of total 4E-BP1 protein is increased in hepatic reprogramming (Figure 5i) and by PIM1 overexpression (Figure 5j). Also, PIM1 overexpression does not affect the phosphorylation of 4E-BP1 (Figure 5j). These data suggest that PIM1 affects the expression of 4E-BP1, rather than phosphorylation of 4E-BP1. Also, both MYC and p-MYC are increased by PIM1 overexpression (Figure 5j), suggesting that PIM1 affects the expression of MYC, rather than phosphorylation of MYC. The data shown in Supplementary Figure 8h support this idea. The authors must correctly interpret the data shown in Figure 5i and 5j and Supplementary Figure 8h and provide agreeable conclusion. In addition, western blot data for p-MYC in Figure 5i is unclear and thus cannot show an increase of the phosphorylation of MYC in hepatic reprogramming. Also, western blot data for total 4E-BP1 protein are missing in Supplementary Figure 8h.

Reviewer #5 (Remarks to the Author):

The authors have adequately addressed the reviewer's concerns.

Detailed Responses to Reviewers' Comments

Reviewer #3:

The authors should correctly respond to the issues related to their previous answers.

1. The authors' claim that “the CKI is a powerful method that can be used for discovery of regulatory PKs” is unreasonable, because false-positive values in the confusion matrix indicate low accuracy of the CKI (Supplementary Figure 1e and 1f). Unless this point is correctly revalued, it is difficult to accept the claims made in this paper regarding the usefulness of the CKI.

Response: CKI is a model-based method, and its performance will not be higher than model-free methods that use the benchmark data set for model training. Thus, we agree to your opinion, and weaken our tone in the paper to describe the usefulness of CKI. The word “powerful” has been removed from the manuscript.

2. In Supplementary Figure 12g, the authors themselves show that transcriptome data are not necessarily essential for the detection of PK predicted from CKI. Therefore, description of the evaluation of CKI in this paper is an overstatement. The authors should correctly rewrite the abstract, introduction, and other parts of manuscript that refer to the universal usefulness of CKI.

Response: We agree to your opinion, and carefully revised the manuscript regarding the usefulness of CKI as below:

On page 2, Line 40-41, changed:

“...but also provides a **helpful** method that **might** be extended to predict central PKs involved in other biological processes.”

On page 5, Line 94-95, changed:

“...but also develops a **computational** method that **might** be **helpful** for discovery of regulatory PKs associated with other key biological processes.”

On page 11, Line 204-205, changed:

“...The successful identification of cAMP-dependent PKs as key regulators for hepatocyte maturation further **supported the reliability** of CKI.”

On page 23, Line 473-475, changed:

“...We also validated a total of five predicted central PKs involved in regulating hepatocyte maturation and hepatic reprogramming (Fig. 2k, Fig 3c, e), **indicating the reliability** of CKI.”

3. *The difference in AUC values is obtained by comparing CKI and "other methods", which is described in the abstract and introduction, is the comparison of CKI with KSEA. Thus, the authors should change "other methods" to KSEA.*

Response: We agree to your comment and revise the manuscript as below:

On page 2, Line 33-34, changed:

“...the accuracy of CKI denoted by area under the curve (AUC) is 5.2% to 29.5% higher than **KSEA**....”

On page 4, Line 75-76, changed:

“...which were higher than **other methods such as Kinase-Substrate Enrichment Analysis (KSEA)**^{21,22}....”

On page 8, Line 143-144, changed:

“...Compared to **KSEA**^{21,22}, the AUC values of CKI were 5.2% (0.8278 vs. 0.7871) and 29.5% (0.7912 vs. 0.6112) higher...”

4. *The authors' comment, such as “In this study, we focused on the early events during hepatic reprogramming. Thus, we performed CKI analysis on HDFs undergoing hepatic conversion for 2.25 days and 5 days. Though the expression of hepatic genes increased after 2.5 days induction (Supplementary Figure 4b), the expression levels of ALBUMIN and Ecadherin in HDF+FHH (5d) are far below those in HDF+FHH (14d). The protein of ALBUMIN and E-cadherin are not detectable by immunofluorescence staining (Response Figure 1). Thus, it is impossible to isolate hiHep cells by FACS sorting during the early stage of hepatic conversion.”, and Response Figure 1 should be included in the revised paper, both of which are important for understanding the experimental system.*

Response: We appreciate the reviewer for this constructive suggestion. We included this important information in the text and Supplementary Figure 4e.

On page 11, Line 218, we changed the portions shown in red:

“...These time points were chosen because liver-specific genes were starting to be induced in FHH-2.25d cells and were induced at much higher levels in FHH-5d cells. **However, the expression levels of ALBUMIN and E-Cadherin in HDF+FHH (5d) are far below those in HDF+FHH (14d), and are not detectable by immunofluorescence**

staining (Supplementary Figure 4e). It is impossible to isolate hiHep cells by FACS sorting during the early stage of hepatic conversion. Thus, the pooled cells were used for the study. HDFs transduced with GFP for 2.25 days (GFP) were used as the control for the trans-omic profiling (Fig. 3a).”

5. *shNEK2 may be correct in Supplementary Figure 6b-d. The authors should carefully check misspelling in the text, figures and supplementary figures.*

Response: We are thankful for indicating this mistake. We have corrected the labelling accordingly (Supplementary Figure 6b-d).

6. *In addition to the comment, such as “...Overexpression of PLK1, PLK4 or NEK2 did not have an obvious effect on the expression of hepatic-lineage genes (Fig. 3c). However, knockdown of PLK1, PLK4 or NEK2 led to significant inhibition of hepatic conversion (Fig. 3e and Supplementary Fig. 6a-d)...”, the authors should correctly discuss the importance of PLK1, PLK2, and NEK2 in the hepatic conversion. Similar to PIM1, at least PLK1, PLK2, and NEK2 could be considered as protein kinases involved in the hepatic conversion.*

Response: We appreciate the reviewer for this constructive suggestion. We discussed the critical roles of *PLK1*, *PLK4* or *NEK2* in hepatic reprogramming accordingly.

On page 14, Line 274, we changed the portions shown in red:

“...Although most of the candidate central PKs showed more or less effects on the regulation of hepatic-lineage genes, two of the 15 predicted central PKs, *PIM1* and *PIM2*, were validated to regulate all examined three hepatic-lineage genes in both overexpression and knockdown experiments. *PLK1*, *PLK4* and *NEK2* are also important for hepatic reprogramming, though overexpression of these genes cannot further improve the efficiency.”

7. *The title may be better than old one, but “overcoming” is too strong to correctly show the findings in this study. “... by suppressing ...” may be suitable.*

Response: We are thankful for this suggestion. We have changed the title as suggested: “*PIM1* promotes hepatic conversion by suppressing reprogramming-induced ferroptosis and cell cycle arrest”.

8. The authors' comment, such as "In this study, 17 pathways were enriched upon PIM1 overexpressing during hepatic reprogramming (Fig. 5c). We only validated two of them, including "cell cycle" and "ferroptosis". Thus, there could be other downstream signaling pathways for PIM1 during hepatic reprogramming...." is insufficient. The previous answer comment of authors, such as "As shown in Fig. 4b (updated Fig. 6f), 17 pathways were significantly changed in PIM1 overexpressors. Fifteen were liver-enriched metabolic or functional pathways (e.g., the PPAR signaling pathway, drug metabolism, pentose and glucuronate interconversions, and other similar pathways). Because we found that PIM1 promotes hepatic reprogramming, it is not surprising that pathways related to liver functions were enriched. Among the 17 pathways, we identified two non-metabolic pathways related to cell cycle progression and ferroptosis. Thus, we focused on these two pathways. We agree that ferroptosis inhibition partially contributes to the effects of PIM1. In Fig. 6g-m, we show that cell cycle also contributes to the effects of PIM1. Based on our results, we cannot exclude that any other pathways are related to the effects of PIM1." is important and thus should be included in the Results.

Response: We appreciate the reviewer for this constructive suggestion. We improved the description accordingly.

On page 16, Line 327, we changed the portions shown in red:

"...Here, we performed KEGG- and GO-based enrichment analyses, and found that 17 pathways were significantly changed upon PIM1 overexpression (Fig. 5c and Supplementary Fig. 8d). Fifteen were liver-enriched metabolic or functional pathways (e.g., the PPAR signaling pathway, drug metabolism, pentose and glucuronate interconversions, and other similar pathways). As we found that PIM1 promotes hepatic reprogramming, it is not surprising that pathways related to liver functions were enriched. Interestingly, we observed that two non-metabolic pathways, cell cycle (KEGG ID: hsa04110) and ferroptosis (KEGG ID: hsa04216), were enriched in response to PIM1 overexpression (Fig. 5c), which is consistent with a previous report..."

On page 25, Line 520, we have discussed this:

"...In this study, 17 pathways were enriched upon PIM1 overexpressing during hepatic reprogramming (Fig. 5c). We only validated two of them, including "cell cycle" and "ferroptosis". Thus, there could be other downstream signaling pathways for PIM1 during hepatic reprogramming..."

9. The amount of total 4E-BP1 protein is increased in hepatic reprogramming (Figure 5i) and by PIM1 overexpression (Figure 5j). Also, PIM1 overexpression does not affect the phosphorylation of 4E-BP1 (Figure 5j). These data suggest that PIM1 affects the expression of 4E-BP1, rather than phosphorylation of 4E-BP1. Also, both MYC and p-MYC are increased by PIM1 overexpression (Figure 5j), suggesting that PIM1 affects the expression of MYC, rather than phosphorylation of MYC. The data shown in Supplementary Figure 8h support this idea. The authors must correctly interpret the data shown in Figure 5i and 5j and Supplementary Figure 8h and provide agreeable conclusion. In addition, western blot data for p-MYC in Figure 5i is unclear and thus cannot show an increase of the phosphorylation of MYC in hepatic reprogramming. Also, western blot data for total 4E-BP1 protein are missing in Supplementary Figure 8h.

Response: We appreciate the reviewer for this constructive suggestion. PIM1 has been found to phosphorylate MYC at Ser62 to stabilize the MYC protein (Y Zhang, et al., Oncogene, 2008). Thus, the increased protein level of MYC is most probably caused by the increased phosphorylation of MYC protein upon PIM1 overexpression. 4E-BP1 has been reported to be stabilized by multiple kinases (Berta Pons, et al., Oncology Reports, 2012). Whether 4E-BP1 could be stabilized by PIM1 should be further investigated in the future, which is not the scope of this manuscript. We discussed this in the text and improved the description accordingly.

We also improved the western blot data for p-MYC (Figure 5j), and performed additional experiment to show the phosphorylation levels of 4E-BP1 protein upon PIM1 knockdown (Supplementary Figure 8h).

On page 17, Line 346, we changed the portions shown in red:

“...PIM1 has been found to promote cell proliferation through various downstream signaling pathways, including phosphorylation of 4E-BP1 to promote cap-dependent translation, phosphorylation on Ser62 to stabilize MYC protein, and activation of mTOR signaling. Here, we also found that increased protein and phosphorylation levels of 4E-BP1 and MYC occurred with increased expression of PIM1 during hepatic reprogramming (Fig. 5i). Overexpression of PIM1 further increased the phosphorylation levels of 4E-BP1, MYC, and an mTOR substrate P70S6K (Fig. 5j). The increased MYC phosphorylation on Ser62 possibly stabilized MYC protein as previously reported. 4E-BP1 has also been reported to be stabilized by multiple kinases. whether 4E-BP1 could be stabilized by PIM1 should be further investigated in the future. Furthermore, we also observed a decrease in MYC expression and phosphorylation after PIM1 knockdown during hepatic reprogramming (Supplementary Fig. 8h)...”

Reviewer #5:

The authors have adequately addressed the reviewer's concerns.